# Dynamic reconfiguration of functional brain networks during working memory training

Karolina Finc 1✉, Kamil Bonna[1,2], Xiaosong He 3, David M. Lydon-Staley 3,4, Simone Kühn[5,6], Włodzisław Duch[1,2] & Danielle S. Bassett 3,7,8,9,10,11

The functional network of the brain continually adapts to changing environmental demands. The consequence of behavioral automation for task-related functional network architecture remains far from understood. We investigated the neural reflections of behavioral automation as participants mastered a dual n-back task. In four fMRI scans equally spanning a 6-week training period, we assessed brain network modularity, a substrate for adaptation in biological systems. We found that whole-brain modularity steadily increased during training for both conditions of the dual n-back task. In a dynamic analysis, we found that the autonomy of the default mode system and integration among task-positive systems were modulated by training. The automation of the n-back task through training resulted in non-linear changes in integration between the fronto-parietal and default mode systems, and integration with the subcortical system. Our findings suggest that the automation of a cognitively demanding task may result in more segregated network organization.

[1] Centre for Modern Interdisciplinary Technologies, Nicolaus Copernicus University, Toruń, Poland. [2] Department of Informatics, Faculty of Physics, Astronomy and Informatics, Nicolaus Copernicus University, Toruń, Poland. [3] Department of Bioengineering, School of Engineering and Applied Science, University of Pennsylvania, Philadelphia, PA, USA. [4] Annenberg School for Communication, University of Pennsylvania, Philadelphia, PA, USA. [5] Lise Meitner Group for Environmental Neuroscience, Max Planck Institute for Human Development, Berlin, Germany. [6] University Medical Center Hamburg–Eppendorf, Hamburg, Germany. [7] Department of Electrical & Systems Engineering, School of Engineering and Applied Science, University of Pennsylvania, Philadelphia, PA, USA. [8] Department of Neurology, Perelman School of Medicine, University of Pennsylvania, Philadelphia, PA, USA. [9] Department of Physics & Astronomy, School of Arts and Sciences, University of Pennsylvania, Philadelphia, PA, USA. [10] Department of Psychiatry, Perelman School of Medicine, University of Pennsylvania, Philadelphia, PA, USA. [11] Santa Fe Institute, Santa Fe, NM 87501, USA. ✉email: finc@umk.pl

The brain constantly adjusts its architecture to meet the demands of the ever-changing environment. Such neural adaptation spans multiple time scales, being observed over seconds to minutes during task performance[1–5], over days to weeks during learning[6–8], and over years during development[9]. Like many other complex biological systems, the adaptability of the brain is supported by its modular structure[10]. Intuitively, modularity allows for dynamic switching between states of segregated and integrated information processing, whose balance is constantly adjusted to meet the requirements of our cognitive faculties[11,12]. Understanding the patterns of these adjustments and determining the rules that explicate their relation to human behavior is one one of the most important challenges for cognitive neuroscience.

It is hypothesized that simple, highly automated sensorimotor tasks can be maintained by a highly segregated brain organization, while more complex and cognitively demanding tasks require integration between multiple subnetworks[13]. Indeed, switching from a segregated to a more costly integrated network architecture is consistently reported as human participants transition to challenging tasks with heavy cognitive load[1–5]; in contrast, network organization during simple motor tasks remains highly segregated[3,4]. Whether shifts toward network integration depend on the level of task complexity or on the level of task automation remains to be delineated[12]. Is it possible that a complex, but fully automated task, can be performed without the need for costly network integration?

Longitudinal studies, during which participants are scanned multiple times while mastering a specific task, can shed light on patterns of network adaptation related to learning and task automation[12]. For example, Bassett et al.[7] showed that training on a visuomotor task over the course of 6 weeks leads to increased autonomy between task-relevant subnetworks in motor and visual cortices. In another study, Mohr et al.[8] found increased segregation of the default-mode system after short-term visuomotor training. Collectively, these findings suggest that an increase in network segregation and a decrease in integration may constitute a natural consequence of task automation. However, these results refer to the training of simple motor tasks, which do not require extensive network integration, in contrast to complex tasks involving higher-order cognitive functions such as cognitive control[12]. The consequence of complex cognitive task automation on the balance between network segregation and network integration remains unknown.

In this study, we investigated whether mastering a demanding working memory task affects the balance between network segregation and integration during task performance. Does effortless performance of the demanding cognitive task lead to the same increase in network segregation that is characteristic of simple motor tasks[3,8]? Is the breakdown of network segregation during the changing demands of the cognitive task still necessary when the cognitive task is automated? Finally, do we observe stronger separation of subnetworks relevant to cognitive control when tracking dynamical brain network reorganization throughout the course of training? To address these questions, participants underwent four functional magnetic resonance imaging (fMRI) scans while performing an adaptive dual n-back task taxing working memory over a 6-week training period. The dual n-back task consisted of visuospatial and auditory tasks that were performed simultaneously[14]. In the visuospatial portion of the task, participants had to determine whether the location of the stimulus square presented on the screen was the same as the location of the square n-back times in the sequence; in the auditory portion of the task, participants had to determine whether the heard consonant was the same as the consonant they heard n-back times in the sequence. To ensure that participants mastered

the task due to training, and not simply due to a repeated exposure to the task, we compared their performance to an active control group. While participants from both the experimental and the control groups performed the same version of the dual n-back task, with interleaved 1-back and 2-back blocks, inside the fMRI scanner, only the experimental group trained their working memory using an adaptive version of the task in 18 training sessions outside the scanner. We examined network reconfiguration using static functional network measures to distinguish distinct task conditions, and using dynamic network measures to study fluctuations of network topology across short task blocks.

First, we investigated global changes in network segregation (modularity) across different task conditions as compared with rest. In line with the aforementioned research, we expected modularity to decrease during dual n-back task performance compared with rest, and also to decrease as the demands of the n-back task increased. We also hypothesized that over the course of training, network segregation during the n-back task would increase, and the extent of demand-related modularity change would decrease. In the systems relevant to working memory performance—the frontoparietal and the default mode systems[15]—we expected an increase in autonomy throughout the course of training. To verify this hypothesis, we utilized previously developed dynamic network methods[7] to assess the recruitment and integration of the default-mode and frontoparietal systems. Finally, we expected that changes in network architecture would correspond to the level of task automation and training progress.

Our results demonstrate that adult human brain functional networks not only reorganize during a working memory task but also can be modulated by the level of expertise in the task. After working memory training, brain networks are more segregated. The increase in segregation is visible at the whole-brain level for static networks, and also evidenced by an increased segregation of the default-mode and task-positive systems when considering dynamic changes in network organization. Automation of the working memory task is accompanied by nonlinear changes in coupling between the default-mode and frontoparietal systems and engagement of the subcortical system. Together, these results shed new light on the mechanisms underlying brain network reorganization accompanying the automation of performance on cognitively demanding tasks.

## Results

**Behavioral changes during training**. Behavioral improvement in the task can either occur as a result of training or occur in response to repeated exposure to a task across multiple scanning sessions. To distinguish the effect of intensive working memory practice and task automation from the effect of repeated exposure, we employed an active control group. When participants from the experimental group underwent the challenging, adaptive, dual n-back working memory training, participants from the control group performed a single, non-adaptive, 1-back working memory task (Fig. 1).

The dual n-back task (1-back and 2-back conditions) was performed in the scanner on the first day of the experiment (Naive), after 2 weeks of training (Early), after 4 weeks of training (Middle), and after 6 weeks of training (Late). We measured participant performance as a $d'$, a measure based on signal detection theory that takes into account both response sensitivity and response bias[16] (see Methods). Better cognitive performance is characterized by higher values of $d'$. We expected that participants from the experimental group would exhibit a substantial increase of $d'$ during training, particularly for the 2-back condition in comparison with the 1-back condition, the latter being easy to master even without extensive training.

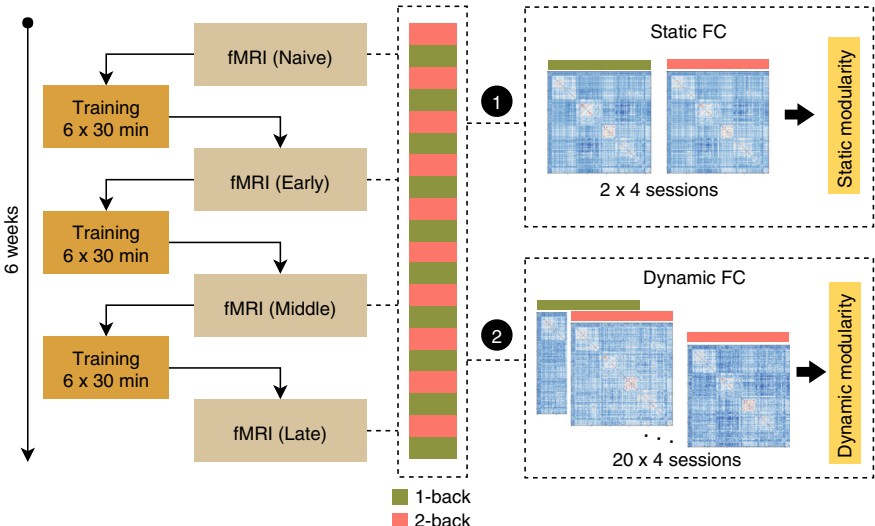

**Fig. 1 Study design.** (Left) The dual n-back working memory task was performed in the scanner on the first day of the experiment (Naive), after 2 weeks of training (Early), after 4 weeks of training (Middle), and after 6 weeks of training (Late). (Right) We investigated (1) changes in static modularity across task conditions (1-back versus 2-back), and (2) dynamic fluctuations in network community structure from block to block.

Using multilevel modeling (see Methods), we found that participants had significantly different $d'$, depending on the training stage (Naive, Early, Middle, Late), condition (1-back vs. 2-back), and group (experimental vs. control). Specifically, we found a significant session × condition × group interaction ($\chi^2(3) = 9.39$, $p = 0.02$; Fig. 2). The greatest improvement was observed in the experimental group when comparing "Naive" to "Late" training phases during the 2-back condition (mean 43.2% $d'$ improvement; paired $t$ test, two-sided: $t(20) = -9.17$, $p < 0.0001$, Bonferroni-corrected). For comparison, the control group exhibited a 24.3% increase in $d'$ during the 2-back condition (paired $t$ test, two-sided: $t(20) = -6.45$, $p < 0.0001$, Bonferroni-corrected). The increase in $d'$ was significantly larger for the experimental group than for the control group (two-sample $t$ test two-sided: $t(20) = -4.12$, $p = 0.004$, Bonferroni-corrected; Fig. 2d).

In the 1-back condition, the experimental group displayed a 12.2% increase in $d'$ (paired $t$ test, two-sided: $t(20) = -3.18$, $p = 0.02$, Bonferroni-corrected); no improvement was found in the control group (paired $t$ test, two-sided: $t(22) = -1.91$, $p = 0.28$, Bonferroni-corrected) (see Fig. 2c). The change in $d'$ during the 1-back condition did not differ between the two groups (two-sample $t$ test, two-sided: $t(39.64) = -0.52$, $p = 0.47$). Interestingly, in the experimental group we observed no significant difference in performance between the 1-back condition and the 2-back condition after training (paired $t$ test, two-sided: $t(20) = 0.02$, $p = 0.98$), while in the control group, the difference in performance between conditions remained substantial (paired $t$ test, two-sided: $t(20) = 4.91$, $p = 0.0016$, Bonferroni-corrected). This finding suggests that the 2-back condition, which was much more effortful before training ("Naive" phase), was performed effortlessly after training, at the same level as the 1-back task.

In sum, the results demonstrate that the experimental group gradually improved in behavioral performance measured during the fMRI scanning sessions, and that this improvement was significantly greater than the corresponding effect in the control group. We also replicated these findings using an alternative measure of behavior, penalized reaction time (pRT) which incorporates a measure of accuracy (see Supplementary Fig. 3 and Supplementary Methods).

**Whole-brain network modularity changes.** To establish whether complex working memory task training leads to increased network segregation at the whole-brain level, we investigated network modularity during different sessions and load conditions. Here, we employed a common community detection algorithm known as modularity maximization[17], which we implemented using a Louvain-like locally greedy algorithm. The modularity quality function to be optimized encodes the extent to which the network can be divided into nonoverlapping communities. Intuitively, a community is a group of densely interconnected nodes with sparse connections to the rest of the network[17]. Modularity is a relatively simple measure of segregation, with high values indicating greater segregation of the brain into nonoverlapping communities, and low values indicating lesser segregation. Because modularity depends upon the network's total connectivity strength, we normalized each modularity score by dividing it by the mean of the corresponding null distribution calculated on a set of randomly rewired versions of the original networks[18] (see Methods for details).

Functional network modularity may vary, depending on the difficulty of the task. Several studies have reported a reduction in modular structure during demanding n-back conditions[2,3,5]. Here, we first investigated the differences between the high-demand 2-back condition and the low-demand 1-back condition as compared with a baseline resting-state scan acquired during the first session ("Naive") for all subjects. Using multilevel modeling, we found a significant main effect of condition ($\chi^2(2) = 84.13$, $p < 0.00001$). Planned contrast analysis revealed that network modularity during the dual n-back task was lower than network modularity during the resting state (paired $t$ test, two-sided: $\beta = -0.20$, $t(88) = -11.37$, $p < 0.00001$). Furthermore, modularity was significantly reduced during the 2-back condition relative to the 1-back condition (paired $t$ test, two-sided: $\beta = -0.08$, $t(296) = -2.60$, $p = 0.01$; Fig. 3). We note that the results reported here use a functional brain parcellation composed of 264 regions of interests provided by Power et al.[19]; in robustness tests, we performed the same analyses using the Schaefer parcellation, and obtained the similar results (see Supplementary Fig. 16).

The modularity of functional brain network architecture decreases appreciably during challenging task conditions, but is the breakdown in modularity still present when the demanding task is mastered? To address this question, we tested whether modularity during the dual n-back task changed, depending on the session, task condition, and group. Using a multilevel model

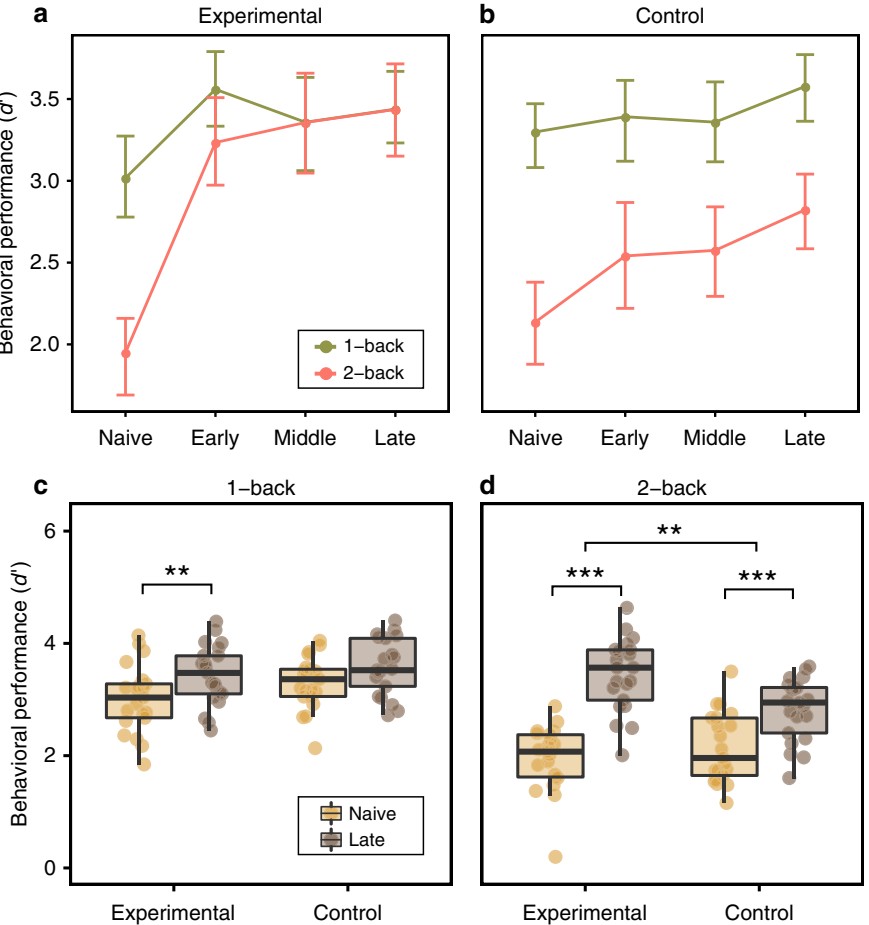

**Fig. 2 Behavioral performance modulated by training. a, b** Line plots representing mean behavioral performance measured as $d'$, calculated for all training phases (Naive, Early, Middle, and Late), dual n-back conditions (1-back and 2-back), and groups (**a** experimental, n = 21, and **b** control, n = 21). We found a significant interaction effect between the session, condition, and group. After training, the experimental group exhibited no difference in behavioral performance between the 1-back and 2-back conditions. Dots represent mean values; error bars represent 95% confidence intervals. **c** No significant difference between groups was found for $d'$ reduction (from Naive to Late sessions) during the 1-back task condition. **d** The experimental group showed a significant reduction in $d'$ compared with the control group during the challenging 2-back condition (two-sample $t$ test, two-sided; $p = 0.004$). Boxes represent the interquartile range (IR) between 25th and 75th percentiles. The thick line in the center of each box represents the median. The upper and lower error bars display the largest and smallest values within 1.5 times IR above 75th percentile and below 25th percentile, respectively. ***$p < 0.001$ Bonferroni-corrected; **$p < 0.05$ Bonferroni-corrected. Source data are provided as a Source Data file.

(see Methods), we found a significant main effect of session ($\chi^2$(2) = 19.40, $p = 0.0002$) and of group ($\chi^2$(1) = 6.62, $p = 0.01$). However, the experimental and control groups did not differ by session ($\chi^2$(1) = 1.44, $p = 0.69$), nor did we observe a significant session by condition interaction ($\chi^2$(1) = 1.50, $p = 0.68$). A planned contrast comparison showed that participants' whole-brain functional network modularity significantly increased from "Naive" to "Middle" sessions (paired $t$ test, two-sided: $\beta = 0.15$, $t$(114) = 2.61, $p = 0.01$) and from "Naive" to "Late" sessions (paired $t$ test, two-sided: $\beta = 0.24$, $t$(114) = 4.05, $p = 0.0001$; Fig. 4a, b). The experimental group showed a higher network modularity ($M = 3.09$) than the control group ($M = 2.87$). To summarize, we showed that the modularity of the functional brain network generally increased during the training period. However, the degree to which modularity changed between load conditions remained stable. Groups did not differ significantly in the change of modularity. These results suggest that the functional brain network shifts toward a more segregated organization as a result of behavioral improvement after training and also after repeated exposure to the task. Although network modularity increased to a similar extent in both conditions, the demand-dependent change in modularity remained stable. One

could interpret these results as suggesting that a general increase in modularity reflects the fact that less expensive information processing is required within segregated brain subsystems after training of the complex task.

To further explore the changes in modularity that might be specific to each group and condition, we performed additional analyses comparing modularity measured before and after training (Fig. 4c, d). Specifically, we employed separate paired $t$ tests to investigate differences in modularity for each group and condition between "Naive" and "Late" sessions. We found a significant increase of modularity in the experimental group in the 1-back condition (paired $t$ test, two-sided: $t$(20) = −3.66, $p = 0.006$, Bonferroni-corrected) and in the 2-back condition (paired $t$ test, two-sided: $t$(20) = −3.33, $p = 0.013$, Bonferroni-corrected). The increase in modularity observed in the control group was not significant for either the 1-back condition (paired $t$ test, two-sided: $t$(20) = −2.35, $p = 0.11$, Bonferroni-corrected) or the 2-back condition (paired $t$ test, two-sided: $t$(20) = −1.88, $p = 0.28$, Bonferroni-corrected). The change of modularity from "Naive" to "Late" sessions did not significantly differ between groups for the 1-back condition (two-sample $t$ test, two-sided: $t$(39.88) = −0.80, $p = 0.42$) or for the 2-back condition (two-sample $t$ test,

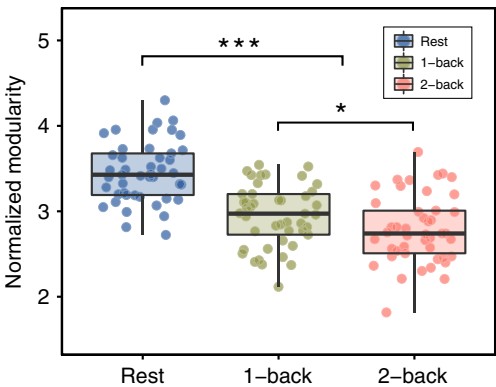

**Fig. 3 Modularity differences between resting and dual n-back task conditions in "Naive" session.** Results of paired $t$ tests (two-sided), comparing task conditions ($n = 64$). Whole-brain modularity was higher during the resting state than during the dual n-back task ($p < 0.00001$), and decreased as demands heightened from the 1-back to the 2-back condition ($p = 0.01$). Boxes represent the interquartile range (IR) between 25th and 75th percentiles. The thick line in the center of each box represents the median. The upper and lower error bars display the largest and smallest values within 1.5 times IR above 75th percentile and below 25th percentile, respectively. ***$p < 0.001$, *$p < 0.05$ (planned contrasts analysis; uncorrected). Source data are provided as a Source Data file.

two-sided: $t(39.99) = -1.05$, $p = 0.30$). These results indicate that the experimental group displays increased network modularity for both task conditions when moving from "Naive" to "Late" sessions, suggesting that network segregation may be a consequence of the 6-week working memory training. While the same effect was not present in the control group, we did not observe a significant group × session interaction, and therefore further work is needed to inform our conclusions.

Behavioral gains resulting from working memory training differed across participants, suggesting the existence of individual differences in learning capabilities. Therefore, we also tested whether the increase of modularity observed during the 2-back condition in the experimental group was correlated with behavioral performance after training as measured by a decrease in $d'$. However, we did not find a significant relationship between these two variables (Pearson's correlation coefficient $r = 0.08$, $p = 0.71$; Supplementary Fig. 20). This finding suggests that the change of modularity is a general consequence of training and may not reflect individual differences in behavioral improvement.

Our results confirmed the existence of a decrease in modularity during increased cognitive demands. However, changes in modularity during training were not different across conditions or experimental groups. A significant increase in modularity from "Naive" to "Late" sessions was found for the 1-back and 2-back conditions for the experimental group, which suggests the enhancement of network segregation associated with task automation.

**Dynamic reorganization of large-scale systems**. The modular architecture of functional brain networks is not static, but instead can fluctuate appreciably over task blocks. Here, we used a dynamic network approach to answer the question of whether large-scale brain systems change in their fluctuating patterns of expression during training. Based on a previous study of motor sequence learning[7], we expected that systems relevant to working memory—the frontoparietal and the default mode—would become more autonomous over the 6 weeks of working memory training (Fig. 5a). To formally test our expectation, we investigated the dynamic reconfiguration of the network's modular

structure as subjects switched between blocks of the dual n-back task. Pooling across conditions and sessions, we constructed a multilayer network model of the data, in which each block corresponds to a unique layer, each region corresponds to a node, and each functional connection corresponds to an edge. We then employed a multilayer community detection algorithm that estimates each node's module assignment in each network layer[20]. The presence of fluctuations in community structure across task blocks is indicated by variable assignments of nodes to modules across layers. For each subject and session, we summarized these data in a module allegiance matrix **P**, where each element $P_{ij}$ represents a proportion of blocks for which node $i$ and node $j$ were assigned to the same module. We also applied a normalization to allegiance matrices, to remove any potential bias introduced by differences in the number of nodes within each subsystem. Following the functional cartography framework described by Mattar et al.[21], we used **P** to calculate the recruitment of all 13 large-scale systems, as well as the pairwise integration among them (see Methods for details). We selected these measures to maintain consistency with the methodology used in a previous study on the effects of motor sequence training on the dynamics of functional brain networks[7]. Recruitment is defined for each system separately, while integration is calculated for pairs of systems. Intuitively, high recruitment indicates that nodes of the system are consistently assigned to the same module across different layers; this consistency reflects the non-random nature of brain dynamics, in which a functional module is persistently recruited for a task. High integration indicates that pairs of nodes (where one region of the pair is located in one system and the other region of the pair is located in the other system) are frequently classified in the same module across layers). We used a multilevel model to test whether recruitment and integration coefficients differed between scanning sessions and experimental groups.

First, we examine dynamic topological changes in the frontoparietal and default-mode systems, which were directly related to our hypothesis. Using a multilevel model, we observed a significant session × group interaction effect when considering changes in the recruitment of the frontoparietal system during training ($\chi^2(3) = 9.03$, $p = 0.028$; Fig. 5b). The largest increase in frontoparietal recruitment was observed in the experimental group when comparing "Early" to "Late" training phases (paired $t$ test, two-sided: $\beta = -0.07$, $t(120) = -2.892$, $p = 0.027$, Bonferroni-corrected; Fig. 5b). No significant changes from "Naive" to "Late" training phases were observed in the control group (paired $t$ test, two-sided: $\beta = -0.03$, $t(120) = -1.169$, $p = 1$, Bonferroni-corrected). Turning to an examination of the default mode, we found a significant main effect of session ($\chi^2(3) = 24.17$, $p < 0.0001$) and of group ($\chi^2(1) = 3.96$, $p = 0.046$) on system recruitment (Fig. 5c). However, the interaction effect between session and group was not significant ($\chi^2(3) = 2.66$, $p = 0.48$). Planned contrasts revealed that the default mode recruitment increased steadily in both groups, and we observed the largest increase between "Naive" and "Late" sessions (paired $t$ test, two-sided: $\beta = 0.09$, $t(123)$ 5.00, $p < 0.0001$). The experimental group displayed a higher default-mode recruitment than the control group (paired $t$ test, two-sided: $t(165.6) = -3.03$, $p = 0.003$). We found a significant session × group interaction effect on the integration between the frontoparietal and default-mode systems ($\chi^2(3) = 14.25$, $p = 0.0025$) (Fig. 5d). The integration between these two systems decreased from "Naive" to "Late" sessions only in the experimental group (paired $t$ test, two-sided: $\beta = 0.07$, $t(120) = 4.37$, $p = 0.0002$, Bonferroni-corrected). However, groups differed from "Naive" to "Early" (two-sample $t$ test, two-sided: $\beta = 0.07$, $t(120) = 2.16$, $p = 0.03$) and from "Early" to "Middle" sessions (two-sample $t$ test, two-sided: $\beta = -0.06$, $t(120) = -2.70$, $p = 0.02$,

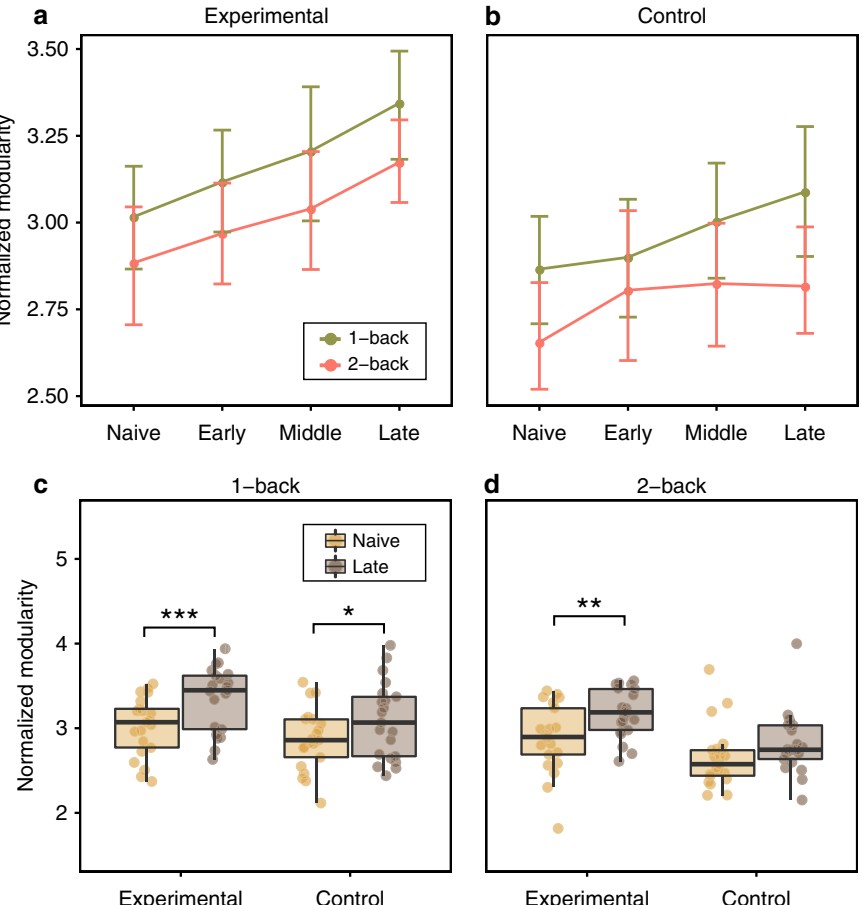

**Fig. 4 Modularity differences across the task, sessions, and groups. a, b** Line plots representing the mean values of modularity for each scanning session (Naive, Late, Middle, and Late) and condition, separately for **a** the experimental group ($n = 21$) and **b** the control group ($n = 21$). Dots represent mean values; error bars represent 95% confidence intervals. **c, d** Modularity changes from "Naive" to "Late" sessions for the 1-back condition and the 2-back condition (paired $t$ test, two-sided). We found a significant increase of modularity in the experimental group **c** in the 1-back condition ($p = 0.006$), and **d** in the 2-back condition ($p = 0.013$). Boxes represent the interquartile range (IR) between 25th and 75th percentiles. The thick line in the center of each box represents the median. The upper and lower error bars display the largest and smallest values within 1.5 times IR above 75th percentile and below 25th percentile, respectively. ***$p < 0.01$ Bonferroni-corrected; **$p < 0.05$ Bonferroni-corrected; *$p < 0.05$ uncorrected; paired $t$ test; two-sided. Source data are provided as a Source Data file.

Bonferroni-corrected): whereas the experimental group displayed an inverted U-shaped curve of integration with training, the control group displayed the opposite pattern. Collectively, these results suggest that the increase of frontoparietal system recruitment and the decrease of integration between the default-mode and frontoparietal systems reflect training-specific changes in dual n-back task automation. In contrast, the increase in default-mode system recruitment may reflect more general effects of behavioral improvement, as it was observed in both experimental and control groups.

Next, we asked whether changes in dynamic topology could be observed in other large-scale systems. Using multilevel modeling, we observed three distinct types of changes occurring over time regardless of the group ($p < 0.05$, FDR-corrected; Fig. 6a–c): an increase in system recruitment, (2) an increase in the integration between task-positive systems, and (3) a decrease in the integration between default mode and task-positive systems (Supplementary Fig. 7, Supplementary Tables 1 and 2). First, we observed an increase in the recruitment beyond the default-mode system—in salience—and auditory systems (Supplementary Fig. 7a–c). Second, we observed an increase in the integration between task-positive systems, including frontoparietal and salience, dorsal attention and salience, and dorsal attention and

cingulo-opercular (Supplementary Fig. 7d–f). Third, for the default-mode system, we observed a decrease in integration with other task-positive systems: salience and cingulo-opercular (Supplementary Fig. 7g–i). In addition, we also observed a decrease in integration between the memory and somatomotor systems, and between the default-mode and auditory systems (Supplementary Fig. 7j, k). We observed a similar pattern of changes for the Schaefer parcellation (Supplementary Figs. 17 and 18a). These results suggest that the increase of within-module stability, the increase of default-mode system independence from task-positive systems, and the decrease of integration between task-positive systems reflect general effects of task training.

We also investigated the relationship between across-session change in system recruitment or integration and across-session change in behavioral performance for all large-scale systems. For both brain and behavioral variables, we measured the change from the first ("Naive") to the last ("Late") training sessions (see Fig. 7a; Supplementary Table 6). We found a significant positive correlation between change in behavior, as operationalized by a change in $d'$ (2-back minus 1-back), and change of the default mode ($r = 0.33$, $p = 0.03$, uncorrected) and salience ($r = 0.34$, $p = 0.03$; uncorrected) systems recruitment. Greater behavioral improvement was also associated with a higher increase of

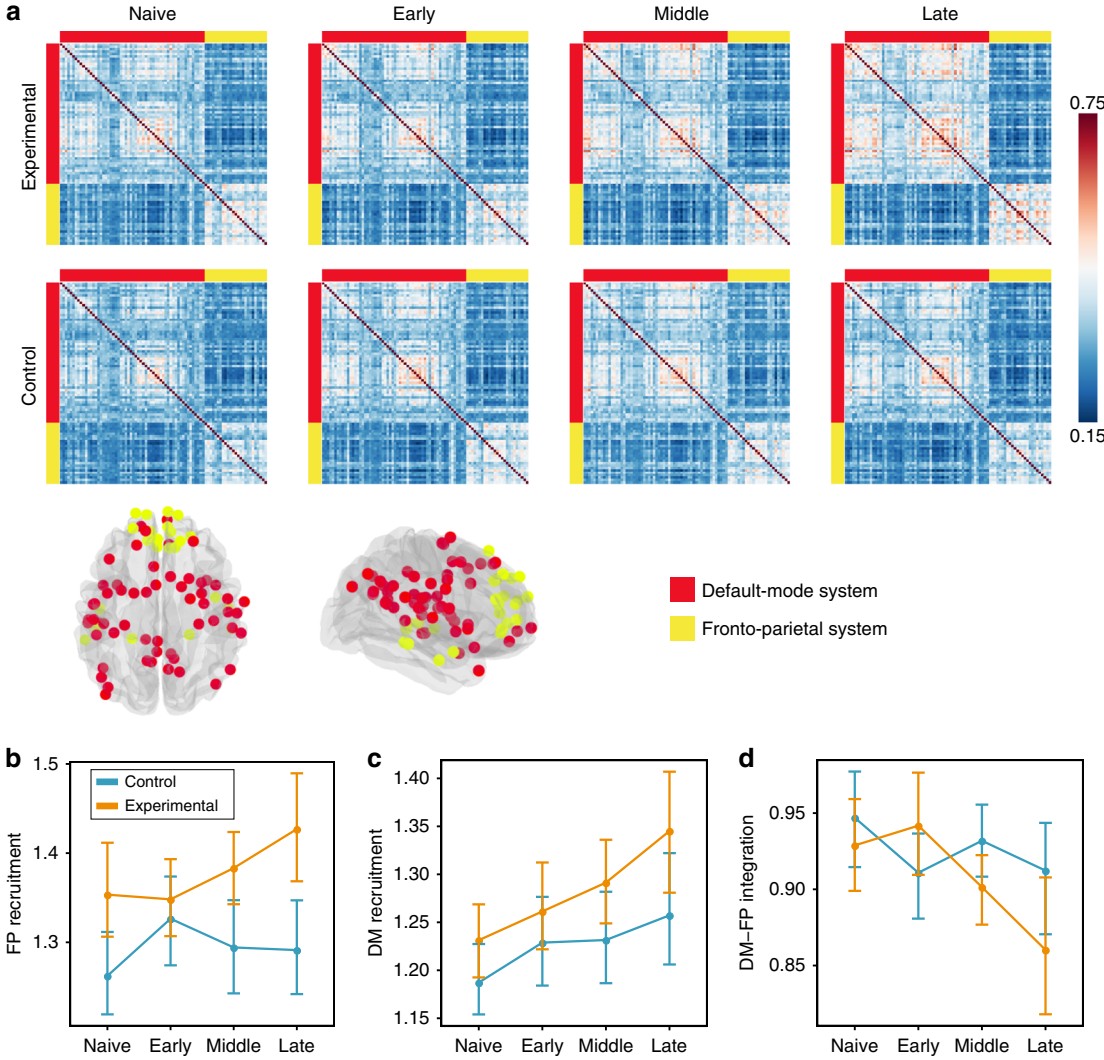

**Fig. 5 Changes in module allegiance of the frontoparietal (FP) and default-mode (DM) systems. a** Module allegiance matrices for the default-mode and frontoparietal systems. Each $ij$th element of the matrix represents the probability that node $i$ and node $j$ are assigned to the same module within a single layer of the multilayer network. **b** Only the experimental ($n = 21$) group exhibited increases in frontoparietal recruitment across sessions. **c** Both experimental and control groups ($n = 21$) exhibited increases in default mode recruitment between "Naive" and "Late" stages of training. **d** In both groups, the integration between the frontoparietal and default-mode systems decreased from "Naive" to "Late" sessions, but groups differed in the pattern of integration changes between "Naive" to "Middle" sessions (see also Supplementary Fig. 6). We observed a similar pattern of changes when considering allegiance matrices calculated based on signed functional connectivity matrices (Supplementary Fig. 14) and alternative brain parcellation[58] (Supplementary Fig. 17). Dots represent mean values; error bars represent 95% confidence intervals. Source data are provided as a Source Data file.

integration between frontoparietal and salience systems ($r = 0.35$, $p = 0.02$, uncorrected) and a higher decrease of integration between default-mode and task-positive systems: frontoparietal ($r = -0.31$, $p = 0.04$, uncorrected) and salience ($r = -0.41$, $p = 0.006$, uncorrected). Analogous relationships for default-mode recruitment and default-mode–frontoparietal integration with behavioral improvement were observed for an alternative measure of performance (pRT; Supplementary Fig. 9a). Note that the correlation for the change in the $d'$ measure has opposite sign when compared with the correlation with the change of pRT, consistent with the fact that these two measures have different interpretations (the lower pRT, the better; the higher the $d'$, the better). In summary, a higher increase of stability in the default-mode and salience systems, together with a decrease of default mode–task-positive system integration may support behavioral improvement in the task, regardless of whether the task was additionally trained or not.

Finally, we also observed session × group interaction effects beyond the default mode and frontoparietal systems ($p < 0.05$, uncorrected; Fig. 6d, e; Supplementary Tables 3 and 4). Specifically, in the experimental group, we observed a nonlinear change in the integration of the subcortical system with the dorsal attention, ventral attention, cingulo-opercular, and auditory systems. An initial increase in integration with the subcortical system (from "Naive" to "Early") was followed by a decrease in the integration at later time intervals. Interestingly, we observed the reverse pattern for the change in integration between the subcortical and default-mode systems: the integration first decreased from "Naive" to "Early" sessions, and then increased from "Early" to "Middle" sessions for the experimental group (Supplementary Fig. 8; Supplementary Table 5). The pattern of changes in integration also differed between the groups, particularly so for the integration between cingulo-opercular and memory systems, cingulo-opercular and uncertain systems,

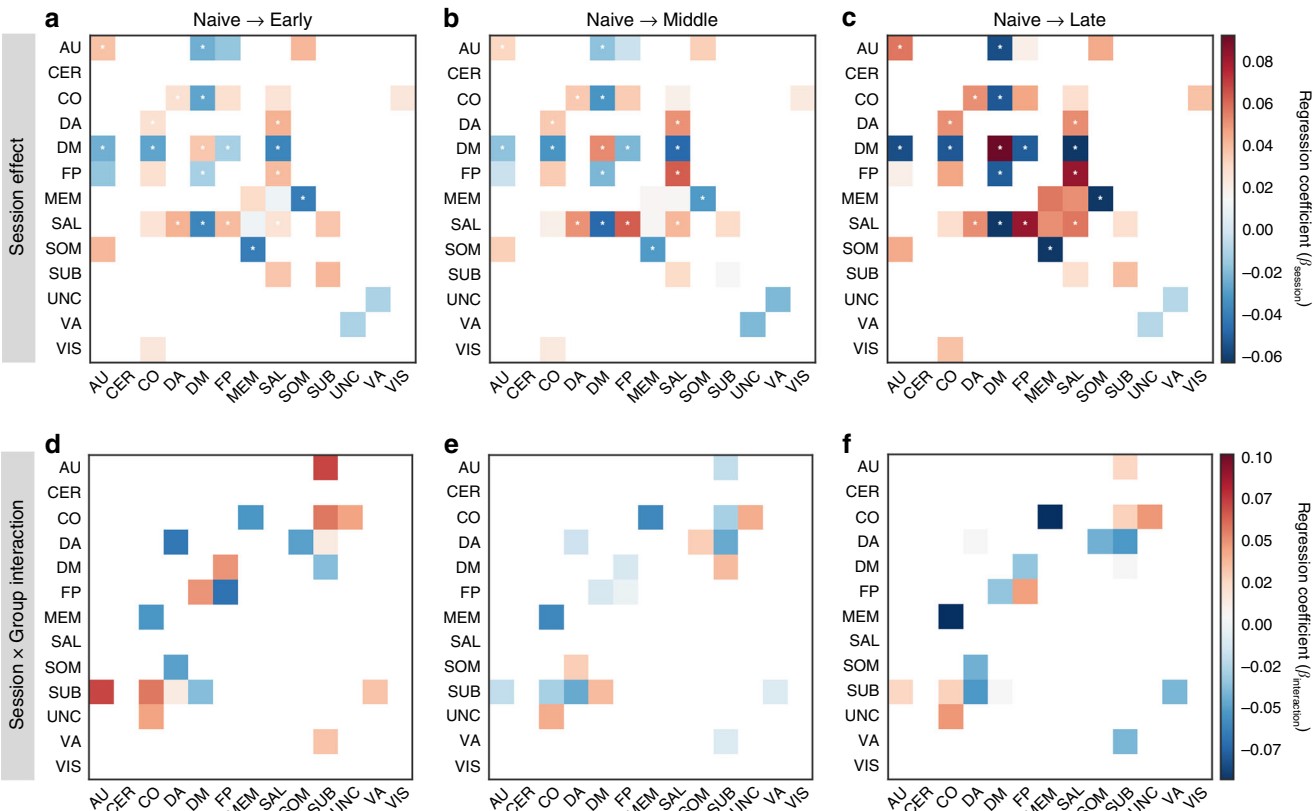

**Fig. 6 Changes of the recruitment and integration of large-scale systems.** Colored tiles represent all significant effects ($p < 0.05$, uncorrected; *$p < 0.05$ FDR-corrected). (top panel) Here, we display the significant main effects of session. Tile color codes a linear regression coefficient ($\beta$), for all main session effects: **a** from "Naive" to "Early", **b** from "Naive" to "Middle", and **c** from "Naive" to "Late". (bottom panel) Here, we display the significant session × group interaction effects. Tile color codes a linear regression coefficient between groups and sessions: **d** from "Naive" to "Early", **e** from "Naive" to "Middle", and **f** from "Naive" to "Late". AU: auditory, CER: cerebellum, CO: cingulo-opercular, DM: default mode, DA: dorsal attention, FP: frontoparietal, MEM: memory, SAL: salience, SOM: somatomotor, SUB: subcortical, UNC: uncertain, VA: ventral attention, VIS: visual. Source data are provided as a Source Data file.

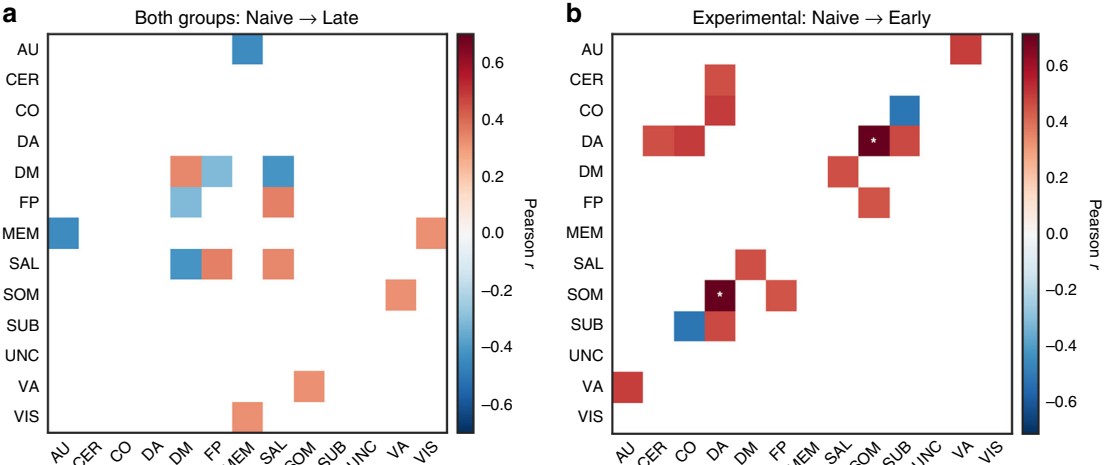

**Fig. 7 Relationship between the change in network dynamics and the change in behavior.** Colored tiles represent all significant correlations ($p < 0.05$, uncorrected; *$p < 0.05$ FDR-corrected). **a** Pearson correlation coefficient ($r$) between the across-session changes in recruitment (or integration) and the across-session changes in $d'$ ($\Delta d'$) observed for both the experimental and control groups. **b** Relationship between the changes in recruitment (or integration) and the changes in $d'$ during early phase of training of the experimental group. AU: auditory, CER: cerebellum, CO: cingulo-opercular, DM: default mode, DA: dorsal attention, FP: frontoparietal, MEM: memory, SAL: salience, SOM: somatomotor, SUB: subcortical, UNC: uncertain, VA: ventral attention, VIS: visual. Source data are provided as a Source Data file.

and dorsal attention and somatomotor systems. These results suggest that task automation during initial stages of working memory training might also be supported by an increased communication between subcortical and other large-scale systems.

We further tested whether changes in systems recruitment or integration from "Naive" to "Early" sessions were associated with performance improvement displayed by the experimental group. Interestingly, we found that the behavioral change was positively correlated with change of integration between multiple systems, in particular: dorsal attention and somatomotor, dorsal attention and subcortical, frontoparietal and somatomotor, dorsal attention and cingulo-opercular, salience, and default mode. In contrast, the increase of integration of subcortical system and cingulo-opercular systems was negatively correlated with the change in task performance (Fig. 7b; Supplementary Table 7). This pattern of associations between behavioral and network changes suggests that inter-systems communication might be necessary for efficient task performance during initial stages of training.

In summary, we observed two patterns of dynamic changes in network topology following working memory training. The first pattern reflects improved behavioral performance, and is characterized by a gradual increase in default-mode autonomy and in the integration between task-positive systems. The second pattern reflects changes related to task automation specifically in the experimental group and is characterized by nonlinear changes in default mode–frontoparietal integration, and in the integration with the subcortical system.

## Discussion

In this study, we aimed to verify the hypothesis that training on an effortful cognitive task—a dual n-back—increases the segregation of task-related functional brain networks. We found that whole-brain modularity significantly differed between task conditions, being the highest in the resting state, lower in the 1-back condition, and even lower in the 2-back condition. In the experimental group, modularity increased in response to working memory training. We also observed two patterns of changes in the dynamic network topology following training: (i) a gradual increase in the segregation of default mode and task-positive systems, and (ii) a nonlinear change in the default mode–frontoparietal integration and integration of the sub-cortical system. The general behavioral improvement in the task in response to training was positively correlated with an increase in the recruitment of the default-mode system and a decrease in its integration with the frontoparietal system. Collectively, these findings suggest that segregation of the default-mode and task-positive systems supports general improvement in the task, while dynamic communication of the default mode with the fronto-parietal and subcortical systems supports more specific network changes related to automation of the working memory task.

We observed that modularity during the resting state was higher than during performance of the dual n-back task, and the modularity during the low-demand task condition (1-back) was higher than the modularity during the high-demand task condition (2-back). Our results are consistent with previous studies providing evidence that network segregation is lowest (while integration is highest) during a demanding n-back task, when compared with a less demanding motor task or resting state[3,4]. The observed difference between working memory loads is consistent with a previous study from Vatansever et al.[2] who reported higher modularity during the 3-back condition compared with the 0-back condition, and also consistent with a previous study from Finc et al.[5] who reported higher modularity during the 2-back condition compared with the 1-back condition. Collectively,

the findings also support the Global Workspace Theory (GWT)[13], by showing that less demanding, highly automated tasks can be performed within segregated modules, while more challenging tasks require integration between multiple modules.

Despite the consistency between our findings and prior work, it is important to note that these previous studies did not address the question of whether a fully mastered demanding cognitive task would still require a costly integrated workspace or could instead be executed within specialized brain modules. Here, our study expands upon prior work by offering the first evidence supporting the latter hypothesis. We observed that although modularity of the network generally increased through n-back training, as measured during both 1-back and 2-back conditions in the experimental group, the modularity difference between the two conditions was preserved. This finding suggests that training resulted in the increase of the baseline network segregation during the task, which supports our hypothesis that mastered cognitive tasks can be executed within a segregated network. Modularity measured during the high-demand 2-back condition after training exceeded the modularity during the low-demand 1-back condition before training. However, even if the baseline network segregation increased after the training, some level of modularity breakdown during increasing cognitive demands seems to be induced.

Interestingly, we did not observe differences between the experimental and control group in the increase of network modularity. The control group displayed a small increase of modularity in the 1-back condition, suggesting that the segregation of the functional brain network may increase rapidly, also in response to repeated exposure to the task. The control group performed the dual n-back task four times during scanning sessions, which resulted in a small behavioral improvement. This result suggests that the increase of network segregation may be sensitive to varying intensity of training in the task. Future studies with a larger sample size should examine whether such gradation exists.

The modular structure of functional brain networks is not static, but instead undergoes dynamic reconfiguration throughout a range of cognitive processes[1,22–26]. Recently developed dynamical approaches to study brain networks are sensitive to the temporal nature of the underlying neural signal, and therefore can be used to probe the fluctuating patterns of connectivity elicited by task performance. Using just such a dynamical approach, Bassett et al.[7] showed that the modular structure of human brain functional networks fluctuates appreciably during motor-visual learning, and that the degree of fluctuations changes during a 6-week training paradigm. Task-relevant, motor and visual networks exhibited increasing autonomy as the duration of training increased, marking the emergence of automatic behavioral responses. In light of this prior work, we hypothesized that networks relevant to working memory function—including the frontoparietal and default-mode systems—would increase their autonomy after extensive training on a working memory task.

Here, we used a multilayer community detection algorithm to determine whether modular structure of large-scale systems change in response to n-back training. We further applied multilevel modeling[27] to test for possible group and session differences in the dynamic network measures while controlling for differences in individual baseline values. In testing our hypothesis, we held in mind the observations of previous studies, which have noted that the frontoparietal and default-mode systems can both cooperate and compete during tasks that require cognitive control, such as the n-back task[15,28]. Understanding the nature of interactions between these two systems is therefore essential for explaining the neural adaptation that occurs in response to evolving cognitive demands. It is also not known whether

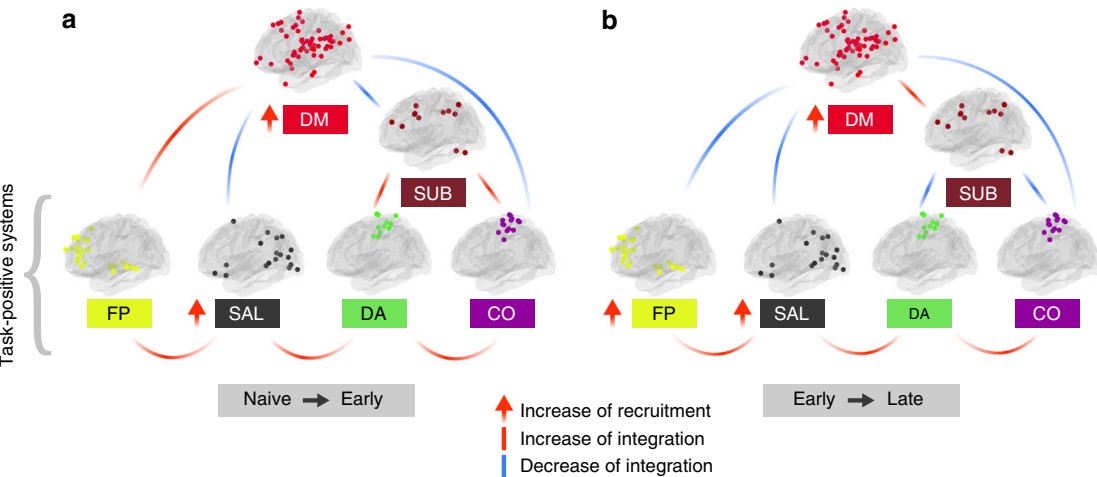

**Fig. 8 Schematic summarizing the main changes in recruitment and integration of large-scale systems observed over the course of working memory training.** We observed a gradual increase of the integration between task-positive systems (FP: frontoparietal, SAL: salience, DA: dorsal attention, CO: cingulo-opercular), greater recruitment of the default-mode (DM) system, and decreased DM–CO and DM–SAL integration. In the early phase of training (**a**), the experimental group displayed an increased FP–DM integration, and increased integration of the subcortical (SUB) system with the DA and CO systems. In the late stage of training (**b**), the FP system reduced its integration with the DM system and the SUB system increased its integration with the DM system,while decreasing coupling with task-positive systems.

dynamic interactions between these two networks may evolve during cognitive training. Using dynamic network metrics, we showed that the default-mode system increased its recruitment in both groups, indicating that regions within this system were coupled with other communities less often. The experimental group displayed an increased frontoparietal recruitment and an inverted U-shaped curve of integration between the default-mode and frontoparietal systems with training. Enhanced default mode intra-communication and decreased inter-communication with the frontoparietal system were associated with better behavioral outcomes after training (Fig. 8). We also observed significant changes in dynamic network topology beyond the frontoparietal and default-mode systems. In particular, regardless of the group, we observed an increased recruitment of the salience and auditory systems, decreased integration between the default-mode and other task-positive systems (including salience and cingulo-opercular), and increased integration between task-positive systems (including frontoparietal and salience, dorsal attention and salience, dorsal attention and cingulo-opercular). These results suggest the existence of the trade-off between segregation and integration: whereas segregation increases between some systems, the integration increases or decreases between others.

Some studies suggest that competitive interactions between the task-positive frontoparietal system and the task-negative default-mode system might be essential for higher-order cognitive functions[15,28]. The frontoparietal system is composed of spatially distributed brain areas, including the lateral prefrontal cortex, anterior cingulate, and inferior parietal cortex[29]. Its activity is commonly linked to the performance of tasks requiring cognitive control, such as the n-back working memory task[29,30]. Prior work offers evidence that the frontoparietal system is highly flexible and dynamically interacts with other systems in response to the changing demands of cognitive tasks[1,23]. In contrast, the default-mode system exhibits high activity during internally directed cognition, such as mind wandering and autobiographical memory[31]. The default-mode system is composed of spatially distributed brain areas, including the medial prefrontal cortex, posterior cingulate, lateral parietal cortex, and both lateral and medial temporal cortices[31,32]. The default mode's activity is frequently anticorrelated with the activity of systems that engage in

demanding cognitive tasks, such as the frontoparietal and dorsal attention systems[33]. Recent studies, however, challenge a common view about existing antagonism between default mode and frontoparietal systems, suggesting that the interaction between these two systems is necessary for efficient behavioral control[34,35]; default-mode regions display a positive coupling with task-positive brain systems during working memory task performance[36,37] and may dynamically switch their connections to support inter-module communication in high-demand n-back task conditions[5,38]. Our observations expand upon prior studies by demonstrating that the increase in default-mode segregation and decrease of integration between the default-mode and frontoparietal systems may be an indicator of behavioral improvement during working memory training. Moreover, the previous study reported the relationship between the default-mode connectivity changes and static modularity changes during n-back task[5]. In our exploratory analysis, we also showed that default-mode recruitment fluctuated between task conditions, and was significantly higher in the 1-back condition than in the 2-back condition (Supplementary Fig. 19) and, similar to modularity, increased steadily in both groups. Here, we also observed a positive relationship between the change in default-mode recruitment and change of modularity from "Naive" to "Late" session (Supplementary Fig. 21). As we did not observe the relationship between changes of modularity and behavioral improvement, we may conclude that studying the dynamics of modular network structure enables a better prediction of behavioral outcomes in response to training.

Our results are also consistent with prior observations that the default-mode and frontoparietal systems may interact in a task-dependent manner with the salience, cingulo-opercular, and dorsal attention systems[15,39]. Bressler and Menon[39] proposed a model whereby efficient cognitive control is supported by the dynamic switching between functionally segregated frontoparietal and default-mode systems mediated by cingulo-opercular and salience systems. Cocchi et al.[15] proposed that task-related reconfiguration is possible through flexible interactions within and between overlapping meta-systems: (i) the executive meta-systems,responsible for the processing of sensory information, and (ii) the integrative meta-system,responsible for flexible

integration of brain systems. These two meta-systems are composed of transient coupling between three large-scale systems: the frontoparietal system, the cingulo-opercular/salience system, and the default-mode system. During high-demand task conditions, the executive meta-system is formed by extensive interactions between frontoparietal and cingulo-opercular/salience systems, and the default-mode system is more segregated and less integrated with the frontoparietal system[40]. Our results extend these findings by presenting the evolving reconfigurations of large-scale networks during mastery of the working memory task. We showed that regardless of the group the default-mode system reduced coupling with the cingulo-opercular and salience systems. These results suggest that increased segregation of the default mode and task-positive networks may be a consequence of more efficient task performance. A similar pattern of changes was observed across two different subdivisions of the cortex into systems (Power and Schaefer), together suggesting that the salience and cingulo-opercular systems that are thought to be responsible for switching between antagonistic frontoparietal and default-mode systems, appear to be more integrated with the frontoparietal system and less integrated with the default-mode system. This pattern of relations may be due to diminished requirements for switching between these two systems when the task is well learned.

Similar to modularity, the lack of group differences in the pattern of these changes suggests that the increased the default-mode autonomy and increased integration of task-positive systems might be related to a general improvement in task performance. Such behavioral improvement, although much smaller than in the experimental group, was also observed in the control group during the 2-back condition. As participants performed the task four times in the scanner, they inevitably trained the task to a small extent. The presence of network reorganization in the control group may suggest that changes in the default-mode autonomy and integration of task-positive systems occur relatively fast, even when the training is not intense. As participants of our study were scanned in 2-week intervals, we could not capture what behavioral improvement is necessary to invoke such network reorganization. To better understand the dynamics of these neuroplastic changes, future studies should examine day-to-day network reorganization in response to training with different intensities.

We also observed that groups differed in patterns of changes in the subcortical system coupling. Specifically, the experimental group displayed an inverted U-shaped curve of changes in (i) the integration between the subcortical system and the dorsal attention system, and (ii) the integration between the ventral attention system and the cingulo-opercular system; notably, the control group displayed the opposite pattern. We observed the opposite effect for coupling between the subcortical and default-mode systems. Nonlinear changes in subcortical activity were also observed in previous studies of the effects of working memory training[41]. Consistent with our results, Kühn et al.[41] found that activity in the subcortical regions increased after 1 week of working memory training, and decreased after 50 days of training. Previous studies suggested that subcortical activity can mediate changes in working memory ability[42]. Because an inverted U-shaped curve of changes in frontoparietal activity was also observed following working memory training[41,43], we speculate that subcortical activity may influence changes in the frontoparietal system. Yet, results based on observation of brain activity changes cannot provide information on how these two systems interact. Our results show that in the initial training phase, the subcortical system switched coupling from task-positive systems to the default mode system. We observed the opposite pattern for the frontoparietal system, which instead first

increased and then decreased its interaction with the default-mode system. We speculate that the subcortical system supports segregation of the task-positive and default-mode systems. Future studies using effective connectivity approach could examine whether such a cause and effect relationship exists.

The dynamic network approach extends our understanding of training-related changes in brain function. Studies focusing on changes in brain activity during a working memory training reported a decrease of task-positive systems activation[41,43], commonly interpreted as a reflection of increased neural efficiency within systems engaged in the task[44]. Here, we reported a similar effect using a standard GLM-based approach (see Supplementary Figs. 10 and 11; Supplementary Tables 8 and 9). We also showed that our findings on the dynamic network changes cannot be simply explained by the changes in brain activity (Supplementary Fig. 12). The frontoparietal system dynamically interacts with other large-scale systems[15], and it is reasonable to expect that working memory training might influence interactions in the whole network. We observed training-related increases in the segregation of the default mode and task-positive systems that suggest more efficient and less costly processing within these systems after training. Greater segregation of the default-mode system and task-positive systems and smaller integration between these systems were associated with behavioral performance improvement. Moreover, we showed that an increase of integration between multiple large-scale systems in early phase of training was related to a greater behavioral improvement in the experimental group, indicating that some level of network integration is necessary when the task is not fully automated. Taken together, the dynamic network approach provides a unique insight into the plasticity and dynamics of the human brain network.

## Methods

**Subjects.** Fifty-three healthy volunteers (26 female; mean age: 21.17; age range: 18–28 years) were recruited from the local community through word-of-mouth and social networks. All participants were right-handed, had normal or corrected-to-normal vision, and had no hearing deficits. Seven participants did not complete the study: one due to brain structure abnormalities detected at the first scanning session, and six due to not completing the training procedure. The final sample consisted of 46 participants who completed the entire training procedure, participated in all four fMRI scanning sessions, and had neither history of neurological or psychiatric disorders nor gross brain structure abnormalities. After the first fMRI scan, participants were matched by sex and randomly assigned to one of the two training groups: experimental and control (see the next section Experimental Procedures). Each group consisted of 23 subjects with no group differences in age (two-sample $t$ test: $t(42.839) = 0.22$, $p = 0.83$) or fluid intelligence (two-sample $t$ test: $t(42.882) = 0.51$, $p = 0.61$), as measured by Raven's Advanced Progressive Matrices (RAPM)[45]. Informed consent was obtained in writing from each participant, and ethical approval for the study was obtained from the Ethics Committee of the Nicolaus Copernicus University Ludwik Rydygier Collegium Medicum in Bydgoszcz, Poland, in accordance with the Declaration of Helsinki.

**Experimental procedures.** The study was performed at the Centre for Modern Interdisciplinary Technologies, Nicolaus Copernicus University in Toruń (Poland). Each participant who completed the entire study procedure attended a total of 24 meetings at the laboratory. During the first meeting, participants were familiarized with the study procedure and timeline, and were asked to provide basic demographic information and informed consent. During the second meeting, participants performed fluid intelligence testing with RAMP[45]. Then, participants were scheduled for fMRI testing, which was performed before training, after 2 weeks of training, after 4 weeks of training, and after 6 weeks of training. Each fMRI session was scheduled to be on the same day of the week and at the same hour for each participant. These schedules varied in exceptional cases (holidays, illness of participant, emergency). However, scanning procedures were always performed between 24 h to 48 h after the last training session. After the first fMRI session, participants were randomly assigned to one of two training groups: (1) experimental, which trained working memory with an adaptive dual n-back task[14], and (2) a passive control group which interchangeably performed an auditory and spatial 1-back task. We included this second group to control for differences in the effect of training on task performance and fMRI signatures driven by repeated exposure to a task.

Two versions of the dual n-back task were used: (1) an adaptive dual n-back was used in the training sessions of the experimental group only, and (2) an identical dual n-back task with two conditions (1-back and 2-back) used during fMRI scanning of both groups. Both scanning and training versions of the dual n-back task consisted of visuospatial and auditory tasks performed simultaneously. Visuospatial stimuli consisted of eight light blue squares presented sequentially for 500 ms on the 3 × 3 grid with a white fixation cross in the middle of the black screen; auditory stimuli consisted of eight polish consonants (b, k, w, s, r, g, n, and z) played sequentially in headphones. Participants were asked to indicate by pressing a button with their left index finger whether the letter heard through the headphones was the same as the letter n-back in the sequence, and by pressing a button with their right index finger to indicate whether the square on the screen was in the same location as the square n-back in the sequence.

In the training version of the task, the n level of the dual n-back task increased adaptively when participants achieved 80% correct responses in the trial, and the n level decreased when participants made >50% errors in the trial. After each trial, the n level achieved by a participant was recorded, and the mean n level during each of 18 training session was used later to calculate the total training progress (Supplementary Figs. 1 and 2). Participants from the control group performed a single 1-back with auditory or visuospatial stimuli variants. To minimize boredom of participants, the order of the 1-back variants was randomly selected at the beginning of each training session. Therefore, each participant from the control group had the same number of training trials on single auditory and visuospatial n-back tasks. Participants completed a total of 18 sessions (30 min each) under the supervision of the experimenter. Each participant completed 20 blocks (each consisting of 20 + n trials, depending on the n level achieved by the participant) of the n-back task during each training session. The study was double-blind; the experimenter performing the fMRI examination was not aware of the group assignment of the participants, and participants were not aware that the study was designed in a way that there were two groups (experimental and control). The apparatus used in the study consisted of two 17" Dell Inspiron Laptops, and two pairs of Sennheiser headphones. Stimulus delivery was controlled by a Python adaptation of the dual n-back task used by Jaeggi et al.[14] (http://brainworkshop.sourceforge.net/).

In the fMRI scanning version of the task, participants performed the dual n-back task with two levels of difficulty: 1-back and 2-back. Each session of the task consisted of 20 blocks (30 s per block; 12 trials with 25% of targets) of alternating 1- and 2-back conditions. To enable dynamic network comparison across blocks, we did not add any systematic variation to block length and block order. The instruction screen was displayed for 4000 ms before each block, informing the participant of the upcoming condition. Both visual and auditory stimuli were presented in a pseudorandom order. Participants were asked to push the button with their right thumb if the currently presented square was in the same location as the previous square (1-back) or two squares back in the sequence (2-back) and, at the same time, push the button with their left thumb when the currently played consonant was the same as the previous consonant (1-back) or two consonants back (2-back). The participants had 2000 ms to respond, and were instructed to respond as quickly and accurately as possible. The experimental protocol execution and control (stimulus delivery and response registration) employed version 17.2. of Presentation software (Neurobehavioral Systems, Albany, NY), as well as MRI compatible goggles (visual stimulation), headphones (auditory stimulation), and response grips (response registration) (NordicNeuroLab, Bergen, Norway). Before each scanning session, participants performed a short dual n-back training session outside the fMRI scanner to (re-)familiarize them with the rules of the task.

All participants received equal monetary remuneration (200 PLN) for study participation together with a radiological description and a CD containing their anatomical brain scans.

## Data acquisition

Neuroimaging data were collected using a GE Discovery MR750 3 Tesla MRI scanner (General Electric Healthcare) with a standard 8-channel head coil. Structural images were collected using a three-dimensional high-resolution T1-weighted gradient-echo (FSPGR BRAVO) sequence (TR = 8.2 s, TE = 3.2 ms, FOV = 256 mm, flip angle = 12 degrees, matrix size 256 × 256, voxel size = 1 × 1 × 1 mm, 206 axial oblique slices). Functional scans were obtained using a T2*-weighted gradient-echo, echo-planar imaging (EPI) sequence sensitive to BOLD contrast (TR = 2000 ms, TE = 30 ms, FOV = 192 mm, flip angle = 90 degrees, matrix size = 64 × 64, voxel size 3 × 3 × 3 mm, 0.5-mm gap). For each functional run, 42 axial oblique slices were acquired in an interleaved acquisition scheme, and 5 dummy scans (10 s) were obtained to stabilize magnetization at the beginning of the EPI sequence. Resting-state (10 min 10 s, 305 volumes) data were acquired at the beginning of each scanning session. During the resting state, participants were asked to focus their eyes on the fixation cross in the middle of the screen. The dual n-back task data (11 min 30 s; 340 volumes) were acquired using the same data acquisition settings (see Experimental Procedures for the task description).

## Behavioral performance

To measure behavioral performance in the dual n-back scanning sessions, we incorporated d', a signal detection theory statistic[16]. This measure combines both response sensitivity and response bias. For every subject, session, task condition, and stimulus modality, we first divided all responses into four categories: (1) hits—button press for targets, (2) misses—lack of response for

targets, (3) false alarms— button press for non-targets, and (4) correct rejections— lack of response for non-targets. We defined hit rate H and false alarm rate F as:

$$H = \frac{\#\text{hits}}{\#\text{hits} + \#\text{misses}} \qquad (1)$$

$$F = \frac{\#\text{false alarms}}{\#\text{false alarms} + \#\text{correct rejections}} \qquad (2)$$

We calculated d' measure as:

$$d' = Z(H) - Z(F), \qquad (3)$$

where $Z(x)$ is the inverse of the cumulative Gaussian distribution. To get finite values of d' for the situations in which H or F was equal to 0 or 1, we used modified values of either 0.01 or 0.99 instead. For each participant, we calculated average d' for both modalities to represent a cumulative measure of performance during the dual n-back task. We also calculated behavioral performance using an alternative measure, penalized reaction time (pRT), which incorporates a measure of accuracy (see Supplementary Fig. 4 for variability changes for these measures).

## Data processing

After converting from DICOM to NifTI format, functional and anatomical data were structured according to the BIDS (Brain Imaging Data Structure) standard[46] and validated with BIDS Validator (https://bids-standard.github.io/bids-validator/). Neuroimaging data was preprocessed using fMRIPrep 1.1.1[47] a Nipype[48]-based tool. See Supplementary Methods for details on anatomical data processing. Functional data were slice time corrected using 3dTshift from AFNI v16.2.07[49] and motion corrected using MCFLIRT (FSL v5.0.9[50]). This process was followed by co-registration to the corresponding T1w using boundary-based registration[51] with 9 degrees of freedom, using bbregister (FreeSurfer v6.0.1). Motion correcting transformations, BOLD-to-T1w transformation and T1w-to-template (MNI) warp were concatenated and applied in a single step using antsApplyTransforms (ANTs v2.1.0) employing Lanczos interpolation.

Physiological noise regressors were extracted by applying CompCor[52]. Principal components were estimated for the two CompCor variants: temporal (tCompCor) and anatomical (aCompCor). A mask to exclude signal with cortical origin was obtained by eroding the brain mask,ensuring that it only contained subcortical structures. Six tCompCor components were then calculated including only the top 5% variable voxels within that subcortical mask. For aCompCor, six components were calculated within the intersection of the subcortical mask and the union of the CSF and WM masks calculated in T1w space, after their projection to the native space of each functional run. Frame-wise displacement[53] (FD) was calculated for each functional run using the implementation of Nipype. The internal operations of fMRIPrep use Nilearn[54], principally within the BOLD-processing workflow. For more details of the pipeline see https://fmriprep.readthedocs.io/en/latest/workflows.html.

Non-smoothed functional images were denoised using Nilearn[54] and Nistats. We implemented voxel-wise confound regression by regressing out (1) signals from six aCompCor components, (2) 24 motion parameters representing 3 translation and 3 rotation timecourses, their temporal derivatives,and quadratic terms of both, (3) outlier frames with FD > 0.5 mm and DVARS (Derivative of rms VARiance over voxelS)[55] with a threshold of ±3 SD, together with their temporal derivatives, (4) task effects and their temporal derivatives[56], and (5) any general linear trend. Time series were filtered using a 0.008–0.25-Hz band-pass filter. We excluded four high motion participants (two from the control group, and two from the experimental group) with a mean FD larger than 0.2 mm and more than 10% of outlier volumes in any scanning session (Supplementary Fig. 5).

## Functional connectivity estimation

Functional connectivity is a measure of the statistical relation between time series of spatially distinct brain regions. Time series can be defined as signals from single voxels or as the mean of the signals from anatomically or functionally defined groups of voxels, also known as brain parcels[57]. Here, we used a functional brain parcellation composed of 264 regions of interests (ROIs) provided by Power et al.[19]. This parcellation was based on meta-analysis, and has previously been used in many studies focused on task-based network reorganization[2,5,23]. To validate our results, we also used a 300-ROI parcellation provided by Schaefer et al.[58], which is based on transitions of functional connectivity patterns.

We created N × N correlation matrices by calculating the Pearson's correlation coefficient between the mean signal time-course of region i and the mean signal time-course of region j, for all pairs of ROIs (i, j). We retained only positive correlations for further analysis. In the case of the dual n-back task, we employed a weighted correlation measure, to control for delays due to the hemodynamic response function (HRF)[56]. In this procedure, we first convolved task block regressors with the HRF and applied a filter to retain only positive values of the resultant time series. Then, original time series were filtered according to the task condition and positive values of the HRF-convolved time series. Next, the weighted correlation coefficient was calculated between the concatenated block time series, with weights taken from the corresponding HRF-convolved signals. Finally, Fisher's transformation was employed to convert Pearson's correlation coefficients to normally distributed z-scores. This procedure resulted in 264 × 264 (Power parcellation) and 300 × 300 (Schaefer parcellation) correlation matrices for each

subject, session, and task condition (resting state, 1-back, 2-back). For the dynamic network analyses, we calculated the weighted correlations for each block of the n-back task, resulting in $264 \times 264 \times 20$ and $300 \times 300 \times 20$ matrices, where the third dimension represents the number of task blocks (20 interleaved blocks of 1-back and 2-back).

**Static modularity**. To calculate the extent of whole-brain network segregation, we employed a Louvain-like community detection algorithm[59] to optimize a common modularity quality function[17]. This algorithm partitions the network into communities, where nodes in a given community are highly interconnected among themselves, and sparsely interconnected to the rest of the network. The modularity quality index, $Q$, to be optimized was defined as follows:

$$Q_S = \frac{1}{2\mu} \sum_{ij} (A_{ij} - \gamma V_{ij}) \delta(g_i, g_j), \quad (4)$$

where $\mu = \frac{1}{2} \sum_{ij} A_{ij}$ is the total edge weight of the network, $A_{ij}$ is the strength of the edge between node $i$ and node $j$, and $\gamma$ is the structural resolution parameter. The Kronecker delta function $\delta(g_i, g_j)$ equals one if nodes $i$ and $j$ belong to the same module, and equals zero otherwise. The term $V_{ij}$ represents the connectivity strength expected by chance in the configuration null model:

$$V_{ij} = \frac{k_i k_j}{2m}, \quad (5)$$

where $k_i$ and $k_j$ are the weighted degrees of nodes $i$ and $j$, respectively, and $m = \frac{1}{2} \sum_{ij} A_{ij}$ is the sum of all nodal weighted degrees.

Since the Louvain algorithm is non-deterministic, we run it 100 times, and then consider the network partition with the highest modularity score across these runs. It is important to note that the values of graph theoretical metrics can vary markedly depending on the sum of connection strengths in the network[60]. To take this effect into account, we normalized each individual modularity value against a set of modularity values calculated for randomly rewired networks[18]. For this purpose, we created 100 null networks using random rewiring of each original functional network. Then, modularity scores were calculated for each null network, thereby creating a null distribution. Finally, we normalized modularity values by dividing them by the mean of the corresponding null distribution.

**Multilayer modularity**. To calculate measures of recruitment and integration, we performed multilayer modularity maximization used a generalized Louvain-like community detection algorithm introduced by Mucha et al.[20]. This algorithm allows the optimization of a modularity quality function on a network with multiple layers. In our study, networks calculated for each separate block were considered as consecutive layers of the multilayer network. For each subject, session, and multilayer network, we ran 100 optimizations of the modularity quality function, defined as:

$$Q_{ML} = \frac{1}{2\mu} \sum_{ijsr} [(A_{ijs} - \gamma_s V_{ijs}) \delta_{sr} + \delta_{ij} \omega_{sr}] \delta(g_{is}, g_{ir}), \quad (6)$$

where $A_{ijs}$ represents the element of the adjacency matrix at slice $s$, $V_{ijs}$ represents the element of the null model matrix at slice $s$, $g_{ir}$ provides the community assignment of node $i$ in slice $r$, $\mu = \frac{1}{2} \sum_{js} \kappa_{jr}$ is the total edge weight of the network, where $\kappa_{js} = k_{js} + c_{js}$ is the strength of node $j$ in slice $s$, the $k_{js}$ is the interslice strength of node $j$ in slice $s$, and $c_{js} = \sum_r \omega_{jsr}$. For all slices, we used the Newman–Girvan null model, also known as the configuration model, defined as:

$$V_{ijs} = \frac{k_{is} k_{js}}{2m_s}, \quad (7)$$

where $m_s = \frac{1}{2} \sum_{ij} A_{ijs}$ is the total edge weight of slice $s$. In this optimization, there are two free-parameters: $\gamma_s$ and $\omega_{jsr}$. The parameter $\gamma_s$ is the structural resolution parameter for slice $s$, and the parameter $\omega_{isr}$ represents the connection strength between node $j$ in slice $s$ and node $j$ in slice $r$. These two parameters can be used to tune the size of communities within each layer and the number of communities detected across all layers, respectively. Here, in line with previous studies we set $\gamma = 1$[21]. Due to the interleaved nature of our experimental design, $\omega = 1$ for slices from the same task condition, and $\omega = 0.5$ for slices from different task conditions.

**Network diagnostics**. Multilayer community detection results in a single-module assignment $N \times T$ matrix, where each matrix element represents the module assignment of a given node for a given slice. To summarize the dynamics of module assignments for each subject and session, we calculated an $N \times N$ module allegiance matrix, $P$, where the element $P_{ij}$ represents the fraction of network layers for which node $i$ and node $j$ belong to the same community[7,21]:

$$P_{ij} = \frac{1}{OT} \sum_{o=1}^{O} \sum_{t=1}^{T} a_{i,j}^{k,o}, \quad (8)$$

where $O$ is the number of repetitions of the multilayer community detection algorithm (here, $O = 100$), and $T$ is the number of slices (here 20 task blocks). For

each optimization $o$ and slice $t$,

$$a_{i,j}^{k,o} = \begin{cases} 0, \text{ if nodes } i \text{ and } j \text{ are in the same module} \\ 1, \text{ otherwise.} \end{cases} \quad (9)$$

To characterize the dynamics of large-scale systems recruitment and integration, we employed methods of functional cartography[7,21]. These measures allow us to summarize how often regions from the system of interest are assigned to the same module. We can define the recruitment of system $S$ as:

$$R_S = \frac{1}{n_S^2} \sum_{i \in S} \sum_{j \in S} P_{i,j}. \quad (10)$$

The recruitment of system $S$ is high when regions within the system tend to be assigned to the same module throughout all task blocks. Similarly, we can define the integration coefficient between system $S_k$ and system $S_l$ as:

$$I_{S_k S_l} = \frac{1}{n_{S_k} n_{S_l}} \sum_{i \in S_k} \sum_{j \in S_l} P_{ij}. \quad (11)$$

Systems of interest are highly integrated when regions belonging to two different systems are frequently assigned to the same community.

To remove potential bias introduced by the differences in the number of nodes within each system, we used permutation approach to normalize values of recruitment and integration coefficients. For each subject and session, we created $N_{perm} = 1000$ null module allegiance matrices by randomly permuting correspondence between ROIs and large-scale systems. We then calculated functional cartography measures for all permuted matrices. This procedure yielded null distributions of recruitment and integration coefficients resulting solely from the size of each system. In order to obtain normalized values of $R_s$ and $I_{S_k S_l}$, we divided them by the mean of the corresponding null distribution. We also calculated recruitment and integration coefficients based on multilevel community detection for signed networks (Supplementary Methods, Supplementary Figs. 13–15).

**Statistical modeling**. Due to the nested nature of the study data, we used two-level (trials nested within participants) and three-level (trials nested within sessions nested within participants) multilevel models[27] (MLM) at four points during our analysis of the data. In all cases, random intercepts were estimated. The significance of models was estimated with chi-square tests, where models with increasing complexity were compared and the resulting value of likelihood ratio test ($\chi^2$) and corresponding $p$-value were reported[61]. The MLM analysis was performed using nlme[62] R package.

Behavioral changes during training. To investigate behavioral changes in behavioral performance depending on the session, task condition, and group, we used a three-level multilevel model with $d'$ as the dependent variable and with group (two factors: experimental and control), condition (two factors: 1-back and 2-back, reference category: 1-back), and session (four factors: Naive, Early, Middle, Late; reference category: Naive) as independent variables. In addition to the main effects (group, condition, session), we included the following interaction terms: group × session, condition × session, group × condition, and group × condition × session.

Modularity at baseline. To investigate the dependence of static modularity at baseline on task condition, we used a two-level multilevel model with static modularity as the dependent variable and with task condition (3 factors: rest, 1-back, 2-back, two orthogonal contrasts: rest vs. 1-back and 2-back, 1-back vs. 2-back) as the independent variable. The main effect of condition was tested.

Training-dependent changes in static modularity. To investigate the dependence of static modularity on the session, task condition, and group, we used a three-level multilevel model with static modularity as the dependent variable and with group (two factors: experimental and control), condition (two factors: 1-back and 2-back), and session (four factors: Naive, Early, Middle, Late, reference category: Naive) as independent variables. In addition to the main effects (group, condition, session), we included the following interaction terms: group × session, condition × session, group × condition, and group × condition × session.

Changes in dynamic network metrics. To investigate changes in the integration and recruitment of large-scale systems, we used a two-level multilevel model with the diagnostic measure (recruitment or integration) as the dependent variable and with group (two factors: experimental and control) and session (four factors: Naive, Early, Middle, Late, reference category: Naive) as independent variables. In addition to the main effects (group, session), we included the following interaction term: group × session.

**Reporting summary**. Further information on research design is available in the Nature Research Reporting Summary

## Data availability

The raw behavioral data and fMRI results are available for download at https://osf.io/wf85u/ (https://doi.org/10.17605/OSF.IO/WF85U). The source data underlying Figs. 2–7, and Supplementary Figs. 1–21 are provided as a Source Data file. The raw fMRI data are available from the corresponding author on request.

## Code availability

All code used for neuroimaging and behavioral data processing and statistical data analyses are publicly available at https://osf.io/wf85u/ (https://doi.org/10.17605/OSF.IO/WF85U).

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

## Acknowledgements

The study was supported by the National Science Centre, Poland (2015/17/N/HS6/03549). K.F. was supported by National Science Centre, Poland (2015/17/N/HS6/03549, 2017/24/T/HS6/00105) and Foundation for Polish Science, Poland (START 23.2018). Calculations have been carried out using resources provided by Wroclaw Centre for Networking and Supercomputing (http://wcss.pl), grant No. 467. D.S.B. also acknowledges support from the John D. and Catherine T. MacArthur Foundation, the Alfred P. Sloan Foundation, the ISI Foundation, the Paul Allen Foundation, the Army Research Laboratory (W911NF-10-2-0022), the Army Research Office (Bassett-W911NF-14-1-0679, Grafton-W911NF-16-1-0474, DCIST- W911NF-17-2-0181), the Office of Naval Research, the National Institute of Mental Health (2-R01-DC-009209-11, R01-MH112847, R01-MH107235, R21-M MH-106799), the National Institute of Child Health and Human Development (1R01HD086888-01), the National Institute of Neurological Disorders and Stroke (R01 NS099348), and the National Science Foundation (BCS-1441502, BCS-1430087, NSF PHY-1554488 and BCS-1631550). D.L.S. acknowledges support from The National Institute on Drug Abuse (K01DA047417). We also thank Maja Dobija, Alex Lubiński, Stanisław Narębski, Monika Muchlado, Bożena Pięta, and Adrianna Przybysz for their assistance in conducting training sessions, and Jaromir Patyk for technical support. The content is solely the responsibility of the authors and does not necessarily represent the official views of any of the funding agencies.

## Author contributions

K.F. provided funding for the study. K.F. designed the study with the contribution from S.K. K.F. and K.B. collected and processed the data. K.F. performed network and statistical analyses with the support and contributions from D.S.B., X.H., D.M.L., and K.B. K.F. wrote the paper. D.S.B., W.D., K.B., X.H., D.M.L, and S.K. revised the paper.

## Competing interests

The authors declare no competing interests.
