## [Peer Review File · Nature Communications]

Reviewers' Comments:

Reviewer #1:

Remarks to the Author:

Thank you for inviting me to review this manuscript by Finc and colleagues, in which the authors conduct a topological analysis of multi-session fMRI data collected on a sample of subjects that performed a challenging N-back task over the course of a 6-week training schedule (Experimental group) or as part of a Control group. The authors report a replication of previous work, which demonstrated a decrease in modularity (and hence, increase in integration) as a function of cognitive challenge. They then go on to show that, over the course of learning, that this decrease in modularity is less extensive in the Experimental cohort, which they take as evidence of less integration as a function of learning. In subsequent analyses, they focus on time-resolved modularity within and between the default mode and frontoparietal networks, and demonstrate distinct patterns of frontoparietal network recruitment between the Experimental and Control groups over the course of the task.

The results of this study are interesting and help to extend our understanding of whole-brain network topology both during complex cognitive tasks but also as a function of training/automaticity. My only real qualm with the approach was that there was less focus placed on characterizing the topology of the broader network -- I understand the logic behind focussing on the frontoparietal and default networks, however the N-back task also requires selective engagement of the dorsal attention, visual and motor networks. The authors may wish to consider tracking these sub-networks as a function of load/learning too, as it would open up a number of interesting questions. For instance, do the primary sensorimotor regions required to perform the task stay relatively similar from Naive  Late conditions? If their activity stays the same, does their integration (perhaps selectively) with parts of the frontoparietal and default networks change substantially over the course of learning?

Figure 4 may be augmented by including a set of graphs depicting the difference between the two cohorts, perhaps as a subpanel of 4a. As it stands, It is difficult to tell by eye whether the topological patterns were matched or distinct across groups.

On Figure 4, are the graphs in 4b meant to depict default regions on the left and frontoparietal on the right? If so, this should be made more explicit in the figure legend. Also, the choice of using a yellow-to-red color bar in 4b clashes perceptually with the choice of labeling the default and frontoparietal groups using the same colors as category labels in 4a. This is unnecessarily confusing.

Although the authors did include them (Fig 7), I note that not much was made of the role of subcortical regions in the analysis. Admittedly, the resolution is less than ideal, however I imagine that subcortical regions (in particular, basal ganglia and cerebellum) would be very important for learning (and hence, automatizing) the processes required to perform the N-back task effortlessly over the course of training. Importantly, the structure of these two systems is far more variable than many cortical RSNs, and hence, the lack of a group-level significant p-value (Fig 7b) is not particularly meaningful in my opinion.

On the topic of 7b, the authors may wish to reconsider the utility of comparing two separate 1-sided statistical tests next to one another. The true test of whether the groups differed should come from a paired-comparison between the two groups.

The language in the abstract was a tad misleading: "We found that whole-brain modularity was higher during the resting state than during the dual n-back task, and increased as demands heightened from the 1-back to

the 2-back condition." This made it sound as though modularity was maximal at rest, decreased during 1-back and then increased during 2-back, which is not what the authors report. They may wish to rephrase this to avoid confusion.

Towards the latter half of the Results section, it began to feel as though there were a deluge of new analyses that weren't clearly justified in the Introduction. The authors may wish to update their manuscript such that the reader has a clear appreciation for which analyses will be reported and what the implications of the analyses might be.

Reviewer #2:

Remarks to the Author:

In this manuscript, Finc et al examine changes in network states over the course of training in a demanding cognitive task (adaptive dual n-back). Building on prior work that revealed change in network modularity with training of a simpler motoric task, they are now testing whether similar changes in network segregation and integration occur with the training of a more complex task.

Overall, the manuscript is quite clearly written and thoughtfully analyzed. At the same time, from a neuroscientific point of view, I am not entirely clear on the take-home message: the network analysis operates at a level of abstraction which can reveal new phenomena, but which also could easily be masking other (simpler) processes that would be more apparent from simpler and more traditional analyses.

~~ There are no task-related changes in signal amplitude or variance reported in this manuscript. This is a problem for two reasons. First, it makes it difficult to relate the present work to prior literature which has studied how (e.g. motor, premotor, prefrontal and basal ganglia) regions change their activity over the course of training. Secondly, the absence of these basic signal measurements leaves it uncertain whether any of the network metrics could be biased by basic changes in the neural response during the task. It is incumbent on the authors to demonstrate that the network metrics (which are more abstract, and based on covariance properties of the signal) are not being affected by changes in the mean and variance of the signal, within and across blocks and sessions. One possible solution to this problem would be via a well-calibrated simulation, which would show that the network metrics are immune to such change (...I believe such a simulation would actually show the opposite). However, a more direct solution would be to estimate and report the changing activation patterns in the data (something closer to a traditional GLM analysis), and then demonstrate that the network results persists even when these activation changes are removed. Finally, the authors could argue that they explicitly refuse to include the analysis of the changes in the mean activation patterns, because they believe that these signals obscure the more fundamental neural processes, which are revealed by the network perspective. In that case, they would need to make that argument compellingly. Overall, I think the simplest thing to do is to report descriptive statistics on the local activation patterns, and how they change over blocks and over training.

~~ Data from a "2-back" condition for the Control group are reported in Figure 2B, and yet neither the manuscript introduction nor methods make any mention of the existence of this condition. Instead, the Methods explicitly state:

"Participants from the control group performed a single 1-back with auditory or visuospatial stimuli variants."

~~ On page 5, there is the following claim:

"Collectively, these results demonstrate that modularity increased for both 1-back and 2-back task conditions in the experimental group but only during the 1-back condition for the control group. The findings suggest that higher brain network segregation during the 2-back condition may be a consequence of the 6-week working memory training."

However, the analyses provided do not support these claims. The analysis shows that some conditions exceed a statistical threshold, while others do not exceed that threshold; this does not provide a statistical test of the difference (or the interaction) across conditions. Please provide a direct test if you want to make this claim.

~~ Does behavior vary cross blocks, and does the behavioral variability decrease with training? Given that one of the main results concerns increases in network recruitment (which measures something like the consistency of module assignment), it seems important to test whether the stability of the networks is related to greater stability (e.g. via more sustained attention) in the cognitive and behavioral performance, as a function of training.

~~ On page 8: The DMN and FPN regions were not the only ones exhibiting increased recruitment with training — as the authors note on page 8, the ventral attention, salience, cingulo-opercular, and auditory systems also show the effect. So why are the DMN and FPN systems emphasized in the Abstract and the Discussion? Have the authors conducted further analyses on these other networks, and did they find fewer links to behavior? Less change in integration over time? It is not clear from the manuscript how we are supposed to understand or interpret the changes in these additional systems, and how they differ fundamentally from the changes reported for the FPN and DMN.

Minor Things:

~~ Abstract:

"behavioral automation" and "behavioral adaptation" seem to be used to refer to the same process, but presumably they mean different things?

~~ The "recruitment" terminology seems a little awkward. On p.5 we read that "Intuitively, high recruitment indicates that nodes of the system are consistently assigned to the same module across different layers." I am struggling to understand why this concept should be summarized as "recruitment", rather than "consistency", "reliability", "community stability", or something else?

~~ The manuscript would be stronger if slightly more detail was provided concerning the n-back tasks (especially related to the specific audiovisual signals that are detected by participants, and the blocking structure) in the Introductory section.

~~ Please make clear whether the consecutive blocks of the n-back tasks are identical, or whether there is any interleaving or other systematic parameter variation across blocks. This is critical for

interpreting the dynamic network analyses, which depend on changes in neural response across blocks.

~~ Please provide the reader with a summary statement of the many analyses included in the section "Dynamic reorganization of default mode and fronto-parietal systems" on pp.5-7; there are many analyses included in the latter part of this section, but their interpretation seems muddy to me. It would help to have some more interpretation or commentary interleaved with the results on page 7 in particular.

~~ "comprised of" should be "composed of" in all instances in the manuscript

Reviewer #3:

Remarks to the Author:

The authors investigate the dynamic reconfiguration of functional brain networks over 6 weeks of training via a static modularity analysis and a multilayer modularity analysis. In the four fMRI scans within the 6 weeks, they assess the brain network modularity, as well as the recruitment and integration of the default mode and frontoparietal systems from a static and a dynamic perspective. The overall results show that whole-brain modularity differed between rest and task conditions, being lowest in the condition with more demands. They also show that through training, the brain modularity and the recruitment of frontoparietal and default mode systems will increase, indicating an increase of segregation of brain network. The study is well conceptualized. There is a relatively clear and reasonable hypothesis of what the authors expected, methods are state of the art and the results are very interesting. However, my major concerns relate to this work from both methodological and clinical perspectives, as detailed below.

1. The author said that "participants from the control group performed a single, non-adaptive, 1-back working memory task". But from the results section, control groups also have graph measures calculated from the 2-back condition, which is a bit confusing.
2. In the statistical analysis, do the authors perform any outlier detection? It seems like some potential outliers might bias the observations (especially Figure 3 b and f, Figure 5).
3. Please provide the statistic results comparing the pRT between 1-back and 2-back conditions after training for the controls groups.
4. Before calculating the whole-brain modularity, the authors estimate correlations between ROI time-courses. However, they only retain the positive correlations for further analysis which do not make sense because the antagonism is typically connected patterns in functional brain networks, especially when one of their major interests is the integration between antagonistic frontoparietal and default mode systems. It is better to include the negative correlations in the analysis. For example, use alternative modularity detection algorithm which can work on negative edges or use the absolute value of the correlation matrix.
5. In Figure 3 b, it is said that "The greater the decrease in modularity from 1-back to 2-back, the smaller the decline in performance, as measured by pRT, from 1- back to 2-back". It seems like the difference is calculated by "1-back minus 2-back" because the difference of pRT is all negative from the x-axis. However, there are more negative values along the y-axis, indicating that 1-back condition has low modularity than 2-back condition, which is not the case showing Figure 3 a. Please clarify this.
6. There is a clear group difference between experimental and controls groups on the modularity in the naive session, especially for the 1-back task. Can you provide a statistical comparison between them? Such difference exists even when there is no difference between the pRT, which can be potential confounding effects of the analysis and needs to be clarified.
7. In figure 3 e and f, although the experimental group shows more difference between naive and late

sessions, such group difference is not significantly larger than the difference for the control group, resulting in less solid conclusion.

8. There is no correlation between the decrease modularity and the increase pRT. The authors argue that is because the change of modularity is a general consequence of training. However, the repeated exposure also results in decrease modularity (1-back condition). If the modularity decrease caused by training is not related to the improved performance, how can authors conclude that the working memory training may help to prevent cognitive decline?

9. Can the authors provide the statistic results by comparing recruitment between experimental group and control group in the naive session? If the difference is significant, please discuss this in the discussion.

10. In figure 5, is there any multiple comparison correction performed? Also, such associations are significant in both groups, which might suggest that DM recruitment changes resulted from the repeated exposure will be the major cause of improved performance.

11. In summary, the overall results cannot fully support that the reconfiguration of the brain network due to the training can improve the working memory performance. The authors need more analysis and discussion to clarify this.

Comments from Reviewer 1

Comment 1: *Thank you for inviting me to review this manuscript by Finc and colleagues, in which the authors conduct a topological analysis of multi-session fMRI data collected on a sample of subjects that performed a challenging N-back task over the course of a 6-week training schedule (Experimental group) or as part of a Control group. The authors report a replication of previous work, which demonstrated a decrease in modularity (and hence, increase in integration) as a function of cognitive challenge. They then go on to show that, over the course of learning, that this decrease in modularity is less extensive in the Experimental cohort, which they take as evidence of less integration as a function of learning. In subsequent analyses, they focus on time-resolved modularity within and between the default mode and frontoparietal networks, and demonstrate distinct patterns of frontoparietal network recruitment between the Experimental and Control groups over the course of the task.*

The results of this study are interesting and help to extend our understanding of whole-brain network topology both during complex cognitive tasks but also as a function of training/automaticity. My only real qualm with the approach was that there was less focus placed on characterizing the topology of the broader network -- I understand the logic behind focussing on the frontoparietal and default networks, however the N-back task also requires selective engagement of the dorsal attention, visual and motor networks. The authors may wish to consider tracking these sub-networks as a function of load/learning too, as it would open up a number of interesting questions. For instance, do the primary sensorimotor regions required to perform the task stay relatively similar from Naive  Late conditions? If their activity stays the same, does their integration (perhaps selectively) with parts of the frontoparietal and default networks change substantially over the course of learning?

Response: We thank the reviewer for this suggestion. When designing the study we hypothesized that training-related changes will be visible in the default mode (DM) and fronto-parietal (FP) systems. Accordingly, we focused our presentation on those two systems, and initially only included a rather brief summary of effects discovered for different large-scale networks. The latter were displayed in Figure 6 in the original version of the manuscript. In the revised version, we follow the reviewer's suggestion and place greater emphasis on the broader network topology. Specifically, we performed analyses on recruitment and integration coefficients for all large-scale systems. To enable across session comparison, as in the case of the DM and FP analysis, we applied multilevel modelling for repeated measures data (Snijders *et al.*, 2012). This enabled us to test for the main effects of session and session \times group interactions for each recruitment (or integration) coefficient. In contrast to the previous version of the manuscript, we also applied a normalization to allegiance matrices, to remove any potential bias introduced by differences in the number of nodes within each subsystem.

Method section:

To remove any potential bias introduced by the differences in the number of nodes within each system, we used a permutation approach to normalize the values of the recruitment and integration coefficients. For each subject and session, we created $N_{perm} = 1000$ null module

allegiance matrices by randomly permuting the correspondence between ROIs and large-scale systems. We then calculated the functional cartography measures for all permuted matrices. This procedure yielded null distributions of recruitment and integration coefficients resulting solely from the size of each system. In order to obtain normalized values of R_s and $I_{Sk,St}$ we divided them by the mean of the corresponding null distribution.

Code for the network normalization can be found here:

[https://github.com/kfinc/wm-training-modularity/blob/master/04-dynamic FC analyses/04-whole-brain normalized recr integ.ipynb](https://github.com/kfinc/wm-training-modularity/blob/master/04-dynamic%20FC%20analyses/04-whole-brain%20normalized%20recr%20integ.ipynb)

To incorporate these findings into the revised version of the manuscript, we reorganized our *Results* section by changing the *Dynamic reorganization of default mode and fronto-parietal systems* subsection into a new *Dynamic reorganization of large-scale systems* subsection. To remain true to our hypotheses, we describe the dynamic changes in the DM and FP first and in greater detail, and we describe the changes in other systems second and in less detail. Note that the statistics that we now present are slightly different than those presented in the previous version of the manuscript, as we normalized allegiance matrices to account for the system size.

Results section:

Dynamic reorganization of large-scale systems ~~default mode and fronto-parietal systems~~

The modular architecture of functional brain networks is not static but instead can fluctuate appreciably over task blocks. Here, we used a dynamic network approach to answer the question of whether *large-scale brain systems relevant to working memory — the fronto-parietal and the default mode* change in their fluctuating patterns of expression during training. Based on a previous study of motor sequence learning (Bassett et al., 2015), we expected that *systems relevant to working memory -- the fronto-parietal and the default mode -- these two systems* would become more autonomous over the 6 weeks of working memory training. (...)

The presence of fluctuations in community structure across task blocks is indicated by variable assignments of nodes to modules across layers. For each subject and session, we summarized these data in a module allegiance matrix \mathbf{P} , where each element \mathbf{P}_{ij} represents a proportion of blocks for which node i and node j were assigned to the same module. We also applied a normalization to allegiance matrices, to remove any potential bias introduced by differences in the number of nodes within each subsystem. Following the functional cartography framework described by Mattar et al. (2015), we used \mathbf{P} to calculate the recruitment of all 13 large-scale systems, as well as the pairwise integration among them ~~of the default mode and fronto-parietal systems~~ (see Methods for details). (...)

First, we examine dynamic topological changes in the fronto-parietal and default mode systems, which were directly related to our hypothesis. Using a multilevel model, we observed a significant session \times group interaction effect when considering changes in the recruitment of

the fronto-parietal system during training ($\chi^2(3) = 9.03, p = 0.028$) (Figure 5b). The largest increase in fronto-parietal recruitment was observed in the experimental group when comparing 'Early' to 'Late' training phases ($\beta = -0.07, t(120) = -2.892, p = 0.027$, Bonferroni-corrected; Figure 5c). No significant changes from 'Naive' to 'Late' training phases were observed in the control group ($\beta = -0.03, t(120) = -1.169, p = 1$, Bonferroni-corrected). Turning to an examination of the default mode, we found a significant main effect of session ($\chi^2(3) = 24.17, p < 0.0001$) and of group ($\chi^2(1) = 3.96, p = 0.046$) on system recruitment (Figure 5c). However, the interaction effect between session and group was not significant ($\chi^2(3) = 2.66, p = 0.48$). Planned contrasts revealed that the default mode recruitment increased steadily in both groups and we observed the largest increase between 'Naive' and 'Late' sessions ($\beta = 0.09, t(123) = 5.00, p < 0.0001$). The experimental group showed a higher default mode recruitment than the control group ($t(165.6) = -3.03, p = 0.003$). We found a significant session \times group interaction effect on the integration between the fronto-parietal and default mode systems ($\chi^2(3) = 14.25, p = 0.0025$) (Fig 5d). The integration between these two systems decreased from 'Naive' to 'Late' sessions only in the experimental group ($\beta = 0.07, t(120) = 4.37, p = 0.0002$, Bonferroni-corrected). However, groups differed from 'Naive' to 'Early' ($\beta = 0.07, t(120) = 2.16, p = 0.03$) and from 'Early' to 'Middle' sessions ($\beta = -0.06, t(120) = -2.70, p = 0.02$, Bonferroni-corrected): whereas the experimental group displayed an inverted U-shaped curve of integration with training, the control group displayed the opposite pattern. Collectively, these results suggest that the increase of fronto-parietal system recruitment and the decrease of integration between the default mode and fronto-parietal systems reflect training-specific changes in dual n-back task automation. In contrast, the increase in default mode system recruitment may reflect more general effects of behavioral improvement, as it was observed in both experimental and control groups. (...)

Alongside the findings regarding the DM and FP systems, we discovered intriguing patterns of widespread network reorganization. Specifically, we identified three distinct types of changes occurring over time regardless of the group: (1) increased recruitment of multiple systems (including DM and salience), (2) decreased integration between DM and task-positive systems, and (3) increased integration between task-positive systems (see Figure 6a-c and Supplementary Figure 7). This pattern of findings suggests that the DM system gradually increases its autonomy, while the task-positive systems become more integrated over time. Interestingly, the performance change from 'Naive' to 'Late' session was positively correlated with change of the DM and salience systems recruitment, whereas change of integration between DM and task-positive systems (fronto-parietal and salience) was negatively correlated with the change of performance (see Figure 7a).

We also found several session \times group interaction effects beyond DM-FP integration and FP recruitment, mainly for subcortical, dorsal-attention, and cingulo-opercular systems (Supplementary Figure 8). There was also an interesting interaction effect directly related to the reviewer's comment: we found that integration between the somatosensory (SOM) and dorsal attention (DA) systems changed differently for the control and experimental groups over the course of training. Specifically, the experimental group exhibited an increase of DA-SOM integration from 'Early' to 'Middle' sessions ($\beta = 0.0790, p = 0.0130$; Bonferroni-corrected), followed by a decrease from 'Middle' to 'Late' session; the control group displayed the opposite direction of changes in these time intervals ($\beta = 0.0790, p =$

0.0130 and $\beta = -0.0711$, $p = 0.0300$ respectively; Bonferroni-corrected; see Supplementary Figure 8h, Supplementary Table 5).

We also observed that the experimental and control group differed in dynamic network changes in particular between ‘Naive’ and ‘Early’ sessions. Similarly, we observed the largest behavioral improvement in the experimental group within this time interval. Accordingly, we tested whether this behavioral change was related to specific changes in the network organization during the early phase of training. We observed that better behavioral outcomes in response to intense working memory training were associated with integration between multiple systems, including system pairs for which we also observed group \times session interactions: subcortical and dorsal attention, somatomotor and dorsal attention, and subcortical and cingulo-opercular systems (Figure 7b and Table 7). We observed consistent results for the Δ pRT measure. Note that correlations had opposite signs as a lower Δ pRT denotes a better behavioral performance (Supplementary Figure 9 and Supplementary Table 7).

Edited *Results* section:

Next, we asked whether changes in dynamic topology could be observed in other large-scale systems. Using multilevel modeling, we observed three distinct types of changes occurring over time regardless of the group ($p < 0.05$, FDR-corrected; Figure 6a-c): (1) an increase in system recruitment, (2) an increase in the integration between task-positive systems, and (3) a decrease in the integration between default mode and task-positive systems (Supplementary Figure 7, Supplementary Table 1, Supplementary Table 2). First, we observed an increase in the recruitment beyond the default mode system -- in salience, and auditory systems (Supplementary Figure 7a-c). Second, we observed an increase in the integration between task-positive systems, including fronto-parietal and salience, dorsal attention and salience, and dorsal attention and cingulo-opercular (Supplementary Figure 7d-f). Third, for the default mode system, we observed a decrease in integration with other task-positive systems: salience and cingulo-opercular (Supplementary Figure 7g-i). Additionally, we also observed a decrease in integration between the memory and somatomotor systems, and between the default mode and auditory systems (Supplementary Figure 7j-k). We observed a similar pattern of changes for the Schaefer parcellation (Supplementary Figure 17, Supplementary Figure 18a). These results suggest that the increase of within-module stability, the increase of default mode system independence from task-positive systems, and the decrease of integration between working memory systems reflect general effects of task training.

We also investigated the relationship between across-session change in system recruitment or integration and across-session change in behavioral performance for all large-scale systems. For both brain and behavioral variables, we measured the change from the first (‘Naive’) to the last (‘Late’) training sessions (see Figure 7a, Supplementary Table 6). We found a significant positive correlation between change in behavior, as operationalized by a change in d' (2-back minus 1-back), and change of the default mode ($r = 0.33$, $p = 0.03$, uncorrected) and salience ($r = 0.34$, $p = 0.03$; uncorrected) systems recruitment. Greater behavioral improvement was also associated with a bigger increase of integration between fronto-parietal and salience systems (0.35 , $p = 0.02$, uncorrected) and a larger decrease of integration between default mode and task-positive systems: fronto-parietal ($r = -0.31$, $p =$

0.04, uncorrected) and salience ($r = -0.41$, $p = 0.006$, uncorrected). Analogous relationships for default mode recruitment and default mode - fronto-parietal integration with behavioral improvement were observed for an alternative measure of performance (pRT; Supplementary Figure 9a). Note that the correlation for the change in the d' measure has opposite sign when compared to the correlation with the change of pRT, consistent with the fact that these two measures have different interpretations (the lower pRT, the better; the larger the d' , the better). In summary, a larger increase of stability in the default mode and salience systems, together with a decrease of default mode - task-positive systems integration may support behavioral improvement in the task, regardless of whether the task was additionally trained or not.

Figure 6 | Changes of the recruitment and integration of large-scale systems. Colored tiles represent all significant effects ($p < 0.05$, uncorrected; $*p < 0.05$ FDR-corrected). (top panel) Here we display the significant main effects of session. Tile color codes a linear regression coefficient (β), for all main session effects: (a) from 'Naive' to 'Early', (b) from 'Naive' to 'Middle', and (c) from 'Naive' to 'Late'. (bottom panel) Here we display the significant session \times group interaction effects. Tile color codes a linear regression coefficient between groups and sessions: (c) from 'Naive' to 'Early', (d) from 'Naive' to 'Middle', and (e) from 'Naive' to 'Late'. Abbreviations: auditory (AU), cerebellum (CER), cingulo-opercular (CO), default mode (DM), dorsal attention (DA), fronto-parietal (FP), memory (MEM), salience (SAL), somatomotor (SOM), subcortical (SUB), uncertain (UNC), ventral attention (VA), and visual (VIS). Source data are provided as a Source Data file.

Supplementary Figure 7 | Session-to-session changes in recruitment and integration of large-scale systems. We observed three main categories of large-scale system reorganization: (a-c) an increase in system recruitment, (d-f) an increase in integration between task-positive systems (TP), (g-i) a decrease in integration between the default mode (DM) system and task-positive systems, and (j-k) other. Error bars represent 95% confidence intervals. Remaining abbreviations: salience (SAL), auditory (AU), fronto-parietal (FP), dorsal attention (DA), cingulo-opercular (CO), memory (MEM), and somatomotor (SOM). Source data are provided as a Source Data file.

Figure 7 | Relationship between dynamic network changes and behavior. Colored tiles represent all significant correlations ($p < 0.05$, uncorrected; $*p < 0.05$ FDR-corrected). **(a)** Pearson correlation coefficient (r) between the across-session changes in recruitment (or integration) and the across-session changes in d' ($\Delta d'$) observed for both experimental and control group. **(b)** Relationship between the changes in recruitment (or integration) and the changes in d' during early phase of training of the experimental group. Abbreviations: auditory (AU), cerebellum (CER), cingulo-opercular (CO), default mode (DM), dorsal attention (DA), fronto-parietal (FP), memory (MEM), salience (SAL), somatomotor (SOM), subcortical (SUB), uncertain (UNC), ventral attention (VA), and visual (VIS). Source data are provided as a Source Data file.

Supplementary Figure 9 | Relationship between dynamic network changes and behavior. Colored tiles represent all correlations ($p < 0.05$, uncorrected; $*p < 0.05$ FDR-corrected). **(a)** Pearson correlation coefficient (r) between the across-session changes in recruitment (or integration) and the across-session changes in penalized reaction time (ΔpRT) observed for both experimental and control group. **(b)** Relationship between the changes in recruitment (or integration) and the changes in pRT during early phase of training of the experimental group. (...)

Finally, we also observed session \times group interaction effects beyond the default mode and fronto-parietal systems ($p < 0.05$, uncorrected, Figure 6d-e, Supplementary Table 3-4). Specifically, in the experimental group, we observed a non-linear change in the integration of the subcortical system with the dorsal attention, ventral attention, cingulo-opercular, and auditory systems. An initial increase in integration with the subcortical system (from 'Naive' to 'Early') was followed by a decrease in the integration at later time intervals. Interestingly, we observed the reverse pattern for the change in integration between the subcortical and default mode systems: the integration first decreased from 'Naive' to 'Early' sessions, and then increased from 'Early' to 'Middle' sessions for the experimental group (Supplementary Figure 8). The pattern of changes in integration also differed between the groups, particularly so for the integration between cingulo-opercular and memory systems, cingulo-opercular and uncertain systems, and dorsal attention and somatomotor systems. These results suggest that task automation during initial stages of working memory training might also be supported by an increased communication between subcortical and other large-scale systems.

We further tested whether changes in systems recruitment or integration from 'Naive' to 'Early' sessions were associated with performance improvement displayed by the experimental group. Interestingly, we found that behavioral change was positively correlated with change of integration between multiple systems, in particular: dorsal attention and somatomotor, dorsal attention and subcortical, fronto-parietal and somatomotor, dorsal attention and cingulo-opercular, salience and default mode. In contrast, increase of integration of subcortical system and cingulo-opercular systems was negatively correlated with the change in task performance (Figure 7b, Supplementary Table 7). This pattern of associations between behavioral and network changes suggest that inter-systems communication might be necessary for efficient task performance during initial stages of training.

In summary, we observed two patterns of dynamic changes in network topology following working memory training. The first pattern reflects improved behavioral performance and is characterized by a gradual increase in default mode autonomy and in the integration between task-positive systems. The second pattern reflects changes related to task automation specifically in the experimental group and is characterized by non-linear changes in default mode - fronto-parietal integration, and in the integration with the subcortical system.

Supplementary Figure 8 | Session \times group interaction effects for across-session changes in recruitment and integration of large-scale systems. We observed group differences in the changes in (a-e) integration of the subcortical (SUB) system with other systems, (f-h) integration of the task-positive (TP) systems with other systems, (i) integration of the default mode (DM) system with the fronto-parietal (FP) system, (j-k) changes in the dorsal attention (DA) and FP recruitment. Error bars represent 95% confidence intervals. Remaining abbreviations: salience (SAL), auditory (AU), memory (MEM), and somatomotor (SOM). Source data are provided as a Source Data file.

Code for multilevel modeling and figures can be found here:

- [https://github.com/kfinc/wm-training-modularity/blob/master/04-dynamic FC analysis/05-whole-brain normalized recr integ stats.ipynb](https://github.com/kfinc/wm-training-modularity/blob/master/04-dynamic%20FC%20analysis/05-whole-brain%20normalized%20recr%20integ%20stats.ipynb)
- [https://github.com/kfinc/wm-training-modularity/blob/master/04-dynamic FC analysis/06-whole-brain mlm networks summary.ipynb](https://github.com/kfinc/wm-training-modularity/blob/master/04-dynamic%20FC%20analysis/06-whole-brain%20mlm%20networks%20summary.ipynb)

Finally, we also revised the corresponding portions of the Discussion.

Using dynamic network metrics, we showed that the default mode system increased its recruitment in both groups, indicating that regions within this system were coupled with other communities less often. Moreover, the experimental group displayed an increased fronto-parietal recruitment and an inverted U-shaped curve of integration between the default mode and fronto-parietal systems with training. Enhanced default mode intra-communication and decreased inter-communication with the fronto-parietal system were associated with better behavioral outcomes after training. We also observed significant changes in dynamic network topology beyond the fronto-parietal and default mode systems. In particular, regardless of the group, we observed an increased recruitment of the salience and auditory systems, decreased integration between the default mode and other task-positive systems (including salience and cingulo-opercular), and increased integration between task-positive systems (including fronto-parietal and salience, dorsal attention and salience, dorsal attention and cingulo-opercular). These results suggest the existence of the trade-off between segregation and integration: whereas segregation increases between some systems, the integration increases or decreases between others. (...)

Figure 8 } Schematic diagram summarizing the main changes in recruitment and integration of large-scale systems observed over the course of working memory training. We observed a gradual increase of the integration between task-positive systems (fronto-parietal - FP, salience - SAL, dorsal attention - DA, and cingulo-opercular - CO), greater recruitment of the default mode (DM) system, and decreased DM-CO and DM-SAL integration. In the early phase of training (a) the experimental group displayed an increased FP-DM integration, and increased integration of the subcortical (SUB) system with the DA and CO systems. In the late stage of training (b), the FP system reduced its integration with the DM system and the SUB system increased its integration with the DM system, while decreasing coupling with task-positive systems.

Our results are also consistent with prior observations that the default mode and fronto-parietal systems may interact in a task-dependent manner with the salience, cingulo-opercular, and dorsal attention systems (Bressler et al 2010; Cocchi et al., 2013). Bressler and Menon (2010) proposed a model whereby efficient cognitive control is supported by the dynamic switching between functionally segregated fronto-parietal and default mode systems mediated by cingulo-opercular and salience systems. Cocchi et al. (2013) proposed

that task-related reconfiguration is possible through flexible interactions within and between overlapping meta-systems: (i) the executive meta-systems, responsible for the processing of sensory information, and (ii) the integrative meta-system, responsible for flexible integration of brain systems. These two meta-systems are composed of transient coupling between three large-scale systems: the frontoparietal system, the cingulo-opercular/salience system, and the default mode system. During high-demand task conditions the executive meta-system is formed by extensive interactions between fronto-parietal and cingulo-opercular/salience systems, and the default mode system is more segregated and less integrated with the fronto-parietal system (Leech et al., 2012). Our results extend these findings by presenting the evolving reconfigurations of large-scale networks during mastery of the working memory task. We showed that regardless of the group the default mode system reduced coupling with the cingulo-opercular and salience systems. These results suggest that increased segregation of the default mode and task-positive networks may be a consequence of more efficient task performance. A similar pattern of changes was observed across two different subdivisions of the cortex into systems (Power and Schaefer), together suggesting that the salience and cingulo-opercular systems that are thought to be responsible for switching between antagonistic fronto-parietal and default mode systems, appear to be more integrated with the fronto-parietal system and less integrated with the default mode system. This pattern of relations may be due to diminished requirements for switching between these two systems when the task is well trained.

(...)

The fronto-parietal system dynamically interacts with other large-scale systems (Cocci et al., 2013), and it is reasonable to expect that working memory training might influence interactions in the whole network. We observed training-related increases in the segregation of the default mode and task-positive systems that suggest more efficient and less costly processing within these systems after training. Accordingly, greater segregation of the default mode system and task-positive systems and smaller integration between these systems were associated with behavioral performance improvement. Moreover, we showed that an increase of integration between multiple large-scale systems in early phase of training was related to a greater behavioral improvement in the experimental group, indicating that some level of network integration is necessary when the task is not fully automated. Taken together, the dynamic network approach provides a unique insight into the plasticity and dynamics of the human brain network.

Comment 2: Figure 4 may be augmented by including a set of graphs depicting the difference between the two cohorts, perhaps as a subpanel of 4a. As it stands, It is difficult to tell by eye whether the topological patterns were matched or distinct across groups.

Response: We thank the reviewer for this suggestion. We created an additional figure representing the difference in topological patterns between the experimental and control groups. We decided to present this figure in the *Supplementary Information* (Supplementary Figure 6), as in the main manuscript we were interested specifically in session-to-session changes in topological patterns, not group differences at each session separately. We believe that line plots at the bottom of the Figure 4 (Figure 5 in the revised manuscript) work best in presenting group differences. Also, because we expanded our analysis to all large-scale systems, we now include an additional figure (Figure 6 in this response document), that

shows the main effects of session and session \times group interaction effects for whole-brain topological patterns. We also now include Supplementary Figure 7 and Supplementary Figure 8, which provide the reader with detailed information regarding changes in the topological network patterns for both groups separately.

Supplementary Information:

Supplementary Figure 6 | Between-group differences in module allegiance matrices for the default mode and fronto-parietal systems. Each ij -th element of the matrix represents a difference between groups (experimental minus control) in the probability that node i and node j are assigned to the same module within a single layer of the multilayer network. Systems are defined using the Power et al. (2011) parcellation. Source data are provided as a Source Data file.

Comment 3: On Figure 4, are the graphs in 4b meant to depict default regions on the left and frontoparietal on the right? If so, this should be made more explicit in the figure legend. Also, the choice of using a yellow-to-red color bar in 4b clashes perceptually with the choice of labeling the default and frontoparietal groups using the same colors as category labels in 4a. This is unnecessarily confusing.

Response: We thank the reviewer for raising these points. In the previous version of the manuscript, our brain visualization on the left of subpanel 4b depicted the default mode system, whereas the brain visualization on the right depicted regions of the fronto-parietal system. We used red and yellow colors to maintain consistency with the colormap for the brain parcellation provided by Power et al. (2011). Because the previous brain visualization was not only misleading but also not very informative, we decided to replace it with a visualization of ROIs colored by network assignment. We provide the new version of the figure (Figure 5 in the revised manuscript) and associated caption below.

Figure 5 | Changes in module allegiance of the fronto-parietal (FP) and default-mode (DM) systems. (a) Module allegiance matrices for the default mode and fronto-parietal systems. Each ij -th element of the matrix represents the probability that node i and node j are assigned to the same module within a single layer of the multilayer network. (b) Only the experimental group exhibited increases in fronto-parietal recruitment across sessions. (c) Both experimental and control groups exhibited increases in default mode recruitment between 'Naive' and 'Late' stages of training. (d) In both groups, the integration between the fronto-parietal and default mode systems decreased from 'Naive' to 'Late' sessions, but [R]groups differed in the pattern of integration changes between 'Naive' to 'Middle' sessions. Source data are provided as a Source Data file.

Comment 4: Although the authors did include them (Fig 7), I note that not much was made of the role of subcortical regions in the analysis. Admittedly, the resolution is less than ideal, however I imagine that subcortical regions (in particular, basal ganglia and cerebellum) would be very important for learning (and hence, automatizing) the processes required to perform the N-back task effortlessly over the course of training.

Response: The reviewer raises an important point. As described in response to the previous comments, we addressed this issue by extending our analysis on dynamic network reorganization to all large-scale systems. Interestingly, we found five significant session \times group interaction effects in the subcortical system integration ($p < 0.05$, uncorrected; Figure 6d-f; Supplementary Figure 8a-e). Differences between groups in subcortical integration were most pronounced during the early stages of training, i.e. between 'Naive' and 'Middle' phase. The experimental group displayed an inverted U-shaped curve of changes in (i) the integration between the subcortical system and the dorsal attention, and (ii) the integration between the ventral attention system and the cingulo-opercular system; notably, the control

group displayed the opposite pattern. Interestingly, the DM system exhibited exactly opposite effect. Also, note that these dynamic changes involving subcortical regions appear to be non-linear, which is consistent with previous work studying subcortical activity during working memory training (Kühn *et al.*, 2013).

We discussed this result in the *Discussion* section:

*We also observed that groups differed in patterns of changes in the subcortical system coupling. Specifically, the experimental group displayed an inverted U-shaped curve of changes in (i) the integration between the subcortical system and the dorsal attention system, and (ii) the integration between the ventral attention system and the cingulo-opercular system; notably, the control group displayed the opposite pattern. We observed the opposite effect for coupling between the subcortical and default systems. Non-linear changes in subcortical activity were also observed in previous studies of the effects of working memory training (Kühn *et al.*, 2013). Consistent with our results, Kühn *et al.* (2013) found that activity in the subcortical regions increased after one week of working memory training and decreased after 50 days of training. Previous studies suggested that subcortical activity can mediate changes in working memory ability (Dahlin *et al.*, 2008). Because an inverted U-shaped curve of changes in fronto-parietal activity was also observed following working memory training (Hempel *et al.*, 2004; Kühn *et al.* 2013), we speculate that subcortical activity may influence changes in the fronto-parietal system. Yet, results based on observation of brain activity changes cannot provide information on how these two systems interact. Our results show that in the initial training phase, the subcortical system switched coupling from task-positive systems to the default mode system. We observed the opposite pattern for the fronto-parietal system, which instead first increased and then decreased its interaction with the default mode system. We speculate that the subcortical system supports segregation of the task-positive and default mode systems. Future studies using effective connectivity approach could examine whether such a cause and effect relationship exists.*

Comment 5: *On the topic of 7b, the authors may wish to reconsider the utility of comparing two separate 1-sided statistical tests next to one another. The true test of whether the groups differed should come from a paired-comparison between the two groups.*

Response: In the revised version of the manuscript, we modified the manner in which we determined the statistical significance of group differences. We applied multilevel modelling (MLM) for repeated measures data (Snijders *et al.*, 2012) in the same way as we did in the previous version of the manuscript in the case of the DM and FP systems. This approach enabled us to test group- and session-related changes, in addition to differences from the ‘Naive’ to ‘Late’ sessions. For significant main or interaction effects, we turned to examining planned contrasts, with each session compared to the baseline ‘Naive’ session. We also ran *post-hoc* tests to investigate group differences in consecutive sessions (with Bonferroni correction for multiple comparisons). We rewrote the subsection entitled *Dynamic reorganization of default mode and fronto-parietal systems*, as well as relevant parts of the *Discussion*, using solely MLM-related statistics. We summarized these statistics in the Supplementary Tables. We also removed Figure 7, and added new Figure 6 (see responses above) to better visualize results for multiple large-scale systems.

Comment 6: *The language in the abstract was a tad misleading: "We found that whole-brain modularity was higher during the resting state than during the dual n-back task, and increased as demands heightened from the 1-back to the 2-back condition." This made it sound as though modularity was maximal at rest, decreased during 1-back and then increased during 2-back, which is not what the authors report. They may wish to rephrase this to avoid confusion.*

Response: We appreciate the reviewer pointing out this error. In the revised manuscript, we have corrected the sentence:

*We found that whole-brain modularity was higher during the resting state than during the dual n-back task, and **decreased** as demands heightened from the 1-back to the 2-back condition.*

However, after shortening the abstract to 150 words, to meet journal submission requirements, we decided to remove this sentence altogether, and instead focus on the description of changes modulated by the training.

Current version of the *Abstract*:

The functional network of the brain continually adapts to changing environmental demands. The consequence of behavioral automation for task-related functional network architecture remains far from understood. We investigated the neural reflections of behavioral automation as participants mastered a dual n-back task. In four fMRI scans equally spanning a 6-week training period, we assessed brain network modularity, a substrate for adaptation in biological systems. We found that whole-brain modularity steadily increased during training for both conditions of the dual n-back task. In a dynamic analysis, we found that the autonomy of the default mode system and integration among task-positive systems were modulated by training. The automation of the n-back task through training resulted in non-linear changes in integration between the fronto-parietal and default mode systems, and integration with the subcortical system. Our findings suggest that the automation of a cognitively demanding task may result in more segregated network organization.

Comment 7: *Towards the latter half of the Results section, it began to feel as though there were a deluge of new analyses that weren't clearly justified in the Introduction. The authors may wish to update their manuscript such that the reader has a clear appreciation for which analyses will be reported and what the implications of the analyses might be.*

Response: We appreciate this point and agree with the reviewer. Indeed, in the first version of the manuscript we reported the results of an exploratory temporal expansion of our analyses (subsection: *Dynamic fluctuations of default mode recruitment*), that we found interesting, but that was not fully justified in the *Introduction*. As we decided to extend our

previous analysis to focus more on the whole-brain network, we moved these results to *Supplementary Information* (Supplementary Figure 19) and referred to them in the *Discussion* section:

In our exploratory analysis, we also showed that default mode recruitment fluctuated between task conditions and was significantly higher in the 1-back condition than in the 2-back condition (Supplementary Figure 19) and, similar to modularity, increased steadily in both groups. Here we also observed a positive relationship between the change in default mode recruitment and change of modularity from 'Naive' to 'Late' session (Supplementary Figure 21).

Comments from Reviewer 2

Comment 1: *In this manuscript, Finc et al examine changes in network states over the course of training in a demanding cognitive task (adaptive dual n-back). Building on prior work that revealed change in network modularity with training of a simpler motoric task, they are now testing whether similar changes in network segregation and integration occur with the training of a more complex task.*

Overall, the manuscript is quite clearly written and thoughtfully analyzed. At the same time, from a neuroscientific point of view, I am not entirely clear on the take-home message: the network analysis operates at a level of abstraction which can reveal new phenomena, but which also could easily be masking other (simpler) processes that would be more apparent from simpler and more traditional analyses.

~ There are no task-related changes in signal amplitude or variance reported in this manuscript. This is a problem for two reasons. First, it makes it difficult to relate the present work to prior literature which has studied how (e.g. motor, premotor, prefrontal and basal ganglia) regions change their activity over the course of training. Secondly, the absence of these basic signal measurements leaves it uncertain whether any of the network metrics could be biased by basic changes in the neural response during the task. It is incumbent on the authors to demonstrate that the network metrics (which are more abstract, and based on covariance properties of the signal) are not being affected by changes in the mean and variance of the signal, within and across blocks and sessions. One possible solution to this problem would be via a well-calibrated simulation, which would show that the network metrics are immune to such change (...I believe such a simulation would actually show the opposite). However, a more direct solution would be to estimate and report the changing activation patterns in the data (something closer to a traditional GLM analysis), and then demonstrate that the network results persists even when these activation changes are removed. Finally, the authors could argue that they explicitly refuse to include the analysis of the changes in the mean activation patterns, because they believe that these signals obscure the more fundamental neural processes, which are revealed by the network perspective. In that case, they would need to

make that argument compellingly. Overall, I think the simplest thing to do is to report descriptive statistics on the local activation patterns, and how they change over blocks and over training.

Response: We thank the reviewer for thoughtful comment. First, we note that prior work in long-term training has previously demonstrated that functional connectivity patterns can provide markers of individual differences in learning that are not obtained from a GLM (Bassett *et al.* 2015), serving to motivate our choice of main analysis in this paper. Of course, we are also aware of the possible influence of task-related brain activity changes on functional connectivity estimates. To overcome this issue, we remove all task-related changes in the signal amplitude by regressing out a boxcar function for the task blocks convolved with the HRF and its derivatives (Fair *et al.* 2007). We then estimated functional connectivity using the signal containing no task-related variability. In this way, our results should not be solely driven by elicited activation, but also by effective inter-regional coupling.

Nevertheless, we agree that a direct study of activation would still be helpful in advancing our understanding, while also being of interest to future readers. Because we designed the study with the goal of investigating network reorganization throughout training, we had not originally included a traditional GLM analysis. In response to this reviewer comment, we now perform a GLM analysis and include the results of that analysis in the *Supplementary Information*; as the reviewer points out, this inclusion allows readers to compare our findings with those of prior literature on the effects of working memory training on activation patterns. In the first level analysis, we compared 2-back vs. 1-back activation patterns (two-sided) for all subjects to identify brain areas activated and deactivated in a more difficult 2-back condition. Next, we ran a second-level GLM analysis to investigate consistent patterns of task activation in all sessions and both groups. To ensure that the GLM analysis was comparable with our functional connectivity analysis, we calculated the mean z -score for the first-level β maps for each ROI from the Power *et al.* (2011) parcellation. Then, for all large-scale systems we calculated the mean z -score that reflected the effect size for that network, and sorted them from the lowest to the highest. As expected, we found the largest condition-related increase in brain activity in the fronto-parietal, dorsal attention, and salience systems, while we found the largest decrease in brain activity in auditory, somatomotor, and default mode systems (see *Supplementary Figure 10*). We shared all first-level normalized β maps in a public repository: <https://osf.io/wf85u/> (WM_training_modularity_data/neuroimaging/04-glm/zmaps)

We added a description of the GLM analysis into the *Supplementary Methods*:

Standard GLM analysis

To enable reference to the prior literature on the effects of working memory training on activation patterns, we additionally performed a standard General Linear Model (GLM) analysis. In the first level of the GLM analysis, we compared 2-back vs. 1-back activation patterns (two-sided) for all subjects to identify brain areas activated and deactivated in a more difficult 2-back condition. Then, we ran a second-level GLM analysis to investigate consistent

patterns of task activation in all sessions and both groups. To make GLM analysis comparable with our functional connectivity analysis, we calculated the mean z-score for the first-level β maps for each ROI from the Power et al. (2011) parcellation (Supplementary Figure 10). Then, for all large-scale systems we calculated the mean z-score that reflected the effect size for each network, and sorted them from the lowest to the highest. Next, we used multilevel modelling to test for session \times group interactions for each system (see Supplementary Figure 11 and Supplementary Table 8-9).

Supplementary Figures:

Supplementary Figure 10 | Brain activity for 2-back vs. 1-back contrast (two-sided) estimated with a standard GLM for all subjects and sessions. (a) Glass brain visualization of activity thresholded at a z-score level +/- 8. **(b)** Brain activity plotted on 264 ROIs from the Power et al. (2011) parcellation. **(c)** Barplot representing the z-score values averaged over ROIs belonging to predefined large-scale systems. The most active ROIs belonged to the fronto-parietal (FP), dorsal attention (DA), and salience systems (SAL). The most deactivated ROIs belonged to the auditory (AU), somatomotor (SOM), and default mode (DM) systems. Remaining abbreviations: cerebellum (CER), cingulo-opercular (CO), memory (MEM), uncertain (UNC), somatomotor (SOM), subcortical (SUB), ventral attention (VA), and visual (VIS). Source data are provided as a Source Data file.

We further ran multilevel modeling, in a manner analogous to our network analysis, to investigate session \times group interactions for all systems from the parcellation. We found significant interaction effects for the salience and visual systems ($p < 0.05$, FDR-corrected; see Supplementary Figure 11 and Supplementary Table 8-9). Specifically, compared to the control group, participants from the experimental group displayed a significantly greater decrease in the activation of the salience system from 'Naive' to 'Early' sessions ($\beta = -1.10$, $t(120) = -3.44$, $p = 0.0008$), from 'Naive' to 'Middle' sessions ($\beta = -1.32$, $t(120) = -4.12$, $p = 0.0001$), and from 'Naive' to 'Late' sessions ($\beta = -0.96$, $t(120) = -2.99$, $p = 0.003$). The experimental group also displayed a larger decrease in the activation of the visual system from the 'Naive' to 'Middle' sessions: $\beta = -0.80$, $t(120) = -2.69$, $p = 0.008$).

Decreased activation in task-related brain areas was reported by several previous studies investigating the effects of working memory training (Garavan *et al.*, 2000; Hempel *et al.*, 2004; Landau *et al.*, 2004; Kühn *et al.*, 2013). These training-related effects were commonly interpreted as a reflection of an increased neural efficiency within the network engaged in the task (Kelly and Garavan, 2005). GLM-based analysis, however, does not provide an explanation for the plastic changes on the whole-brain network. The fronto-parietal system dynamically interacts with other large-scale systems (Cocchi *et al.*, 2013), and it is therefore reasonable to expect that working memory training might influence these interactions. To characterize these changes in greater detail, we performed additional analysis of training-related changes in the dynamics of all large-scale systems in the revised version of the manuscript (for a detailed description of our findings see our response to the comment #1 raised by reviewer #1). In summary, we observed training-related increases in the segregation of the default mode and task-positive systems (Figure 6a-c, Supplementary Figure 7). We also found that the experimental and control groups differed in their patterns of task-related changes in integration between the fronto-parietal and default mode systems, and in the integration between the subcortical system and other large-scale systems (Figure 6d-f; Supplementary Figure 8). These results fill an existing gap in our understanding of the dynamic network reorganization that occurs during the automation of a demanding working memory task.

Finally, we also sought to determine whether the activity of large-scale systems estimated with the standard GLM is correlated with the network recruitment in corresponding systems (Supplementary Figure 12). We did not find a significant correlation between activation, measured as z-score of β estimates, and recruitment values when considering all systems, all sessions, and all subjects ($r = 0.02$, $p = 0.41$). We further tested whether changes in systems activity from 'Naive' to 'Late' sessions were correlated with corresponding changes in systems recruitment. We did not find any significant correlation between these variables, either when considering both control and experimental groups ($r = -0.06$, $p = 0.15$), or when considering the experimental group alone ($r = -0.07$, $p = 0.26$). We also did not observe a significant relationship between changes in the FP or DM activity and changes in recruitment of these systems or FP-DM integration either for the experimental group or for both groups (all $p > 0.05$). These findings support the notion that the dynamic network approach provides a unique view on task-related and training-related brain reorganization. Further studies, however, are necessary to relate dynamic network measures and brain activity estimates obtained with a standard GLM-based approach.

The code for the supplementary GLM analysis can be found here:
https://github.com/kfinc/wm-training-modularity/tree/master/05-glm_analysis

Supplementary Figures:

Supplementary Figure 12 | Relationship between systems recruitment and systems activation estimates. (a) There was no significant relationship between systems activation (z-score; 2-back minus 1 back) and systems recruitment values when considering all systems, all sessions, and all subjects. We further tested whether changes (Δ) in systems activity from 'Naive' to 'Late' sessions were correlated with changes in systems recruitment. We did not find any significant correlation between these two variables, either when considering (b) all subjects, or (c) when considering only the experimental group. Source data are provided as a Source Data file.

Supplementary Figure 11 | Cross-sessions changes in brain activity for 2-back vs. 1-back contrast (two-sided) estimated with a standard GLM. Groups differed significantly by session for the salience (SAL) and visual (VIS) systems ($p < 0.05$, FDR-corrected). Specifically, compared to the control group, participants from the experimental group displayed significantly greater decreases in the activation of the salience system from the 'Naive' to 'Early' sessions ($\beta = -1.10$, $t(120) = -3.44$, $p = 0.0008$), from the 'Naive' to 'Middle' sessions ($\beta = -1.32$, $t(120) = -4.12$, $p = 0.0001$), and from the 'Naive' to 'Late' sessions ($\beta = -0.96$, $t(120) = -2.99$, $p = 0.003$). The experimental group also displayed a larger decrease in the activation of the visual system from the 'Naive' to 'Middle' sessions: $\beta = -0.80$, $t(120) = -2.69$, $p = 0.008$. Source data are provided as a Source Data file.

Discussion section:

The dynamic network approach extends our understanding of training-related changes in brain function. Studies focusing on changes in brain activity during a working memory training reported a decrease of task-positive systems activation (Hempel et al., 2004; Kühn et al., 2013), commonly interpreted as a reflection of increased neural efficiency within systems engaged in the task (Kelly and Garavan, 2005). Here, we reported a similar effect using a standard GLM-based approach (see Supplementary Figure 10-11, Supplementary Table 8-9). However, we also showed that our findings on the dynamic network changes can not be simply explained by the changes in brain activity (Supplementary Figure 12). The fronto-parietal system dynamically interacts with other large-scale systems (Cocci et al., 2013), and it is reasonable to expect that working memory training might influence interactions in the whole network. We observed training-related increases in the segregation of the default mode and task-positive systems that suggest more efficient and less costly processing within these systems after training. Accordingly, greater segregation of the default mode system and task-positive systems and smaller integration between these systems were associated with behavioral performance improvement. Moreover, we showed that an increase of integration between multiple large-scale systems in early phase of training was related to a greater behavioral improvement in the experimental group, indicating that some level of network integration is necessary when the task is not fully automated. Taken together, the dynamic network approach provides a unique insight into the plasticity and dynamics of the human brain network.

Comment 2: *~~ Data from a “2-back” condition for the Control group are reported in Figure 2B, and yet neither the manuscript introduction nor methods make any mention of the existence of this condition. Instead, the Methods explicitly state:*

“Participants from the control group performed a single 1-back with auditory or visuospatial stimuli variants.”

Response: We thank the reviewer for raising this point and apologize for the ambiguity. Only participants from the experimental group trained their working memory using an adaptive dual n-back task. However, during four fMRI scanning sessions, both groups performed the same version of the dual n-back task with two task conditions (1-back and 2-back). This experimental structure enabled us to compare the effect of mastering the task during training to the effect of repeated exposure to the task. All participants were refamiliarized with the task before each fMRI scanning session. To clarify these points, we now include the following additional explanation in the manuscript:

Introduction:

To address these questions, participants underwent four functional magnetic resonance imaging (fMRI) scans while performing an adaptive dual n-back task taxing working memory over a 6-week training period. The dual n-back task consisted of visuospatial and auditory tasks that were performed simultaneously (Jaeggi et al., 2008). In the visuospatial portion of

the task, participants had to determine whether the location of the stimulus square presented on the screen was the same as the location of the square n-back times in the sequence; in the auditory portion of the task participants had to determine whether the heard consonant was the same as the consonant they heard n-back times in the sequence. To ensure that participants mastered the task due to training, and not simply due to a repeated exposure to the task, we compared their performance to an active control group. While participants from both the experimental and the control groups performed the same version of the dual n-back task, with interleaved 1-back and 2-back blocks, inside the fMRI scanner, only the experimental group trained their working memory using an adaptive version of the task in 18 training sessions outside the scanner. We examined network reconfiguration using static functional network measures to distinguish distinct task conditions, and using dynamic network measures to study fluctuations of network topology across short task blocks.

Methods (subsection *Experimental Procedures*):

Two versions of the dual n-back task were used: (1) an adaptive dual n-back was used in the training sessions of the experimental group only, and (2) an identical dual n-back task with two conditions (1-back and 2-back) used during fMRI scanning of both groups. Both scanning and training versions of the dual n-back task consisted of visuospatial and auditory tasks performed simultaneously.

For clarification purposes we also moved the portion of the text that describes the task used in the fMRI scanning from the *Data acquisition* subsection to the *Experimental Procedures* subsection, which resulted in two distinct paragraphs, starting with:

In the training version of the task, the n-level of the dual n-back task increased adaptively when participants achieved 80% correct responses in the trial, and the n-level decreased when participants made more than 50% errors in the trial. (...)

In the fMRI scanning version of the task, participants performed the dual n-back task with two levels of difficulty: 1-back and 2-back.

Comment 3: ~ On page 5, there is the following claim:

“Collectively, these results demonstrate that modularity increased for both 1-back and 2-back task conditions in the experimental group but only during the 1-back condition for the control group. The findings suggest that higher brain network segregation during the 2-back condition may be a consequence of the 6-week working memory training.”

However, the analyses provided do not support these claims. The analysis shows that some conditions exceed a statistical threshold, while others do not exceed that threshold; this does not

provide a statistical test of the difference (or the interaction) across conditions. Please provide a direct test if you want to make this claim.

Response: We agree that these sentences could be misleading, particularly as we report that the group \times session interaction for the static modularity analysis was not significant. In the original version of the manuscript, we wrote:

Results (subsection *Whole-brain network modularity changes*):

However, the experimental and control groups did not differ by session ($\chi^2(1) = 1.44, p = 0.69$), nor did we observe a significant session by condition interaction ($\chi^2(1) = 1.50, p = 0.68$). To summarize, we showed that the modularity of the functional brain network generally increased during the training period. (...)

The change of modularity from 'Naive' to 'Late' sessions did not significantly differ between groups for the 1-back condition ($t(39.88) = -0.80, p = 0.42$) or for the 2-back condition ($t(39.99) = -1.05, p = 0.30$).

Also see the summary in the *Results*:

To summarize, we showed that the modularity of the functional brain network generally increased during the training period. However, the degree to which modularity changed between load conditions remained stable. Groups did not differ significantly in the change of modularity. These results suggest that the functional brain network shifts towards a more segregated organization as a result of behavioral improvement after training and also after repeated exposure to the task.

In the revised manuscript, we now rephrase the concluding sentences to focus exclusively on significant changes within the experimental group.

Rephrased sentences:

~~*Collectively, these results demonstrate that modularity increased for both 1-back and 2-back task conditions in the experimental group but only during the 1-back condition for the control group. The findings suggest that higher brain network segregation during the 2-back condition may be a consequence of the 6-week working memory training.*~~

These results indicate that the experimental group displays increased network modularity for both task conditions when moving from 'Naive' to 'Late' sessions, suggesting that network segregation may be a consequence of the 6-week working memory training. While the same effect was not present in the control group, we did not observe a significant group \times session

interaction, and therefore further work is needed to inform our conclusions.

The absence of a group \times session interaction effect is an important finding, and we therefore decided to discuss this issue thoroughly in the revised *Discussion*. We argue that the exposure of the control group to the task during the fMRI scanning sessions was sufficient to considerably increase their performance, leading to lower statistical power in detecting group differences. In other words – the control group has also been influenced by the effect of training just by mere exposure to the task. This observation could potentially explain our finding regarding static modularity.

Discussion:

Interestingly, we did not observe differences between the experimental and control group in the increase of network modularity. The control group displayed a small increase of modularity in 1-back condition, suggesting that the segregation of the functional brain network may increase rapidly, also in the response to repeated exposure to the task. The control group performed the dual n-back task four times during scanning sessions, which resulted in a small behavioral improvement. This result suggests that the increase of the network segregation may be sensitive to varying intensity of training in the task. Future studies with a larger sample size should examine whether such gradation exists.

Comment 4: *~~ Does behavior vary cross blocks, and does the behavioral variability decrease with training? Given that one of the main results concerns increases in network recruitment (which measures something like the consistency of module assignment), it seems important to test whether the stability of the networks is related to greater stability (e.g. via more sustained attention) in the cognitive and behavioral performance, as a function of training.*

Response: The reviewer's idea is very interesting, and we performed the suggested analysis. Here we provide the revised supplementary text reporting the results.

Supplementary Methods (subsection *Behavioral variability analysis*):

To assess measures of behavioral variability, we calculated (1) block-wise variants of the two behavioral performance measures, d' and penalized reaction time (pRT), and (2) the standard deviation of these measures over task blocks. For consistency with the measures used in the main text, for both block-wise measures we considered the average value over both stimulus modalities (visual and auditory). This procedure resulted in two measures of block-to-block behavioral variability for each participant and session: the standard deviation of d' ($\sigma_{d'}$) and the standard deviation of pRT (σ_{pRT}). We then used a multilevel analysis to investigate group \times session interactions. Note that these measures of behavioral variability can potentially

capture two distinct effects: (1) more or less consistent performance during the 1-back or 2-back blocks, and (2) greater or lesser decreases in behavioral performance from the 1-back to the 2-back condition. Both effects of more consistent performance during a single task condition and a lesser decrease in performance from the 1-back to the 2-back condition would result in an overall decrease in the behavioral variability measures of $\sigma_{d'}$ and σ_{pRT} .

Supplementary Figures:

Supplementary Figure 4 | Block-to-block variability in behavioral performance modulated by training. (a) Standard deviation of d' ($\sigma_{d'}$) estimated across task blocks, for which we found a significant main effect of session ($\chi^2(3) = 9.61$, $p = 0.02$). Specifically, the standard deviation of d' decreased from 'Naive' to 'Early' sessions for all participants ($\beta = -0.14$, $t(39) = -2.46$, $p = 0.02$). Both group and session \times group interaction effects were not significant ($p > 0.05$). Error bars represent the 95% confidence intervals. (b) Standard deviation of penalized reaction time (σ_{pRT}), for which we found a significant group effect ($\chi^2(1) = 7.39$, $p = 0.006$). In general, participants from the experimental group had lower pRT variability ($\beta = -29.00$, $t(40) = -2.80$, $p = 0.008$) than participants from the control group. Both the effect of session and the session \times group interaction were not significant ($p > 0.05$). (c, d) correlation between the across-session change in behavioral variability measured as standard deviation of d' and the across-session change in (c) somatomotor and (d) subcortical systems recruitment ($p < 0.05$, uncorrected). Source data are provided as a Source Data file.

Because the only significant difference in behavioral variability was a decrease in the standard deviation of d' from 'Naive' to 'Early' sessions, we only focused on $\sigma_{d'}$ for these two sessions and its possible relation to the change in network stability for the corresponding sessions. For each large-scale network we calculated the correlation between (i) the change in $\sigma_{d'}$ from 'Naive' to 'Early' sessions and (ii) the change in network recruitment from 'Naive' to 'Early' sessions. We found no significant correlations between the change of network

stability and the change of behavioral performance stability (all adjusted p -values > 0.05 , FDR-corrected). We did, however, observe weak positive correlations between the change in $\sigma_{d'}$ and the change in the recruitment of the somatomotor ($r = 0.35$, $p = 0.02$; uncorrected) and subcortical ($r = 0.33$, $p = 0.03$; uncorrected) systems.

Also, note that in the main manuscript, we used a change of d' ($\Delta d'$; 2-back minus 1-back) and penalized reaction time (Δ pRT; 2-back minus 1-back) as a measure of behavioral performance. Before training the differences in d' and pRT between conditions were substantial (a much lower d' and longer pRT were observed for the 2-back task); after training, the experimental group displayed no difference in performance (no difference between the 1-back and 2-back conditions). We may conclude that $\Delta d'$ and Δ pRT also captured variability in the sense of differences between conditions.

Comment 5: ~ On page 8: The DMN and FPM regions were not the only ones exhibiting increased recruitment with training — as the authors note on page 8, the ventral attention, salience, cingulo- opercular, and auditory systems also show the effect. So why are the DMN and FPN systems emphasized in the Abstract and the Discussion? Have the authors conducted further analyses on these other networks, and did they find fewer links to behavior? Less change in integration over time? It is not clear from the manuscript how we are supposed to understand or interpret the changes in these additional systems, and how they differ fundamentally from the changes reported for the FPN and DMN.

Response: The reviewer raises a good point, which also occurred to reviewer 1. In the original version of the manuscript we focused our analyses on dynamic network changes in the task-relevant FP and DM systems, for which we had specific hypotheses. Nonetheless, in light of both reviewers' feedback, we now broaden the scope by extending the multilevel modelling analysis of recruitment and integration to all 13 large-scale systems.

Interestingly, apart from our original findings on the DM and FP systems, we found more patterns of network reorganization, encompassing other task-positive systems like the salience, cingulo-opercular and dorsal attention systems, as well as the auditory and subcortical systems. We divided the significant session effects into three categories: (1) increased recruitment of multiple systems (including the DM and salience), (2) decreased integration between DM and task-positive systems, and (3) increased integration between task-positive systems (Figure 6a-c). Together, this set of results is consistent with our hypothesis that DM system independence should increase with task automation.

We also investigated possible links to behavior for all recruitment and integration changes over the course of training. For d' (the difference in d' between 1-back and 2-back conditions), we found several significantly correlated dynamical network measures. We found that the change in the DM and salience recruitment and integration between the salience and FP systems is positively correlated with the change in d' , while the change in DM-FP integration and DM-salience integration is negatively correlated with the change in d' . For penalized reaction time (pRT; the difference in pRT between the 1-back and 2-back conditions), we found that only the change in DM recruitment and the change in DM-FP integration was significantly correlated with the change in behavior. Note, that the

correlation for the change in the d' measure has opposite sign when compared to the correlation with the change of pRT, consistent with the fact that these two measures have different interpretations (the lower pRT, the better; the higher the d' , the better). We now depict these correlations on Figure 7a (see below).

Figure 7 | Relationship between the change in network dynamics and the change in behavior. Colored tiles represent all significant correlations ($p < 0.05$, uncorrected; * $p < 0.05$ FDR-corrected). (a) Pearson correlation coefficient (r) between the cross-session changes in recruitment (or integration) and the cross-session changes in d' ($\Delta d'$) observed for both experimental and control group. (b) Relationship between the changes changes in recruitment (or integration) and the changes in d' during early phase of training of the experimental group. Abbreviations: auditory (AU), cerebellum (CER), cingulo-opercular (CO), default mode (DM), dorsal attention (DA), fronto-parietal (FP), memory (MEM), salience (SAL), somatomotor (SOM), subcortical (SUB), uncertain (UNC), ventral attention (VA), and visual (VIS). Source data are provided as a Source Data file.

Supplementary Figure 9 | Relationship between the change in network dynamics and the change in behavior. Colored tiles represent all correlations ($p < 0.05$, uncorrected; * $p < 0.05$ FDR-corrected). (a) Pearson correlation coefficient (r) between the cross-session changes in recruitment (or integration) and the cross-session changes in penalized reaction time (ΔpRT) observed for both experimental and

control group. **(b)** Relationship between the changes in recruitment (or integration) and the changes in pRT during early phase of training of the experimental group. (...)

Abstract:

In a dynamic analysis, we found that the autonomy of the default mode system and integration among task-positive systems were modulated by training. The automation of the n-back task through training resulted in non-linear changes in integration between the fronto-parietal and default mode systems, and integration with the subcortical system.

Discussion:

Using dynamic network metrics, we showed that the default mode system increased its recruitment in both groups, indicating that regions within this system were coupled with other communities less often. Moreover, the experimental group displayed an increased fronto-parietal recruitment and an inverted U-shaped curve of integration between the default mode and fronto-parietal systems with training. Enhanced default mode intra-communication and decreased inter-communication with the fronto-parietal system were associated with better behavioral outcomes after training. We also observed significant changes in dynamic network topology beyond the fronto-parietal and default mode systems. In particular, regardless of the group, we observed an increased recruitment of the salience and auditory systems, decreased integration between the default mode and other task-positive systems (including salience and cingulo-opercular), and increased integration between task-positive systems (including fronto-parietal and salience, dorsal attention and salience, dorsal attention and cingulo-opercular). These results suggest the existence of the trade-off between segregation and integration: whereas segregation increases between some systems, the integration increases or decreases between others.

(...)

Our results are also consistent with prior observations that the default mode and fronto-parietal systems may interact in a task-dependent manner with the salience, cingulo-opercular, and dorsal attention systems (Bressler et al 2010; Cocchi et al., 2013). Bressler and Menon (2010) proposed a model whereby efficient cognitive control is supported by the dynamic switching between functionally segregated fronto-parietal and default mode systems mediated by cingulo-opercular and salience systems. Cocchi et al. (2013) proposed that task-related reconfiguration is possible through flexible interactions within and between overlapping meta-systems: (i) the executive meta-systems, responsible for the processing of sensory information, and (ii) the integrative meta-system, responsible for flexible integration of brain systems. These two meta-systems are composed of transient coupling between three large-scale systems: the frontoparietal system, the cingulo-opercular/salience system, and the default mode system. During high-demand task conditions the executive meta-system is formed by extensive interactions between fronto-parietal and cingulo-opercular/salience systems, and the default mode system is more segregated and less integrated with the fronto-parietal system (Leech et al., 2012). Our results extend these findings by presenting the evolving reconfigurations of large-scale networks during mastery of the working memory task.

We showed that regardless of the group the default mode system reduced coupling with the cingulo-opercular and salience systems. These results suggest that increased segregation of the default mode and task-positive networks may be a consequence of more efficient task performance. A similar pattern of changes was observed across two different subdivisions of the cortex into systems (Power and Schaefer), together suggesting that the salience and cingulo-opercular systems that are thought to be responsible for switching between antagonistic fronto-parietal and default mode systems, appear to be more integrated with the fronto-parietal system and less integrated with the default mode system. This pattern of relations may be due to diminished requirements for switching between these two systems when the task is well trained.

Comment 6: ~ Abstract:

“behavioral automation” and “behavioral adaptation” seem to be used to refer to the same process, but presumably they mean different things?

Response: The reviewer is correct; “behavioral automation” and “behavioral adaptation” do not refer to the same process, and should not be used interchangeably. We used the word “adaptation” to refer to a broader range of network and behavior adjustments in response to a changing environment, including the effects of learning and training. We used the word “automation” to refer to mastering the trained task, which after training can be performed automatically, with almost no effort. Nevertheless, to minimize the potential for confusion, in the revised version of the paper, we only use the term “behavioral automation” in the *Abstract*.

Comment 7: ~ The “recruitment” terminology seems a little awkward. On p.5 we read that “Intuitively, high recruitment indicates that nodes of the system are consistently assigned to the same module across different layers.” I am struggling to understand why this concept should be summarized as “recruitment”, rather than “consistency”, “reliability”, “community stability”, or something else?

Response: We agree that the “recruitment” terminology may not be as self-descriptive as “consistency” or “stability”. Yet, we decided to use the “recruitment” terminology to maintain consistency with previous work, especially a study by Mattar *et al.* (2015) who described the recruitment and integration coefficients in their *functional cartography* framework, and the study by Bassett *et al.* (2015) who used these measures to examine the effects of motor training on the dynamics of functional network organization. In those studies, the term recruitment was chosen because the temporal consistency of the measure reflected the non-random nature of brain dynamics in which a functional module is persistently recruited for a task. To make this decision clearer earlier in the manuscript, we have added the following explanation in the *Results* section:

Following the functional cartography framework described by Mattar et al. (2015), we used P to calculate the recruitment of all 13 large-scale systems, as well as the pairwise integration among them (see Methods for details). We selected these measures to maintain consistency

with the methodology used in a previous study on the effects of motor sequence training on the dynamics of functional brain networks (Bassett et al., 2015). Recruitment is defined for each system separately, while integration is calculated for pairs of systems. Intuitively, high recruitment indicates that nodes of the system are consistently assigned to the same module across different layers; this consistency reflects the non-random nature of brain dynamics in which a functional module is persistently recruited for a task. High integration indicates that pairs of nodes (where one region of the pair is located in one system and the other region of the pair is located in the other system) are frequently classified in the same module across layers).

Comment 8: *~~ The manuscript would be stronger if slightly more detail was provided concerning the n-back tasks (especially related to the specific audiovisual signals that are detected by participants, and the blocking structure) in the Introductory section.*

Response: We appreciate the suggestion and are happy to provide further detail. In the visuospatial version of the task, we used a blue square as a stimulus that was presented sequentially in one of 8 locations on a 3×3 grid (with a white fixation cross in the middle). As an auditory signal we used 8 Polish consonants (b, k, w, s, r, g, n, z) that were played sequentially in headphones. The task was block-designed without any systematic variation to block length or to block order. Each session of the task consisted of 20 blocks (30 s per block; 12 trials) of alternating 1- and 2-back conditions. In the revised manuscript, we now include additional description of the dual n-back task (including the stimuli used and the blocking structure) in the *Introduction* section, as the reviewer recommends. A more detailed description is also provided in the *Methods* section (*Experimental procedures*).

Introduction:

The dual n-back task consisted of visuospatial and auditory tasks that were performed simultaneously (Jaeggi et al., 2008). In the visuospatial portion of the task, participants had to determine whether the location of the stimulus square presented on the screen was the same as the location of the square n-back times in the sequence; in the auditory portion of the task participants had to determine whether the heard consonant was the same as the consonant they heard n-back times in the sequence. To ensure that participants mastered the task due to training, and not simply due to a repeated exposure to the task, we compared their performance to an active control group. While participants from both the experimental and the control groups performed the same version of the dual n-back task, with interleaved 1-back and 2-back blocks, inside the fMRI scanner, only the experimental group trained their working memory using an adaptive version of the task in 18 training sessions outside the scanner.

Method (Experimental procedures):

Both scanning and training versions of the dual n-back task consisted of visuospatial and auditory tasks performed simultaneously}. Visuospatial stimuli consisted of 8 blue squares presented sequentially for 500 ms on the 3 × 3 grid with a white fixation cross in the middle of the black screen; auditory stimuli consisted of 8 Polish consonants (b, k, w, s, r, g, n, z) played sequentially in headphones.

Also see comment below for more details about the blocking structure.

Comment 9: *~~ Please make clear whether the consecutive blocks of the n-back tasks are identical, or whether there is any interleaving or other systematic parameter variation across blocks. This is critical for interpreting the dynamic network analyses, which depend on changes in neural response across blocks.*

Response: The length of the task blocks was identical and we did not add any other systematic parameter variation across the blocks. We have now clarified these facts in the revised version of the *Experimental Procedures* subsection (*Methods* section):

In the fMRI scanning version of the task, participants performed the dual n-back task with two levels of difficulty: 1-back and 2-back. Each session of the task consisted of 20 blocks (30 s per block; 12 trials with 25% of targets) of alternating 1- and 2-back conditions. To enable for dynamic network comparison across blocks, we did not add any systematic variation to block length or to block order.

Comment 10: *~~ Please provide the reader with a summary statement of the many analyses included in the section “Dynamic reorganization of default mode and fronto-parietal systems” on pp.5-7; there are many analyses included in the latter part of this section, but their interpretation seems muddy to me. It would help to some more interpretation or commentary interleaved with the results on page 7 in particular.*

Response: We agree with the reviewer. In the revised manuscript, we now include summary statements in each paragraph of the subsection *Dynamic reorganization of large-scale systems*.

In the paragraph summarizing changes in the recruitment (and integration) of the default mode and fronto-parietal systems, we now state:

Collectively, these results suggest that the increase of fronto-parietal system recruitment and the decrease of integration between the default mode and fronto-parietal systems reflect training-specific changes in dual n-back task automation. In contrast, the increase in default mode system recruitment may reflect more general effects of behavioral improvement, as it was observed in both experimental and control groups.

In the paragraph summarizing changes in the recruitment (and integration) of all large-scale systems, we now state:

These results suggest that the increase of within-module stability, the increase of default mode system independence from task-positive systems, and the decrease of integration between task-positive systems reflect general effects of task training.

In the paragraph summarizing the relationship between across-sessions change in behavioral performance and across-sessions changes in the recruitment (or integration) of large scale systems, we now state:

In summary, a higher increase of stability in the default mode and salience systems, together with a decrease of default mode - task-positive systems integration may support behavioral improvement in the task, regardless of whether the task was additionally trained or not.

In the paragraph summarizing the the session \times group interactions:

These results suggest that task automation during initial stages of working memory training might also be supported by an increased communication between subcortical and other large-scale systems.

In the paragraph summarizing the relationship between change in behavioral performance during initial stage of training and changes in the recruitment or integration in the experimental group:

This pattern of associations between behavioral and network changes suggests that inter-systems communication might be necessary for efficient task performance during initial stages of training.

In the paragraph summarizing all results presented in the *Dynamic reorganization of large-scale systems* subsection:

In summary, we observed two patterns of dynamic changes in network topology following working memory training. The first pattern reflects improved behavioral performance and is characterized by a gradual increase in default mode autonomy and in the integration between task-positive systems. The second pattern reflects changes related to task automation specifically in the experimental group and is characterized by non-linear changes in default mode - fronto-parietal integration, and in the integration with the subcortical system.

Comment 11: ~~ “comprised of” should be “composed of” in all instances in the manuscript

Response: We have corrected these instances in the revised manuscript.

Comments from Reviewer 3

Comment 1: *The authors investigate the dynamic reconfiguration of functional brain networks over 6 weeks of training via a static modularity analysis and a multilayer modularity analysis. In the four fMRI scans within the 6 weeks, they assess the brain network modularity, as well as the recruitment and integration of the default mode and frontoparietal systems from a static and a dynamic perspective. The overall results show that whole-brain modularity differed between rest and task conditions, being lowest in the condition with more demands. They also show that through training, the brain modularity and the recruitment of frontoparietal and default mode systems will increase, indicating an increase of segregation of brain network. The study is well conceptualized. There is a relatively clear and reasonable hypothesis of what the authors expected, methods are state of the art and the results are very interesting. However, my major concerns relate to this work from both methodological and clinical perspectives, as detailed below.*

1. *The author said that “participants from the control group performed a single, non-adaptive, 1-back working memory task”. But from the results section, control groups also have graph measures calculated from the 2-back condition, which is a bit confusing.*

Response: We thank the reviewer for raising this point and apologize for the ambiguity. Only participants from the experimental group trained their working memory using an adaptive dual n-back task. However, during four fMRI scanning sessions, both groups performed the same version of the dual n-back task with two task conditions (1-back and 2-back). This experimental structure enabled us to compare the effect of mastering the task during training to the effect of repeated exposure to the task. All participants were refamiliarized with the task before each fMRI scanning session. To clarify these points, we now include the following additional explanation in the manuscript:

Introduction:

To address these questions, participants underwent four functional magnetic resonance imaging (fMRI) scans while performing an adaptive dual n-back task taxing working memory over a 6-week training period. The dual n-back task consisted of visuospatial and auditory tasks that were performed simultaneously (Jaeggi et al., 2008). In the visuospatial portion of the task, participants had to determine whether the location of the stimulus square presented on the screen was the same as the location of the square n-back times in the sequence; in the auditory portion of the task participants had to determine whether the heard consonant was the same as the consonant they heard n-back times in the sequence. To ensure that participants mastered the task due to training, and not simply due to a repeated exposure to the task, we compared their performance to an active control group. While participants from

both the experimental and the control groups performed the same version of the dual n-back task, with interleaved 1-back and 2-back blocks, inside the fMRI scanner, only the experimental group trained their working memory using an adaptive version of the task in 18 training sessions outside the scanner. We examined network reconfiguration using static functional network measures to distinguish distinct task conditions, and using dynamic network measures to study fluctuations of network topology across short task blocks.

Methods (subsection *Experimental Procedures*):

Two versions of the dual n-back task were used: (1) an adaptive dual n-back was used in the training sessions of the experimental group only, and (2) an identical dual n-back task with two conditions (1-back and 2-back) used during fMRI scanning of both groups. Both scanning and training versions of the dual n-back task consisted of visuospatial and auditory tasks performed simultaneously.

For clarification purposes we also moved the portion of the text that describes the task used in the fMRI scanning from the *Data acquisition* subsection to the *Experimental Procedures* subsection, which resulted in two distinct paragraphs, starting with:

In the training version of the task, the n-level of the dual n-back task increased adaptively when participants achieved 80% correct responses in the trial, and the n-level decreased when participants made more than 50% errors in the trial. (...)

In the fMRI scanning version of the task, participants performed the dual n-back task with two levels of difficulty: 1-back and 2-back.

Comment 2: 2. In the statistical analysis, do the authors perform any outlier detection? It seems like some potential outliers might bias the observations (especially Figure 3 b and f, Figure 5).

Response: In the original version of the manuscript we did not perform any outlier detection. During revision, we closely inspected the results presented on Figure 3b and noticed a mistake that we had made when calculating the behavioral measures. Our first implementation of the function calculating pRT did not include situations when subjects were supposed to answer but omitted the response (referred to as a *miss* in signal detection theory). That oversight resulted in a decreased measure accuracy for subjects with high number of misses (described as *low sensitivity* in signal detection theory). For these subjects, we had therefore underestimated the values of pRT. In other words, we did not penalize for situations where there was no reaction time but subjects were supposed to answer for congruent stimuli. We calculated new values of pRT using the right formula, and we then reran all parts of the analysis where behavioral data were used. The new script implementing the functions to calculate behavioral measures can be found here:

https://github.com/kfinc/wm-training-modularity/blob/master/01-behavioral_data_analysis/00-log_processing.ipynb.

We updated all behavioral results using the corrected measure. The general pattern of between-group differences remained the same. In the revised version of the manuscript we rewrote the proper definition of the penalized reaction time:

(...) where n is the sum of all subject responses and incorrect response omissions, and x_i was obtained from the following formula:

RT_i , if answer was correct
 2000, if answer was incorrect
 2000, if the correct response was omitted

We also decided to shift our focus to d' in the main text and move results calculated using pRT into the *Supplementary Information*. We based this decision on two arguments. First, d' is not suffering from drawbacks of pRT. In pRT, due to the design of our dual n-back task, we had to equally penalize for missing responses and false alarms, whereas d' treats these two events differently. Second, d' is more sensitive measure than pRT, with a higher inter-subject variability. Despite these changes, our main results and conclusions remained unchanged.

Next, we performed an outlier detection on the three behavioral measures: d' , pRT and accuracy. We used behavioral data from from both the easier 1-back condition and the more demanding 2-back condition during the ‘Naive’ session (averaged over stimulus modalities – auditory and visual). We considered subjects as outliers if their behavioral score was either lower than the average score minus three standard deviations or higher than the average score plus three standard deviation (z-score: $|z| > 3$). This criterion led to the following thresholds for outlier detections:

- d' :
 - 1-back: $d'_{min} = 1.59$, $d'_{max} = 4.75$
 - 2-back: $d'_{min} = 0.33$, $d'_{max} = 3.81$
- Penalized reaction time:
 - 1-back: $pRT_{min} = 636\ ms$, $pRT_{max} = 1796\ ms$
 - 2-back: $pRT_{min} = 1004\ ms$, $pRT_{max} = 2009\ ms$
- Accuracy:
 - 1-back: $acc_{min} = 61.5\%$, $acc_{max} = 100\%$
 - 2-back: $acc_{min} = 25.4\%$, $acc_{max} = 100\%$

Applying these criteria to subjects’ behavioral data, we found single outlier score for d' measured during more demanding 2-back condition ($d' = 0.20$). However, this subject was neither classified as an outlier in 1-back condition when all three behavioral measures were considered, nor in 2-back condition when pRT and accuracy were considered. Therefore, we did not exclude this subject from the subsequent analysis.

We also performed outlier detection for the normalized modularity values from both task conditions. We used the same criterion as for behavioral measures. Outlier detection thresholds for the 1-back condition were $Q_{min} = 1.88$ and $Q_{max} = 4.03$, and for the 2-back

condition were $Q_{min} = 1.62$ and $Q_{max} = 3.97$. No subjects met these outlier criterion in either the 1-back or the 2-back conditions.

Note that for some subjects, the change in the normalized modularity or pRT was substantial; however, both their performance and network segregation did not differ significantly from other observations when analyzed in the context of a single task condition.

Code for the outlier detection can be found here:

https://github.com/kfinc/wm-training-modularity/blob/master/01-behavioral_data_analysis/03-exclusion_variability.ipynb

Comment 3: 3. Please provide the statistic results comparing the pRT between 1-back and 2-back conditions after training for the controls groups.

As requested, we now provide the statistical results comparing the pRT between 1-back and 2-back conditions in the *Behavioral changes during training* subsection of the *Results* section:

Interestingly, in the experimental group we observed no significant difference in performance between the 1-back condition and the 2-back condition after training ($t(20) = 1.52, p = 0.14$), while in control group, the difference in performance between conditions remained substantial (Bonferroni-corrected, $t(20) = -5.71, p < 0.001$).

However, due to the aforementioned limitations of pRT measure, in the revised version of the manuscript, we decided to report behavioral results as estimated with d' measure:

Interestingly, in the experimental group we observed no significant difference in performance between the 1-back condition and the 2-back condition after training ($t(20) = 0.02, p = 0.98$), while in the control group, the difference in performance between conditions remained substantial (Bonferroni-corrected, $t(20) = 4.91, p < 0.0016$). This finding suggests that the 2-back condition, which was much more effortful before training ('Naive' phase), was performed effortlessly after training, at the same level as the 1-back task.

Comment 4: 4. Before calculating the whole-brain modularity, the authors estimate correlations between ROI time-courses. However, they only retain the positive correlations for further analysis which do not make sense because the antagonism is typically connected patterns in functional brain networks, especially when one of their major interests is the integration between antagonistic frontoparietal and default mode systems. It is better to include the negative correlations in the analysis. For example, use alternative modularity detection algorithm which can work on negative edges or use the absolute value of the correlation matrix.

Response: We appreciate the reviewer's suggestion. We did not include negative edge weights, since we were mostly interested in the integration between the FP and the DM systems, and not their antagonism *per se*. Also, note that recent findings challenge the

common view regarding antagonism between these two systems and highlight the importance of positive FP-DM coupling during demanding cognitive tasks (Cocchi et al., 2013). Moreover, the coupling between FP and DM predicts individual differences in performance during working memory task (Murphy *et al.*, 2019). By excluding negative connections reflecting the competition between large-scale systems, we highlighted the system synchronization required for the formation of a global workspace (Dehaene et al., 1998).

However, we agree with the reviewer that zeroing negative edges could inevitably remove some information contained in the data. To investigate this, we repeated our whole-brain analysis using multilayer community detection on unthresholded, signed connectivity matrices (Traag & Bruggeman, 2009; Zhang *et al.*, 2017). Then, we applied the same analysis steps as for the networks with zeroed negative edges presented in the main manuscript. First, we summarized assignments of nodes to modules across layers in a form of module allegiance matrix \mathbf{P} , where each element P_{ij} represents a proportion of network layers for which nodes i and j were assigned to the same module. Second, we normalized allegiance matrices, to remove any potential bias introduced by differences in the number of nodes within each subsystem (see our response to comment #1 by reviewer #1). Third, we recalculated all recruitment and integration coefficients using functional cartography framework described by Mattar *et al.* (2015).

Supplementary Methods section:

Multilevel community detection for signed networks

We ran multilayer community detection on networks with both positive and negative edges (Traag & Bruggeman, 2009; Zhang *et al.*, 2017), to investigate whether the antagonism between large-scale systems (reflected by anticorrelated time-series) could influence the recruitment and integration values. First, we defined $N \times N$ matrix A_{ijs}^+ by zeroing negative elements of A_{ijs} and $N \times N$ matrix A_{ijs}^- by zeroing positive elements of A_{ijs} . We used this decomposition to represent both A_{ijs} and the corresponding null model p_{ijs} as a linear combination of networks with positive and networks with negative edges:

$$A_{ijs} = A_{ijs}^+ - A_{ijs}^-$$

$$\gamma_s p_{ijs} = \gamma_s^+ p_{ijs}^+ - \gamma_s^- p_{ijs}^-$$

Then, we maximized following modularity quality function:

$$Q_{\pm} = \frac{1}{\mu} \sum_{ijsr} \left[\left(A_{ijs}^+ - \gamma_s^+ \frac{k_{is}^+ k_{js}^+}{2m_s^+} \right) - \left(A_{ijs}^- - \gamma_s^- \frac{k_{is}^- k_{js}^-}{2m_s^-} \right) \right] \delta_{is} \delta_{ir}$$

With this approach we consider the negative network edges as separate networks when calculating within-layer modularity.

Code for the signed version of multilayer community detection can be found here: https://github.com/kfinc/wm-training-modularity/blob/master/04-dynamic_FC_analyses/08-multilayer_modularity_calculation_signed.m

First we used Pearson correlation to test whether values of unsigned and signed versions of the recruitment or integration coefficients were similar. When considering all large-scale systems, values of unsigned and signed version of recruitment and integration were highly correlated (recruitment: $r = 0.97$; $p = 0$; integration: $r = 0.97$, $p = 0$; Supplementary Figure 13ab). We also found high correlation of signed and unsigned version of the integration between frontoparietal and default mode systems ($r = 0.98$, $p = 0$; Supplementary Figure 13c).

Supplementary Figures:

Supplementary Figure 13 | Relationship between recruitment and integration values calculated based on unsigned and signed functional connectivity matrices. Unsigned and signed recruitment (a) and integration (b) coefficients estimated for all large-scale systems were highly correlated. (c) Values of integration between fronto-parietal (FP) and default mode (DM) systems were also highly correlated. Source data are provided as a Source Data file.

Next, we investigated whether signed version of recruitment or integration coefficients changed over the course of training in the same way as unsigned version of these measures. We reran multilevel modeling analysis to test for possible main session effects and session \times group interactions. Compared to the results on unsigned networks presented in the main text, for signed networks we observed very similar pattern of training-related changes when focusing on the FP and DM systems. Both experimental and control groups increased default mode recruitment from 'Naive' to 'Late' stage of training. Only the experimental group exhibited increase in fronto-parietal recruitment across sessions. In both groups, the integration between the fronto-parietal and default mode systems decreased from 'Naive' to 'Late' session, but the pattern of changes was different between groups from 'Naive' to 'Middle' session (Supplementary Figure 14).

Supplementary Figures:

Supplementary Figure 14 | **Changes in module allegiance of the fronto-parietal (FP) and default-mode (DM) systems calculated for signed functional connectivity matrices.** We observed a significant session \times group interaction effect when considering changes in the recruitment of the fronto-parietal system during training ($\chi^2(3) = 9.31, p = 0.025$) (a). The largest increase in fronto-parietal recruitment was observed in the experimental group when comparing 'Early' to 'Late' training phases ($\beta = -0.07, t(120) = -3.057, p = 0.016$, Bonferroni-corrected). No significant changes from 'Naive' to 'Late' training phases were observed in the control group ($\beta = -0.05, t(120) = -2.35, p = 0.12$, Bonferroni-corrected). (b) Turning to an examination of the default mode, we found a significant main effect of session ($\chi^2(3) = 23.89, p < 0.0001$) on system recruitment. However, the interaction effect between session and group was not significant ($\chi^2(3) = 2.00, p = 0.57$). Planned contrasts revealed that the default mode recruitment increased steadily in both groups and we observed the largest increase between 'Naive' and 'Late' sessions ($\beta = 0.08, t(123) = 5.02, p < 0.0001$). (c) We found a significant session \times group interaction effect on the integration between the fronto-parietal and default mode systems ($\chi^2(3) = 13.30, p = 0.004$). The integration between these two systems decreased from 'Early' to 'Late' sessions only in the experimental group ($\beta = 0.08, t(120) = 4.86, p = 0.0035$, Bonferroni-corrected). However, groups differed from 'Naive' to 'Early' ($\beta = 0.05, t(120) = 2.13, p = 0.03$) and from 'Early' to 'Middle' sessions ($\beta = -0.06, t(120) = -2.81, p = 0.02$). Source data are provided as a Source Data file.

In the revised version of the manuscript we also decided to focus more on changes of the recruitment or integration in other large scale systems (see our response to the comment #1 by the reviewer #1). Here, we repeated the same multilevel modeling analysis using recruitment and integration coefficients calculated for the signed functional networks. Similar to our analysis based on unsigned connectivity matrices, we identified three distinct types of changes regardless of the group: (1) increased recruitment of multiple systems (including default mode and salience), (2) decreased integration between DM and task-positive systems (fronto-parietal, cingulo-opercular, salience), and (3) increased integration between task-positive systems (compare signed version presented at Supplementary Figure 15a to unsigned version presented at Figure 6c). This pattern of findings suggests that the DM system gradually increases its autonomy, while the task-positive systems become more integrated over time. We also found several session \times group interaction effects mainly for subcortical, dorsal-attention, and cingulo-opercular systems (compare signed version presented at Supplementary Figure 15b to unsigned version presented at Figure 6e).

Supplementary Figures (signed version):

Supplementary Figure 15 | Changes of the recruitment and integration of large-scale systems calculated based on signed functional connectivity matrices. Colored tiles represent all significant effects ($p < 0.05$, uncorrected; $*p < 0.05$ FDR-corrected). (a) Here we display the significant main effects of session. Tile color codes a linear regression coefficient (β), for all main session effects (from 'Naive' to 'Late'). (b) Here we display the significant session \times group interaction effects. Tile color codes a linear regression coefficient between groups and sessions (from 'Naive' to 'Late'). Abbreviations: auditory (AU), cerebellum (CER), cingulo-opercular (CO), default mode (DM), dorsal attention (DA), fronto-parietal (FP), memory (MEM), salience (SAL), somatomotor (SOM), subcortical (SUB), uncertain (UNC), ventral attention (VA), and visual (VIS). Source data are provided as a Source Data file.

Results section (unsigned version; see Figure 6c and Figure 6e):

Figure 6 / Changes of the recruitment and integration of large-scale systems. Colored tiles represent all significant effects ($p < 0.05$, uncorrected; $*p < 0.05$ FDR-corrected). (top panel) Here we display the significant main effects of session. Tile color codes a linear regression coefficient (β), for all main session effects: (a) from ‘Naive’ to ‘Early’, (b) from ‘Naive’ to ‘Middle’, add (c) from ‘Naive’ to ‘Late’. (bottom panel) Here we display the significant session \times group interaction effects. Tile color codes a linear regression coefficient between groups and sessions: (a) from ‘Naive’ to ‘Early’, (b) from ‘Naive’ to ‘Middle’, add (c) from ‘Naive’ to ‘Late’. Abbreviations: auditory (AU), cerebellum (CER), cingulo-opercular (CO), default mode (DM), dorsal attention (DA), fronto-parietal (FP), memory (MEM), salience (SAL), somatomotor (SOM), subcortical (SUB), uncertain (UNC), ventral attention (VA), and visual (VIS). Source data are provided as a Source Data file.

Taken together, we conclude that our results and conclusions presented in the main manuscript are robust when considering both signed and unsigned functional connectivity matrices. Importantly, we did not observe substantial differences for recruitment and integration of the FP and DM system, which confirms that antagonism between these systems is a negligible factor not confounding our main results. We decided to keep analyses based on unsigned connectivity matrices and to present results based on signed connectivity matrices in the *Supplementary Information*.

Comment 5: 5. In Figure 3 b, it is said that “The greater the decrease in modularity from 1-back to 2-back, the smaller the decline in performance, as measured by pRT, from 1- back to 2-back”. It seems like the difference is calculated by “1-back minus 2-back” because the difference of pRT is all negative from the x-axis. However, there are more negative values along the y-axis, indicating that 1-back condition has low modularity than 2-back condition, which is not the case showing Figure 3 a. Please clarify this.

Response: We appreciate the reviewer noticing this inconsistency. Indeed, Figure 3b displays incorrect values of ΔpRT . On the x-axis we plotted $\Delta pRT = pRT_{1-back} - pRT_{2-back}$ instead of the appropriate $pRT_{2-back} - pRT_{1-back}$. We have fixed this issue in the new version of Figure 3b. As we mentioned in the answer for one of this same reviewer’s previous comments, we also discovered a mistake that we made when calculating the measure of penalized reaction time. We have since corrected the mistake and re-run all subsequent calculations. We found that new, corrected values differed only slightly from the original ones. Similarly, we found that the correlation presented in Figure 3b remained significant ($r=-0.30$, $p=0.04$). We did not find any relationship between the change in modularity (2-back minus 1-back) and the change in d' (2-back - 1-back) in the ‘Naive’ session. In the revised version of the manuscript we added additional figures presenting the relationship between the change of modularity and the change of behavioral performance (measured as pRT during and d') for ‘Naive’ session, all session, and the change from ‘Naive’ to ‘Late’ session (see Supplementary Figure 20).

Supplementary Figures:

Supplementary Figure S20: Relationship between modularity and behavioral performance. (a) We observed a weak negative correlation between the change (Δ) of modularity (2-back - 1-back) and the change in penalized reaction time (Δ pRT) during ‘Naive’ session. (b) Change of modularity was not related to the changes in pRT when considered all scanning sessions. (e, f) We did not observe any relationship between the change in d' and the change in modularity for ‘Naive’ and for all scanning sessions. (c, d, g, h) The change of modularity during 2-back was not correlated to the changes in pRT or d' from ‘Naive’ to ‘Late’ session. Source data are provided as a Source Data file.

Comment 6: 6. There is a clear group difference between experimental and controls groups on the modularity in the naive session, especially for the 1-back task. Can you provide a statistical comparison between them? Such difference exists even when there is no difference between the pRT, which can be potential confounding effects of the analysis and needs to be clarified.

Response: The functional network modularity did not differ between groups during the first ('Naive') scanning session (1-back: $t(39.92) = 1.37$, $p = 0.35$, Bonferroni-corrected; 2-back: $t(39.51) = 1.92$, $p = 0.12$, Bonferroni-corrected).

Note that testing and reporting group differences at baseline in randomized control trials is widely criticised by statisticians as it defies the logic of hypothesis testing (Austin *et al.*, 2010; Harvey, 2018). Accordingly, we decided to not include these results in the main manuscript. We also note that we used a statistical approach that allowed us to correct for baseline individual differences; the multilevel modeling analysis directly controls for baseline values.

Comment 7: 7. In figure 3 e and f, although the experimental group shows more difference between naive and late sessions, such group difference is not significantly larger than the difference for the control group, resulting in less solid conclusion.

Response: We agree that this difference was not statistically significant and we reported this result in our *Results* section. Modularity increased in the experimental (1-back and 2-back) and control group (1-back only), suggesting that the shift toward a more segregated network organization happens both as a result of training and repeated exposure to the task. Note, that as participants from the control group performed the dual n-back task four times in the fMRI scanner, they also trained the task to a small extent. Indeed, the control group displayed significantly better performance during the 2-back condition when comparing 'Naive' to 'Late' session, yet this change was significantly smaller than a change observed in the experimental group. Many previous studies investigating the effects of learning on the network reorganization did not include a control group; therefore, they were not able to capture such repeated exposure effects. These results also suggest that the network modularity is a highly sensitive measure that can capture immediate changes in task-related network organization. As modularity increased in both groups, the effect of training might be too small and not detectable with our sample size (power = ~ 0.30 for detecting small effect for within-between interactions in repeated measures design).

When solely considering the network changes in the experimental group, we clearly observe an increase in modularity for both conditions. These results extend upon prior research on task-related network reorganization, by showing that the baseline modularity during the effortful task can change in response to training or repeated exposure. Future studies should examine whether we can observe a gradation of modularity changes after training with varying intensity. We rewrote the concluding paragraph of *Result* section:

These results indicate that the experimental group displays increased network modularity for both task conditions when moving from 'Naive' to 'Late' sessions, suggesting that network segregation may be a consequence of the 6-week working memory training. While the same effect was not present in the control group, we did not observe a significant group \times session interaction, and therefore further work is needed to inform our conclusions.

We also added a paragraph to the *Discussion* section:

Interestingly, we did not observe differences between the experimental and control group in the increase of network modularity. The control group displayed a small increase of modularity in the 1-back condition, suggesting that the segregation of the functional brain network may increase rapidly, also in response to repeated exposure to the task. The control group performed the dual n-back task four times during scanning sessions, which resulted in a small behavioral improvement. This result suggests that the increase of network segregation may be sensitive to varying intensity of training in the task. Future studies with a larger sample size should examine whether such gradation exists.

Comment 8: 8. There is no correlation between the decrease modularity and the increase pRT. The authors argue that is because the change of modularity is a general consequence of training. However, the repeated exposure also results in decrease modularity (1-back condition). If the

modularity decrease caused by training is not related to the improved performance, how can authors conclude that the working memory training may help to prevent cognitive decline?

Response: We based this speculative part of the *Discussion* on studies that reported a link between high brain network modularity and cognitive plasticity (see Gallen *et al.*, 2019 for review). Moreover, several studies reported age-related decrease of modularity (Chan *et al.*, 2014, Song *et al.*, 2014, Onoda *et al.*, 2013) that was also related to a higher cognitive decline (Chan *et al.*, 2014). From these results, we speculate that an increased modularity in response to training can potentially help to prevent cognitive decline.

Now the question is why we observed no correlation between the decrease in modularity caused by training and improved performance in our sample. One possible explanation is that there might exist a ceiling effect on performance levels causing the correlation to be unobservable after a certain amount of training. We tested this correlating the change of modularity and the change of pRT during 2-back condition from 'Naive' to 'Early' sessions in the experimental group and all subjects, however we also did not observe a significant relationship ($r = -0.07$ and $r = -0.12$ respectively, $p > 0.05$). We note that for most participants, we observed both decreased modularity and decreased penalized reaction time which may suggest an as yet undefined link between network segregation and behavioral performance. For example, in our analysis of dynamic modularity changes, we observed that the recruitment of the default mode (DM) increased from 'Naive' to 'Late' session and that this increase was related to a higher performance improvement. In further exploratory analysis we showed that DM recruitment fluctuated between task conditions and was significantly higher in the 1-back condition than in the 2-back condition ($t(167) = -10.43$, $p < 0.00001$) (Supplementary Figure 19). Similar to modularity, the default mode recruitment also increased steadily in both groups and we observed the largest increase between 'Naive' and 'Late' sessions ($t(123) = 5.02$, $p < 0.0001$). In response to the reviewer's comment, we also decided to test whether DM recruitment was related to static modularity. We observed a strong positive correlations between DM recruitment and static modularity during 1-back ($r = 0.71$, $p < 0.0001$, Supplementary Figure 21a) and 2-back ($r = 0.59$, $p < 0.0001$, Supplementary Figure 21b) conditions. We also observed that the increase of DM recruitment from 'Naive' to 'Late' session was positively correlated to changes of modularity in 1-back condition ($r = 0.41$, $p = 0.007$; Supplementary Figure 21c). Trend towards positive correlation was also observed for the increased modularity during 2-back condition ($r = 0.30$, $p = 0.056$; Supplementary Figure 21d). Such relationship between DM functional connectivity and modularity changes during n-back task was also observed in our previous study (Finc *et al.*, 2017).

Supplementary Figure 21 | Relationship between default mode recruitment and static modularity. Correlation between DM recruitment and modularity during (a) 1-back and (b) 2-back conditions calculated for all subjects and all sessions. Correlation between the change (Δ) of DM recruitment and change of modularity during (c) 1-back condition and (d) 2-back condition. Source data are provided as a Source Data file.

As static modularity changes constitute a reflection of specific changes in the dynamic network topology, we may speculate that focusing on the recruitment and integration coefficients of all large-scale systems provides us a more detailed picture of the brain network reorganization. This results also suggests that DM recruitment coefficient estimated with the dynamic network approach may provide a more precise prediction of behavioral performance than static modularity.

We also added the following fragment in the *Discussion* section to refer to this result:

Our observations expand upon prior studies by demonstrating that the increase in default mode segregation and decrease of integration between the default mode and fronto-parietal systems may be an indicator of behavioral improvement during working memory training. Moreover, the previous study reported the relationship between the default mode connectivity changes and static modularity changes during n-back task (Finc et al., 2017). In our exploratory analysis, we also showed that default mode recruitment fluctuated between task conditions and was significantly higher in the 1-back condition than in the 2-back condition (Supplementary Figure 19) and, similar to modularity, increased steadily in both groups. Here

we also observed a positive relationship between the change in default mode recruitment and change of modularity from ‘Naive’ to ‘Late’ session (Supplementary Figure 21). As we did not observe the relationship between changes of modularity and behavioral improvement, we may conclude that studying the dynamics of modular network structure enables a better prediction of behavioral outcomes in response to training.

Moreover, to ensure that future readers appreciate the fact that more work is needed to assess the relationship between modularity and training-induced performance, we have now added a sentence at the end of the paragraph in question.

Importantly, modularity is also altered in patients with disorders of mental health or patients sustaining brain injury. Studies have found that modular organization of a network is disrupted in patients with cognitive control deficits (Alexander-Bloch et al., 2010), and increases over the early stage of stroke recovery in a manner that is related to the recovery of higher cognitive functions (Siegel et al., 2018). Further longitudinal studies in these patient populations could provide clarity on the role of modularity -- and its variation over a range of time scales -- in higher-order cognitive function. Our findings suggest that there is a possibility to increase brain network modularity via intensive working memory training. This phenomenon may have potential beneficial implications for designing cognitive training interventions to prevent aging-related cognitive decline, reduce cognitive control deficits, or intensify effects of neurorehabilitation through increasing brain plasticity. To verify this, future studies should examine the direct effect of training-induced increase of brain modularity in healthy ageing and clinical populations.

Comment 9: 9. Can the authors provide the statistic results by comparing recruitment between experimental group and control group in the naive session? If the difference is significant, please discuss this in the discussion.

Response: We tested whether group differences exist during the ‘Naive’ session for each network recruitment and pairwise integration. We did not observe group differences that passed FDR correction for multiple comparisons, but we did observe some differences that crossed uncorrected statistical threshold (see Table 10 in this response). For the recruitment value, we indeed observed a difference in the fronto-parietal system ($p = 0.015$, uncorrected) and in the auditory system ($p = 0.01$, uncorrected). We did not observe any characteristic pattern of these differences compared to other network interactions (see for example Supplementary Figure 7). Note that the p -values less than 0.05 among many baseline statistical tests (in this case $n_{\text{tests}} = 91$) may reflect a spurious findings (Harvey, 2018). Also, in the case of randomized control trials, testing and reporting group differences at baseline is widely criticised by statisticians (Austin et al., 2010). Subjects were randomly assigned to either control or experimental groups after the first fMRI session and we did not observe any age, IQ, gender, or behavioral performance differences. Thus, we are reluctant to interpret the observed differences in recruitment in the FP and auditory systems in the ‘Naive’ session, as they are likely due to individual baseline differences rather than any effects of

training relevant to our experimental manipulations. Finally, we note that we used a statistical approach that allowed us to correct for these baseline individual differences; the multilevel modeling analysis directly controls for baseline values.

Supplementary Figure 7 | Session-to-session changes in recruitment and integration of large-scale systems. We observed three main categories of large-scale system reorganization: (a-c) an increase in system recruitment, (d-f) an increase in integration between task-positive systems (TP), (g-i) a decrease in integration between the default mode (DM) system and task-positive systems, and (j-k) other. Error bars represent 95% confidence intervals. DM - default mode, SAL - salience, AU-auditory, FP - fronto-parietal, DA - dorsal attention, CO - cingulo-opercular, MEM - memory, and SOM-somatomotor. Source data are provided as a Source Data file.

Moreover in the Discussion section we added:

Here, we used a multilayer community detection algorithm to determine whether modular structure of large-scale systems change in response to n-back training. We further applied multilevel modeling (Snijders et al., 2012) to test for possible group and session differences in the dynamic network measures while controlling for differences in individual baseline values. In testing our hypothesis, we held in mind the observations of previous studies, which have noted that the fronto-parietal and default mode systems can both cooperate and compete

during tasks that require cognitive control, such as the n-back task (Spreng et al., 2010, Cocchi et al., 2013).

Comment 10: 10. In figure 5, is there any multiple comparison correction performed? Also, such associations are significant in both groups, which might suggest that DM recruitment changes resulted from the repeated exposure will be the major cause of improved performance.

Response: In the original version of the manuscript we tested only three dependent variables: DM system recruitment, FP system recruitment, and FP-DM integration. We did not use multiple comparison corrections for these tests. Due to helpful feedback from reviewers, we now consider the recruitment of all systems and the integration between all pairs of systems. This change has vastly increased the number of statistical tests that we performed, from 3 in the original version of the manuscript to 91 in the revised version of the manuscript. We therefore now include multiple comparison corrections throughout. In the Figure 6 and Figure 7 (see our response to comment #4), we present all significant effects, uncorrected and corrected using the FDR method.

As the reviewer noted, the DM recruitment increased significantly when both groups were considered. Interestingly, the group \times session interaction effect was not significant for DM recruitment, suggesting that the effect results from repeated exposure to the task. A higher increase in DM recruitment from 'Naive' to 'Late' sessions was also correlated with an increase in behavioral performance reflected by the change in $\Delta d'$ and ΔpRT . As the reviewer mentioned, this relationship suggests that repeated exposure to the task is also related to an increased independence of the DM system, which may be sufficient to improve behavioral performance.

We clarified this in the *Discussion* section:

Similar to modularity, the lack of group differences in the pattern of these changes suggests that the increased DM autonomy and increased integration of task-positive systems might be related to a general improvement in task performance. Such behavioral improvement, although much smaller than in the experimental group, was also observed in the control group during the 2-back condition. As participants performed the task four times in the scanner, they inevitably trained the task to a small extent. The presence of network reorganization in the control group may suggest that changes in DM autonomy and integration of task-positive systems occur relatively fast, even when the training is not intense. As participants of our study were scanned in 2-week intervals, we could not capture what behavioral improvement is necessary to invoke such network reorganization. To better understand the dynamics of these neuroplastic changes, future studies should examine day-to-day network reorganization in response to training with different intensities

Comment 11: 11. In summary, the overall results cannot fully support that the reconfiguration of the brain network due to the training can improve the working memory performance. The authors need more analysis and discussion to clarify this.

Response: The main goal of the present study was to investigate the dynamics of the brain network reorganization during a demanding cognitive task. We based our research question on previous studies that reported functional network organization is changing in response to varying demands of the working memory task (Finc *et al.*, 2012; Braun *et al.*, 2015, Vatansever *et al.*, 2015). In particular, these studies reported that during less demanding task conditions, the functional brain network becomes more segregated. It remained unclear how this task-related brain network reorganization is altered in response to training. A previous longitudinal study on motor sequence training provided evidence that the network may become more segregated in response to learning (Bassett *et al.*, 2015). No study described a pattern of dynamic network reorganization during more complex tasks, such as a working memory n-back task. Our present study covers the existing gap and provides such a description.

In response to comments from reviewers, we performed additional tests on the relationship between the change in the dynamic network organization of all large-scale systems and the change in the behavioral performance of participants. First, we examined whether the increase of autonomy of the default mode system and increased integration of task-positive systems between ‘Naive’ and ‘Late’ session was associated with increased performance. Specifically, we found the change of DM recruitment was positively correlated with the change of behavioral performance as measured by the change in $\Delta d'$ ($r = 0.33$, $p = 0.03$, uncorrected). Moreover, the decrease of its integration with the frontoparietal and salience systems was related to behavioral improvement ($r = -0.31$, $p = 0.04$ and $r = -0.44$, $p = 0.006$ respectively; uncorrected), as well as the increase of integration of the frontoparietal and salience systems ($r = 0.35$, $p = 0.02$, uncorrected) and the increase of salience network recruitment ($r = 0.35$, $p = 0.03$, uncorrected) (see Figure 7a and Supplementary Table 6). We also observed that the experimental and control group differed in dynamic network changes in particular between ‘Naive’ and ‘Early’ sessions. Similarly, we observed the largest behavioral improvement in the experimental group within this time interval. Accordingly, we tested whether this behavioral change was related to specific changes in the network organization during the early phase of training. We observed that better behavioral outcomes in response to intense working memory training were associated with integration between multiple systems, including system pairs for which we also observed group \times session interactions: subcortical and dorsal attention, somatomotor and dorsal attention, and subcortical and cingulo-opercular systems (Figure 7b and Table 7). We observed consistent results for the Δ pRT measure. Note that correlations had opposite signs as a lower Δ pRT denotes a better behavioral performance (Supplementary Figure 9 and Supplementary Table 7). We believe that these findings provide additional support for the claim that dynamic reconfiguration of the functional brain systems is associated with improvements in working memory performance.

Figure 7 | Relationship between the change in network dynamics and the change in behavior. Colored tiles represent all significant correlations ($p < 0.05$, uncorrected; $*p < 0.05$ FDR-corrected). (a) Pearson correlation coefficient (r) between the across-session changes in recruitment (or integration) and the across-session changes in d' ($\Delta d'$) observed for both experimental and control group. (b) Relationship between the changes in recruitment (or integration) and the changes in d' during early phase of training of the experimental group. Abbreviations: auditory (AU), cerebellum (CER), cingulo-opercular (CO), default mode (DM), dorsal attention (DA), fronto-parietal (FP), memory (MEM), salience (SAL), somatomotor (SOM), subcortical (SUB), uncertain (UNC), ventral attention (VA), and visual (VIS). Source data are provided as a Source Data file.

Supplementary Figure 9 | Relationship between the change in network dynamics and the change in behavior. Colored tiles represent all correlations ($p < 0.05$, uncorrected; $*p < 0.05$ FDR-corrected). (a) Pearson correlation coefficient (r) between the across-session changes in recruitment (or integration) and the across-session changes in penalized reaction time (ΔpRT) observed for both experimental and control group. (b) Relationship between the changes in recruitment (or integration) and the changes in pRT during early phase of training of the experimental group. (...)

Moreover in the *Discussion section* we added:

Enhanced default mode intra-communication and decreased inter-communication with the fronto-parietal system were associated with better behavioral outcomes after training.

(...)

The fronto-parietal system dynamically interacts with other large-scale systems (Cocci et al., 2013), and it is reasonable to expect that working memory training might influence interactions in the whole network. We observed training-related increases in the segregation of the default mode and task-positive systems that suggest more efficient and less costly processing within these systems after training. Accordingly, greater segregation of the default mode system and task-positive systems and smaller integration between these systems were associated with behavioral performance improvement. Moreover, we showed that an increase of integration between multiple large-scale systems in early phase of training was related to a greater behavioral improvement in the experimental group, indicating that some level of network integration is necessary when the task is not fully automated. Taken together, the dynamic network approach provides a unique insight into the plasticity and dynamics of the human brain network.

References

- Austin, P. C., Manca, A., Zwarenstein, M., Juurlink, D. N., & Stanbrook, M. B. (2010). A substantial and confusing variation exists in handling of baseline covariates in randomized controlled trials: a review of trials published in leading medical journals. *Journal of Clinical Epidemiology*, 63(2), 142-153.
- Bassett, D. S., Yang, M., Wymbs, N. F., & Grafton, S. T. (2015). Learning-induced autonomy of sensorimotor systems. *Nature Neuroscience*, 18(5), 744.
- Braun, U., Schäfer, A., Walter, H., Erk, S., Romanczuk-Seiferth, N., Haddad, L., ... & Meyer-Lindenberg, A. (2015). Dynamic reconfiguration of frontal brain networks during executive cognition in humans. *Proceedings of the National Academy of Sciences*, 112(37), 11678-11683.
- Chan, M. Y., Park, D. C., Savalia, N. K., Petersen, S. E., & Wig, G. S. (2014). Decreased segregation of brain systems across the healthy adult lifespan. *Proceedings of the National Academy of Sciences*, 111(46), E4997-E5006.
- Dehaene, S., Kerszberg, M., & Changeux, J. P. (1998). A neuronal model of a global workspace in effortful cognitive tasks. *Proceedings of the National Academy of Sciences*, 95(24), 14529-14534.

- Fair, D. A., Schlaggar, B. L., Cohen, A. L., Miezin, F. M., Dosenbach, N. U., Wenger, K. K., ... & Petersen, S. E. (2007). A method for using blocked and event-related fMRI data to study “resting state” functional connectivity. *Neuroimage*, 35(1), 396-405.
- Finc, K., Bonna, K., Lewandowska, M., Wolak, T., Nikadon, J., Dreszer, J., ... & Kühn, S. (2017). Transition of the functional brain network related to increasing cognitive demands. *Human Brain Mapping*, 38(7), 3659-3674.
- Gallen, C. L., & D’Esposito, M. (2019). Brain Modularity: A Biomarker of Intervention-related Plasticity. *Trends in Cognitive Sciences*.
- Garavan, H., Kelley, D., Rosen, A., Rao, S. M., & Stein, E. A. (2000). Practice-related functional activation changes in a working memory task. *Microscopy research and technique*, 51(1), 54-63.
- Harvey, L. A. (2018). Statistical testing for baseline differences between randomised groups is not meaningful. *Spinal Cord*, 56, 919.
- Hempel, A., Giesel, F. L., Garcia Caraballo, N. M., Amann, M., Meyer, H., Wüstenberg, T., ... & Schröder, J. (2004). Plasticity of cortical activation related to working memory during training. *American Journal of Psychiatry*, 161(4), 745-747.
- Kelly, A. C., & Garavan, H. (2004). Human functional neuroimaging of brain changes associated with practice. *Cerebral Cortex*, 15(8), 1089-1102.
- Kühn, S., Schmiedek, F., Noack, H., Wenger, E., Bodammer, N. C., Lindenberger, U., & Lövdén, M. (2013). The dynamics of change in striatal activity following updating training. *Human Brain Mapping*, 34(7), 1530-1541.
- Landau, S. M., Garavan, H., Schumacher, E. H., & D’Esposito, M. (2007). Regional specificity and practice: dynamic changes in object and spatial working memory. *Brain Research*, 1180, 78-89.
- Mattar, M. G., Cole, M. W., Thompson-Schill, S. L., & Bassett, D. S. (2015). A functional cartography of cognitive systems. *PLoS Computational Biology*, 11(12), e1004533.
- Murphy, A. C., Bertolero, M. A., Papadopoulos, L., Lydon-Staley, D. M., & Bassett, D. S. (2019). Multiscale and multimodal network dynamics underpinning working memory. arXiv preprint arXiv:1901.06552.
- Onoda, K., & Yamaguchi, S. (2013). Small-worldness and modularity of the resting-state functional brain network decrease with aging. *Neuroscience Letters*, 556, 104-108.
- Power, J. D., Cohen, A. L., Nelson, S. M., Wig, G. S., Barnes, K. A., Church, J. A., Vogel, A. C., Laumann, T. O., Miezin, F. M., Schlaggar, B. L., et al. (2011). Functional network organization of the human brain. *Neuron*, 72(4):665–678.

- Snijders, T. A. B., & Bosker, R. J. (2012). Discrete dependent variables. Multilevel analysis: an introduction to basic and advanced multilevel modeling, 304-307.
- Song, J., Birn, R. M., Boly, M., Meier, T. B., Nair, V. A., Meyerand, M. E., & Prabhakaran, V. (2014). Age-related reorganizational changes in modularity and functional connectivity of human brain networks. *Brain Connectivity*, 4(9), 662-676.
- Traag, V. A., & Bruggeman, J. (2009). Community detection in networks with positive and negative links. *Physical Review E*, 80(3), 036115.
- Vatansever, D., Menon, D. K., Manktelow, A. E., Sahakian, B. J., & Stamatakis, E. A. (2015). Default mode dynamics for global functional integration. *Journal of Neuroscience*, 35(46), 15254-15262.
- Zhang, H., Wang, C. D., Lai, J. H., & Philip, S. Y. (2017, December). Modularity in complex multilayer networks with multiple aspects: a static perspective. In *Applied Informatics* (Vol. 4, No. 1, p. 7). SpringerOpen.

Supplementary Tables

Repeated measures MLM: main effect of session			
Systems	Statistics		
	χ^2	p _{uncorr.}	p _{FDR}
Recruitment			
SAL	24.3038	0.0000	0.0005
DM	24.1711	0.0000	0.0005
AU	14.7401	0.0021	0.0208
MEM	9.8965	0.0195	0.1181
SUB	8.5157	0.0365	0.1747
Integration with default mode (DM) system			
DM-SAL	31.3720	0.0000	0.0001
DM-AU	20.1747	0.0002	0.0024
DM-FP	19.7550	0.0002	0.0025
DM-CO	15.8257	0.0012	0.0140
Integration with task-positive systems			
FP-SAL	28.8229	0.0000	0.0001
DA-SAL	21.8571	0.0001	0.0013
CO-DA	14.0292	0.0029	0.0261
MEM-SOM	13.1715	0.0043	0.0354
SAL-MEM	11.0954	0.0112	0.0785
SAL-SUB	10.0385	0.0182	0.1181
FP-CO	9.7186	0.0211	0.1201
CO-VIS	9.499441	0.023337	0.124923
SAL-CO	8.9760	0.0296	0.1497
FP-AU	8.2521	0.0411	0.1869
VA-UNC	7.9361	0.0474	0.2032
Other			
AU-SOM	11.6143	0.0088	0.0669

Supplementary Table 1 | Results of the multilevel modeling (MLM) analysis reflecting main session effects for systems recruitment or integration (4 sessions). In all cases, random intercepts were estimated. The significance of models was estimated with chi-square tests, where models with increasing complexity were compared and the resulting value of Likelihood Ratio Test (χ^2) and corresponding p-value (uncorrected and FDR-corrected) were reported. Abbreviations: auditory (AU), cerebellum (CER), cingulo-opercular (CO), default mode (DM), dorsal attention (DA), fronto-parietal (FP), memory (MEM), salience (SAL), somatomotor (SOM), subcortical (SUB), uncertain (UNC), ventral attention (VA), and visual (VIS).

Planned contrasts: main effect of session						
Systems	Naive vs. Early		Naive vs. Middle		Naive vs. Late	
	β	p	β	p	β	p
Recruitment						
SAL	0.0259	0.0279	0.041229	0.0006	0.057214	0.0000
DM	0.0358	0.0528	0.052197	0.0051	0.091959	0.0000
AU	0.0377	0.0127	0.0314	0.0369	0.0572	0.0002
MEM	0.0292	0.1145	0.016409	0.3740	0.05636	0.0027
SUB	0.0409	0.0145	0.015256	0.3569	0.039226	0.0190
Integration with default mode (DM) system						
DM-SAL	-0.0378	0.0008	-0.047378	0.0000	-0.0623	0.0000
DM-AU	-0.0237	0.0579	-0.0174	0.1633	-0.0560	0.0000
DM-FP	-0.0116	0.3310	-0.021237	0.0758	-0.051636	0.0000
DM-CO	-0.0273	0.0399	-0.0323	0.0154	-0.0529	0.0001
Integration with task-positive systems						
FP-SAL	0.0395	0.0120	0.062749	0.0001	0.082824	0.0000
DA-SAL	0.0420	0.0009	0.05039	0.0001	0.051894	0.0001
CO-DA	0.0271	0.0512	0.0356	0.0109	0.0511	0.0003
MEM-SOM	-0.0408	0.0201	-0.030599	0.0798	-0.062644	0.0004
SAL-MEM	0.0109	0.4989	0.015714	0.3298	0.050995	0.0019
SAL-SUB	0.0364	0.0037	0.029535	0.0179	0.027495	0.0272
FP-CO	0.0276	0.0736	0.0349	0.0240	0.0459	0.0032
CO-VIS	0.024047	0.0535	0.022812	0.0668	0.037705	0.0027
SAL-CO	0.0257	0.0152	0.0192	0.0683	0.0285	0.0072
FP-AU	-0.0167	0.1912	-0.0025	0.8442	0.0196	0.1242
VA-UNC	-0.0103	0.1671	-0.020728	0.0059	-0.008145	0.2726
Other						
AU-SOM	0.0407	0.0056	0.0337	0.0213	0.0446	0.0025

Supplementary Table 2 | Planned contrasts for all significant main session effects, reflecting changes of systems recruitment or integration (4 sessions). Contrasts: 'Naive' vs. 'Early', 'Naive' vs. 'Middle', 'Naive' vs. 'Late'.

Repeated measures MLM: session \times group interaction			
Systems	Statistics		
	χ^2	$p_{uncorr.}$	p_{FDR}
Recruitment			
FP	6.831	0.029	0.327
DA	2.697	0.041	0.339
Integration with default mode (DM) system			
DM-FP	19.755	0.003	0.235
Integration with subcortical (SUB) system			
SUB-DM	5.999	0.015	0.278
SUB-AU	5.843	0.02	0.309
SUB-VA	4.868	0.037	0.334
SUB-DA	1.193	0.026	0.327
SUB-CO	0.257	0.011	0.244
Integration with other task-positive systems			
DA-SOM	2.514	0.011	0.244
CO-MEM	0.865	0.034	0.334
CO-UNC	2.437	0.005	0.236

Supplementary Table 3 | Results of the multilevel modeling (MLM) analysis reflecting session \times group interaction effects for systems recruitment or integration (4 sessions, 2 groups). In all cases, random intercepts were estimated. The significance of models was estimated with chi-square tests, where models with increasing complexity were compared and the resulting value of Likelihood Ratio Test (χ^2) and corresponding p -value (uncorrected and FDR-corrected) were reported.

Planned contrasts: session \times group interaction						
Systems	Naive vs. Early		Naive vs. Middle		Naive vs. Late	
	β	p	β	p	β	p
Recruitment						
FP	-0.0690	0.0740	-0.0020	0.9570	0.0450	0.2490
DA	-0.0670	0.0200	-0.0160	0.5700	0.0050	0.8560
Integration with default mode (DM) system						
DM-FP	0.0490	0.0330	-0.0120	0.5890	-0.0340	0.1380
Integration with subcortical (SUB) system						
SUB-DM	-0.0380	0.1010	0.0360	0.1230	0.0040	0.8630
SUB-AU	0.0720	0.0230	-0.0200	0.5150	0.0250	0.4250
SUB-VA	0.0330	0.1970	-0.0090	0.7180	-0.0400	0.1180
SUB-DA	0.0140	0.6090	-0.0460	0.0850	-0.0510	0.0540
SUB-CO	0.0570	0.0350	-0.0280	0.2940	0.0270	0.3140
Integration with other task-positive systems						
DA-SOM	-0.0500	0.0690	0.0290	0.2850	-0.0420	0.1250
CO-MEM	-0.0530	0.0980	-0.0600	0.0630	-0.0920	0.0050
CO-UNC	0.0440	0.0040	0.0410	0.0080	0.0460	0.0030

Supplementary Table 4 | Planned contrasts for all significant session \times group interaction effects, reflecting group differences in changes of systems recruitment or integration (4 sessions, 2 groups). Contrasts: 'Naive' vs. 'Early', 'Naive' vs. 'Middle', 'Naive' vs. 'Late'.

Post-hoc tests: session \times group interaction				
Systems	Early vs. Middle		Middle vs. Late	
	β	$p_{\text{bonferroni}}$	β	$p_{\text{bonferroni}}$
Recruitment				
FP	0.0673	0.2499	0.0467	0.6842
DA	0.0506	0.2245	0.0212	1.0000
Integration with default mode (DM) system				
DM-FP	-0.0613	0.0237	-0.0216	1.0000
Integration with subcortical (SUB) system				
SUB-DM	0.0739	0.0052	-0.0318	0.5121
SUB-AU	-0.0919	0.0114	0.0453	0.4462
SUB-VA	-0.0418	0.2997	-0.0305	0.6836
SUB-DA	-0.0593	0.0786	-0.0055	1.0000
SUB-CO	-0.0854	0.0055	0.0554	0.1229
Integration with other task-positive systems				
DA-SOM	0.0790	0.0130	-0.0711	0.0300
CO-MEM	-0.0068	1.0000	-0.0321	0.9536
CO-UNC	-0.0033	1.0000	0.0054	1.0000

Supplementary Table 5 | Post-hoc tests for all significant session \times group interaction effects, reflecting group differences in changes of systems recruitment or integration (4 sessions, 2 groups). Tests: 'Naive' vs. 'Early', 'Early' vs. 'Middle', 'Middle' vs. 'Late'.

Correlation with behavior: Naive vs. Late (both groups)			
Systems	r	$P_{uncorr.}$	$PFDR$
$\Delta d'$ and Δ recruitment			
SAL	0.3377095	0.028723065	1
DM	0.3354091	0.029898227	1
$\Delta d'$ and Δ integration			
FP-SAL	0.3530235	0.021836677	1
MEM-VIS	0.3215687	0.037836231	1
SOM-VA	0.3198655	0.038922327	1
DM-FP	-0.3099325	0.04577489	1
DM-SAL	-0.4142908	0.006378749	0.5740874
AU-MEM	-0.4412674	0.003442311	0.3132503
Δ pRT and Δ recruitment			
DM	-0.3478555	0.02398683	1
VIS	-0.3378605	0.02864729	1
Δ pRT and Δ integration			
AU-MEM	0.3178699	0.04022718	1
CER-DM	0.3405356	0.02733201	1
CO-UNC	-0.3630498	0.01812347	1
CO-VIS	0.3338558	0.03071395	1
DM-FP	0.3560102	0.02066934	1

Supplementary Table 6 | Correlations between the change in network dynamics and the change in behavior. Pearson correlation coefficient (r) between the across-session changes (Naive vs. Late) in recruitment (or integration) and the across-session changes in d' ($\Delta d'$) and pRT (Δ pRT) observed for both the experimental and control groups.

Correlation with behavior: Naive vs. Early (experimental)			
Systems	r	$P_{uncorr.}$	$PFDR$
$\Delta d'$ and Δ integration			
AU-VA	0.4903851	0.0240127927	1
CER-DA	0.4585068	0.0365743723	1
CO-DA	0.4957494	0.0222872683	1
CO-SUB	-0.515905	0.0166670006	1
DA-SOM	0.710518	0.0003068366	0.02792213
DA-SUB	0.4700412	0.0315444066	1
DM-SAL	0.4558561	0.037814337	1
FP-SOM	0.449137	0.0411045974	1
Δ pRT and Δ integration			
AU-VA	-0.4498745	0.04073297	1
CER-DA	-0.4463136	0.042551842	1
CO-DA	-0.4645735	0.033856106	1
CO-SUB	0.5106186	0.018016262	1
DA-SOM	-0.5641078	0.007729913	0.7034221
UNC-VA	0.5189307	0.015932156	1

Supplementary Table 7 | Correlations between the change in network dynamics and the change in behavior in the experimental group. Pearson correlation coefficient (r) between the changes in recruitment (or integration) and the changes in d' ($\Delta d'$) and pRT (Δ pRT) during early phase of training (Naive vs. Early) of the experimental group.

Repeated measures MLM: session × goup interaction (GLM)			
System	Statistics		
	χ^2	$P_{uncorr.}$	$PFDR$
SAL	19.289622	0.000238	0.003096
VIS	12.21532	0.006681	0.043425
DM	10.220241	0.016784	0.060016
UNC	9.538138	0.022929	0.060016
FP	9.523477	0.023083	0.060016
MEM	8.635576	0.03455	0.074858
VA	7.979876	0.046429	0.086226
DA	5.976738	0.112747	0.183215
SOM	4.77535	0.189006	0.273008
CO	4.233999	0.23728	0.298229
SUB	4.085796	0.252347	0.298229
CER	3.326098	0.344027	0.372696
AU	2.794743	0.424366	0.424366

Supplementary Table 8 | Results of the multilevel modeling (MLM) analysis reflecting session × group interaction effects for systems activity estimated with a standard GLM (2-back vs. 1-back contrast, two-sided). In all cases, random intercepts were estimated. The significance of models was estimated with chi-square tests, where models with increasing complexity were compared and the resulting value of Likelihood Ratio Test (χ^2) and corresponding p-value (uncorrected and FDR-corrected) were reported.

Planned contrasts: session × group interaction effect (GLM)						
System	Naive vs. Early		Naive vs. Middle		Naive vs. Late	
	β	p	β	p	β	p
SAL	-1.09808	0.000812	-1.317647	0.000069	-0.956718	0.003351
VIS	0.121433	0.684265	-0.80268	0.00806	-0.002907	0.992229
DM	-0.750571	0.01187	-0.759561	0.010918	-0.779999	0.009004
UNC	-0.255562	0.27055	-0.689108	0.003441	-0.206578	0.372717
FP	-0.679019	0.095265	-1.235466	0.002735	-0.695647	0.087521
MEM	-0.574095	0.111209	-1.02874	0.004776	-0.401063	0.264535
VA	-0.404449	0.195934	-0.76255	0.01565	-0.732185	0.020184

Supplementary Table 9 | Planned contrasts for all significant session × group interaction effects, reflecting group differences in changes of systems activity estimated with a standard GLM (2-back vs. 1-back contrast, two-sided). Contrasts: 'Naive' vs. 'Early', 'Naive' vs. 'Middle', 'Naive' vs. 'Late'.

Group differences in Naive session			
Systems	t	$p_{uncorr.}$	$pFDR$
Recruitment			
FP	-2.5482	0.0148553	0.45061
AU	-2.6631	0.0111777	0.45061
Integration			
FP-VA	-2.7704	0.0084522	0.45061
AU-DA	-2.418	0.0205102	0.46661
DA-DM	2.2111	0.0328063	0.54232
AU-UNC	2.1189	0.0412432	0.54232
SOM-VIS	-2.1153	0.041717	0.54232

Table 10 | Results of the comparison of recruitment or integration coefficients values between groups during 'Naive' session, as estimated using two-sample t-test (two-sided).

Reviewers' Comments:

Reviewer #1:

Remarks to the Author:

The authors should be commended for an outstanding update to an already compelling paper. I am happy to recommend publication.

Reviewer #2:

Remarks to the Author:

I thank the authors for their thorough and thoughtful response to the reviews, and for their open scientific approach. Overall, this is a greatly improved manuscript, and a substantial contribution to the field.

(I did note a small typo on line 1058 of the revised text: "To enable for...")

Reviewer #3:

Remarks to the Author:

The authors have addressed all my concerns and I don't have further comments.

Response to Reviewers' Comments

Comments from Reviewer 1

The authors should be commended for an outstanding update to an already compelling paper. I am happy to recommend publication.

Response: We thank the reviewer for appreciating our work.

Comments from Reviewer 2

I thank the authors for their thorough and thoughtful response to the reviews, and for their open scientific approach. Overall, this is a greatly improved manuscript, and a substantial contribution to the field.

(I did note a small typo on line 1058 of the revised text: "To enable for...")

Response: We thank the reviewer for appreciating our work. We corrected an error on line 1058.

Comments from Reviewer 3

The authors have addressed all my concerns and I don't have further comments.

Response: We thank the reviewer for appreciating our work.